# A neural-level model of spatial memory and imagery

**Andrej Bicanski\*, Neil Burgess\***

Institute of Cognitive Neuroscience, University College London, London, United Kingdom

**Abstract** We present a model of how neural representations of egocentric spatial experiences in parietal cortex interface with viewpoint-independent representations in medial temporal areas, via retrosplenial cortex, to enable many key aspects of spatial cognition. This account shows how previously reported neural responses (place, head-direction and grid cells, allocentric boundary- and object-vector cells, gain-field neurons) can map onto higher cognitive function in a modular way, and predicts new cell types (egocentric and head-direction-modulated boundary- and object-vector cells). The model predicts how these neural populations should interact across multiple brain regions to support spatial memory, scene construction, novelty-detection, 'trace cells', and mental navigation. Simulated behavior and firing rate maps are compared to experimental data, for example showing how object-vector cells allow items to be remembered within a contextual representation based on environmental boundaries, and how grid cells could update the viewpoint in imagery during planning and short-cutting by driving sequential place cell activity.

DOI: https://doi.org/10.7554/eLife.33752.001

## Introduction

The ability to reconstruct perceptual experiences into imagery constitutes one of the hallmarks of human cognition, from the ability to imagine past episodes (Tulving 1985) to planning future scenarios (*Schacter et al., 2007*). Intriguingly, this ability (also known as 'scene construction' and 'episodic future thinking') appears to depend on the hippocampal system (*Schacter et al., 2007*; *Hassabis et al., 2007*; *Buckner, 2010*), in which direct (spatial) correlates of the activities of single neurons have long been identified in rodents (*O'Keefe and Nadel, 1978*; *Taube et al., 1990a*; *Hafting et al., 2005*) and more recently in humans (*Ekstrom et al., 2003*; *Jacobs et al., 2010*). The rich catalog of behavioral, neuropsychological and functional imaging findings on one side, and the vast literature of electrophysiological research on the other (see e.g. *Burgess et al., 2002*), promises to allow an explanation of higher cognitive functions such as spatial memory and imagery directly in terms of the interactions of neural populations in specific brain areas. However, while attaining this type of understanding is a major aim of cognitive neuroscience, it cannot usually be captured by a few simple equations because of the number and complexity of the systems involved. Here, we show how neural activity could give rise to spatial cognition, using simulations of multiple brain areas whose predictions can be directly compared to experimental data at neuronal, systems and behavioral levels.

Extending the Byrne, Becker and Burgess model of spatial memory and imagery of empty environments (*Burgess et al., 2001a*; *Byrne et al., 2007*), we propose a large-scale systems-level model of the interaction between Papez' circuit, parietal, retrosplenial, and medial temporal areas. The model relates the neural response properties of well-known cells types in multiple brain regions to cognitive phenomena such as memory for the spatial context of encountered objects and mental navigation within familiar environments. In brief, egocentric (i.e. body-centered) representations of the local sensory environment, corresponding to a specific point of view, are transformed into

\*For correspondence:
a.bicanski@ucl.ac.uk (AB);
n.burgess@ucl.ac.uk (NB)

**Competing interest:** See
page 30

**Reviewing editor:** Laura Colgin,
The University of Texas at Austin,
Center for Learning and
Memory, United States

viewpoint-independent (allocentric or world-centered) representations for long-term storage in the medial temporal lobes (MTL). The reverse process allows reconstruction of viewpoint-dependent egocentric representations from stored allocentric representations, supporting imagery and recollection.

Neural populations in the medial temporal lobe (MTL) are modeled after cell types reported in rodent electrophysiology studies. These include place cells (PCs), which fire when an animal traverses a specific location within the environment (*O'Keefe and Dostrovsky, 1971*); head direction cells (HDCs), which fire according to the animal's head direction relative to the external environment, irrespective of location (*Taube and Ranck, 1990*; *Taube et al., 1990a*; *Taube et al., 1990b*); boundary vector cells (*Lever et al., 2009*); henceforth BVCs), which fire in response to the presence of a boundary at a specific combination of distance and allocentric direction (i.e. North, East, West, South, irrespective of an agent's orientation); and grid cells (GCs), which exhibit multiple, regularly spaced firing fields (*Hafting et al., 2005*). Evidence for the presence of these cell types in human and non-human primates is mounting steadily (*Robertson et al., 1999*; *Ekstrom et al., 2003*; *Jacobs et al., 2010*; *Doeller et al., 2010*; *Bellmund et al., 2016*; *Horner et al., 2016*; *Nadasdy et al., 2017*).

The egocentric representation supporting imagery has been suggested to reside in medial parietal cortex (e.g. the precuneus; *Fletcher et al., 1996*; *Knauff et al., 2000*; *Formisano et al., 2002*; *Sack et al., 2002*; *Wallentin et al. (2006)*; *Hebscher et al., 2018*). In the model, it is referred to as the 'parietal window' (PW). Its neurons code for the presence of scene elements (boundaries, landmarks, objects) in peri-personal space (ahead, left, right) and correspond to a representation along the dorsal visual stream (the 'where' pathway; *Ungerleider, 1982*; *Mishkin et al., 1983*). The parietal window boundary coding (PWb) cells are egocentric analogues of BVCs (*Barry et al., 2006*; *Lever et al., 2009*), consistent with evidence that parietal areas support egocentric spatial processing (*Bisiach and Luzzatti, 1978*; *Nitz, 2009*; *Save and Poucet, 2009*; *Wilber et al., 2014*).

The transformation between egocentric (parietal) and allocentric (MTL) reference frames is performed by a gain-field circuit in retrosplenial cortex (*Burgess et al., 2001a*; *Byrne et al., 2007*; *Wilber et al., 2014*; *Alexander and Nitz, 2015*; *Bicanski and Burgess, 2016*), analogous to gain-field neurons found in posterior parietal cortex (*Snyder et al., 1998*; *Salinas and Abbott, 1995*; *Pouget and Sejnowski, 1997*; *Pouget et al., 2002*) or parieto-occipital areas (*Galletti et al., 1995*). Head-direction provides the gain-modulation in the transformation circuit, producing directionally modulated boundary vector cells which connect egocentric and allocentric boundary coding neurons. That this transformation between egocentric directions (left, right, ahead) and environmentally-referenced directions (nominally North, South, East, West) requires input from the head-direction cells found along Papez's circuit (*Taube et al., 1990a*; *Taube et al., 1990b*) is consistent with its involvement in episodic memory (e.g. *Aggleton and Brown, 1999*; *Delay and Brion, 1969*).

During perception the egocentric parietal window representation is based on (highly processed) sensory inputs. That is, it is driven in a bottom-up manner, and the transformation circuit maps the egocentric PWb representation to allocentric BVCs. When the transformation circuit acts in reverse (top-down mode), it reconstructs the parietal representation from BVCs which are co-active with other medial temporal cell populations, forming the substrate of viewpoint-independent (i.e. allocentric) memory. This yields an orientation-specific (egocentric) parietal representation (a specific point of view) and constitutes the model's account of (spatial) imagery and explicit recall of spatial configurations of known spaces (*Burgess et al., 2001a*; *Byrne et al., 2007*). *Figure 1* depicts a simplified schematic of the model.

To account for the presence of objects within the environment, we propose allocentric object vector cells (OVCs) analogous to BVCs, and show how object-locations can be embedded into spatial memory, supported by visuo-spatial attention. Importantly, the proposed object-coding populations in the MTL map onto recently discovered neuronal populations (*Deshmukh and Knierim, 2013*; *Hoydal et al., 2017*). We also predict a population of egocentric object-coding cells in the parietal window (PWo cells: egocentric analogues to OVCs), as well as directionally modulated boundary and object coding neurons (in the transformation circuit). Finally, we include a grid cell population to account for mental navigation and planning, which drives sequential place cell firing reminiscent of hippocampal 'replay' (*Wilson and McNaughton, 1994*; *Foster and Wilson, 2006*; *Diba and Buzsáki, 2007*; *Karlsson and Frank, 2009*; *Carr et al., 2011*) and preplay (*Dragoi and Tonegawa, 2011*; *Ólafsdóttir et al., 2015*). We refer to this model as the BB-model.

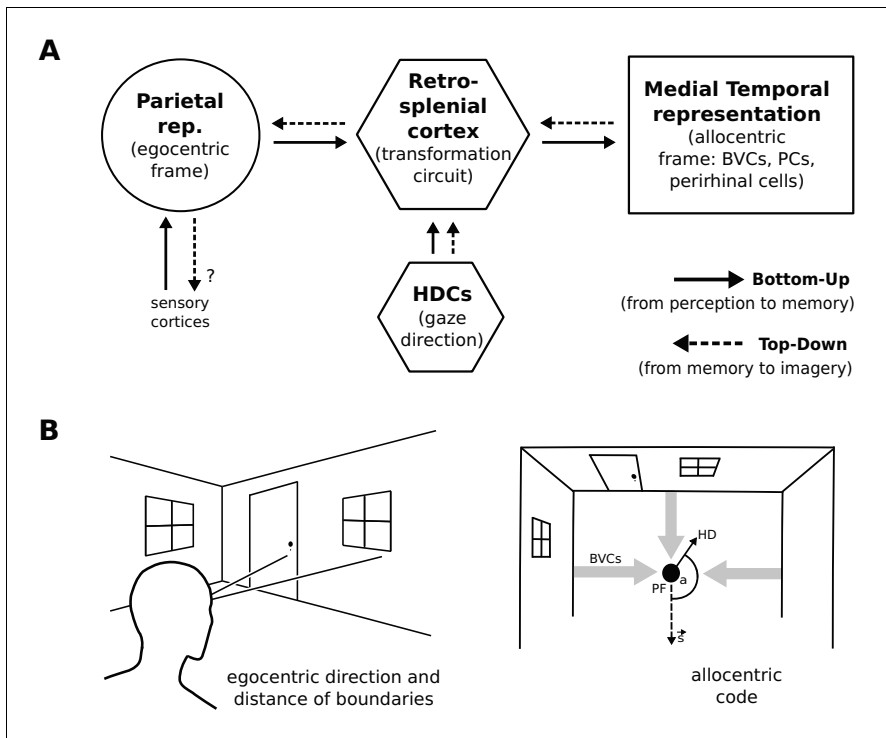

**Figure 1.** Simplified model schematic. (**A**) Processed sensory inputs reach parietal areas and support an egocentric representation of the local environment (in a head-centered frame of reference). Retrosplenial cortex uses current head or gaze direction to perform the transformation from egocentric to allocentric coding. At a given location, environmental layout is represented as an allocentric code by activity in a set of BVCs, the place cells (PCs) corresponding to the location, and perirhinal neurons representing boundary identities (in a familiar environment, all these representations are associated via Hebbian learning to form an attractor network). Black arrows indicate the flow of information during perception and memory encoding (bottom-up). Dotted arrows indicate the reverse flow of information, reconstructing the parietal representation from view-point invariant memory (imagery, top-down). (**B**) Illustration of the egocentric (left panel) and allocentric frame of reference (right panel), where the vector *s* indicates South (an arbitrary reference direction) and the angle *a* is coded for by head direction cells, which modulate the transformation circuit. This allows BVCs and PCs to code for location within a given environmental layout irrespective of the agent's head direction (HD). The place field (PF, black circle) of an example PC is shown together with possible BVC inputs driving the PC (broad grey arrows).

DOI: https://doi.org/10.7554/eLife.33752.002

## Methods

Here, we describe the neural populations of the BB-model and how they interact in detail. Technical details of the implementation, equations, and parameter values can be found in the Appendix.

### Receptive field topology and visualization of data

We visualize the firing properties of individual spatially selective neurons as firing rate maps that reflect the activity of a neuron averaged over time spent in each location. We also show population activity by arranging all neurons belonging to one population according to the relative locations of their receptive fields (see *Figure 2A–C*), plotting a snapshot of their momentary firing rates. In the case of boundary-selective neurons such a population snapshot will yield an outline of the current sensory environment (*Figure 2C*). Naturally, these neurons may not be physically organized in the same way, and these plots should not be confused with the firing rate maps of individual neurons (*Figure 2D*). Hence, population snapshots (heat maps) and firing rate maps (Matlab 'jet' colormap) are shown in distinct color-codes (*Figure 2*).

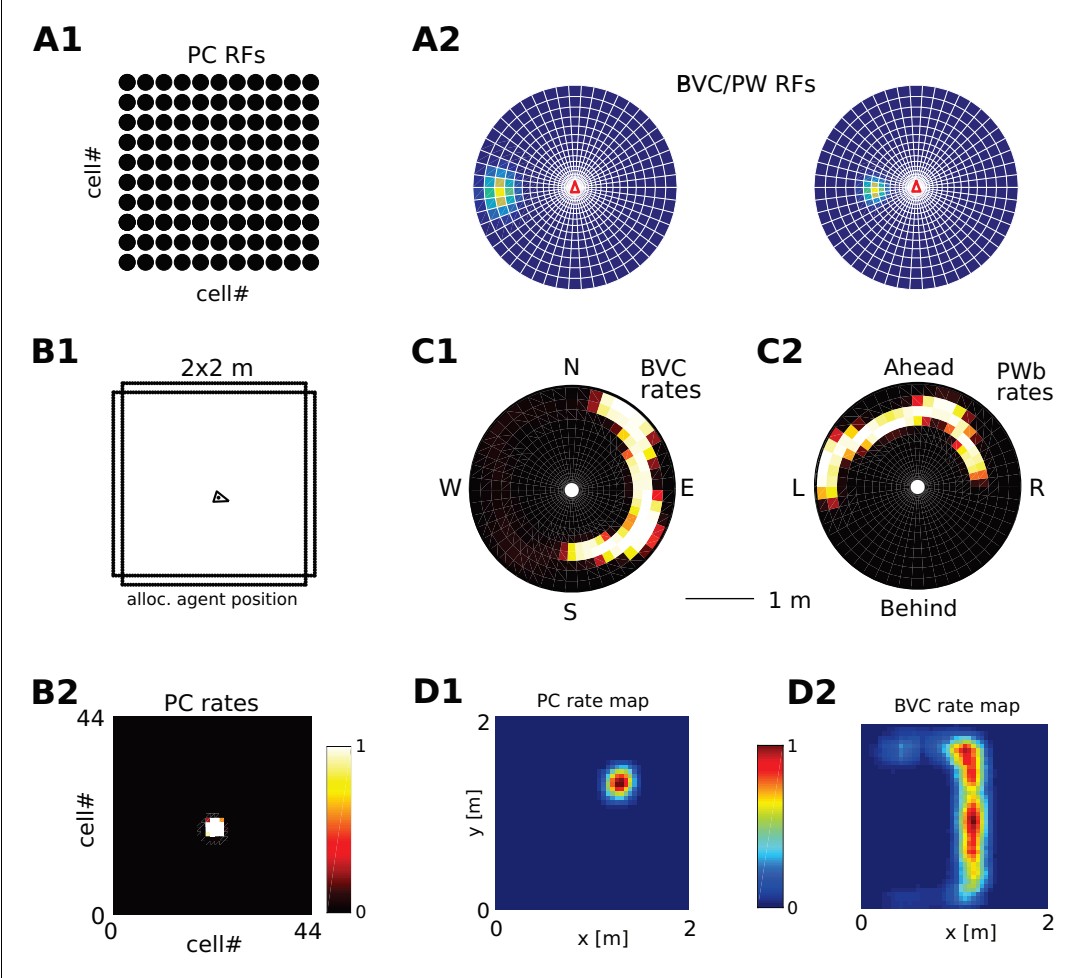

**Figure 2.** Receptive field topology and visualization of neural activity. (**A1**) Illustration of the distribution of receptive field centers (RFs) of place cells (PCs), which tile the environment. (**A2**) Receptive fields of boundary responsive neurons, be they allocentric (BVCs) or egocentric (PWb neurons), are distributed on a polar grid, with individual receptive fields centered on each delineated polygon. Two example receptive fields (calculated according to *Equation 14*) are overlaid (bright colors) on the polar grids for illustration. Note that each receptive field covers multiple polygons, that is neighboring receptive fields overlap. The polar grids of receptive fields tile space around the agent (red arrow head at center of plots), that is they are anchored to the agent and move with it (for both BVCs and PWb neurons). In addition, for PWb neurons the polar grid of receptive fields also rotates with the agent (i.e. their tuning is egocentric). (**B1**) As the agent (black arrowhead) moves through an environment, place cells (B2) track its location. (**B2**) Snapshot of the population activity of all place cells arranged according to the topology of their firing fields (see A1). (**C1,2**) Snapshots of the population activity for BVCs and boundary selective PW neurons (PWb), respectively. Cells are again distributed according to the topology of their receptive fields (see A2), that is each cell is placed at the location occupied by the centre of its receptive field in peri-personal space (ahead is shown as up for PW neurons; North is shown as up for BVCs). See Section on the transformation circuit, *Video 1*, and *Figure 2—figure supplement 1* for the mapping between PW and BVCs patterns via the transformation circuit. (**D1,2**) Unlike snapshots of population activity, firing rate maps show the activity of individual neurons averaged over a whole trial in which the agent explores the environment, here for a place cell (D1) and for a boundary vector cell with a receptive field due East (D2, tuning distance roughly 85 cm).

DOI: https://doi.org/10.7554/eLife.33752.003

The following figure supplement is available for figure 2:

**Figure supplement 1.** Caption: Illustration of single cell coding in the retrosplenial transformation circuit.
DOI: https://doi.org/10.7554/eLife.33752.004

## The parietal window

Perceived and imagined egocentric sensory experience is represented in the 'parietal window' (PW), which consists of two neural populations - one coding for extended boundaries ('PWb neurons'), and one for discrete objects ('PWo neurons'). The receptive fields of both populations lie in peri-personal space, that is are tuned to distances and directions ahead, left or right of the agent, tile the ground

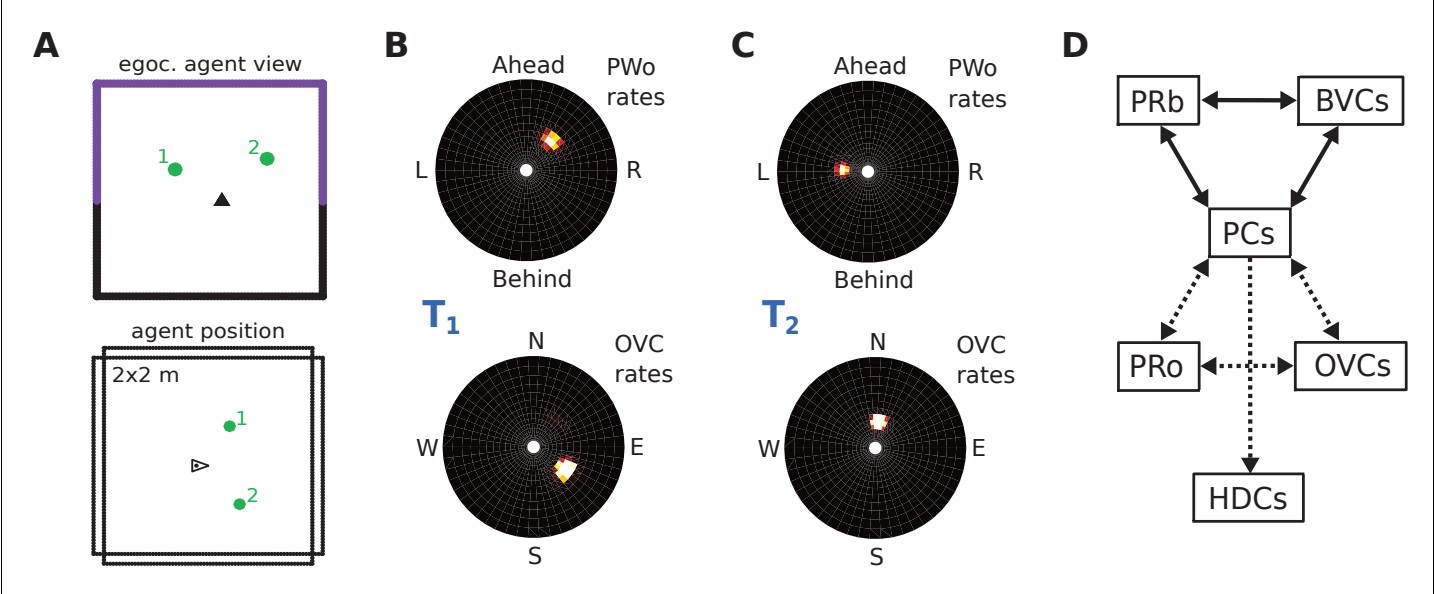

**Figure 3.** The agent model and population snapshots for object representations. (A) Top panel: The egocentric field of view of the agent (black arrow head). Purple boundaries fall into the forward-facing 180 degree field of view and provide bottom-up drive to the parietal window (PWb; not shown, but see *Figure 2C2*). The environment contains two discrete objects (green circles). Bottom panel: Allocentric positions of the agent (black triangle) and objects (green circles). (B) Object-related parietal window (PWo) activity (top panel) and OVC activity (bottom panel) due to object 2, South-East of the agent, at time T1. (C) PWo activity (top panel) and OVC activity (bottom panel) due to object 1, North-East of the agent, at time T2. A heuristically implemented attention model ensures that only one object at a time drives the parietal window (PWo). (D) Illustration of the encoding of an object encountered in a familiar environment. Dashed connections are learned (as Hebbian weight updates) between active cells. Solid lines indicate connections learned in the training phase, representing the spatial context. Note that place cells (PCs) anchor the object representation to the spatial context.

DOI: https://doi.org/10.7554/eLife.33752.006

around the agent, and rotate together with the agent (*Figure 2A2*, *Figure 3*). Reciprocal connections to and from the retrosplenial transformation circuit (RSC/TR, see below) allow the parietal window representations to be transformed into allocentric (orientation-independent) representations (i.e. boundary and object vector cells) in the MTL and vice versa. Intriguingly, cells that encode an egocentric representation of boundary locations (akin to parietal window neurons in the present model) have recently been described (*Hinman et al., 2017*).

## The agent model and perceptual drive

An agent model supplies perceptual information, driving the parietal window in a bottom-up manner. The virtual agent moves along trajectories in simple 2D environments (*Figure 2B1*). Turning motions of the agent act on the head direction network to shift the activity packet in the head direction ring attractor. Egocentric distances to environmental boundaries in a 180-degree field of view in front of the agent are used to drive the corresponding parietal window (PWb) neurons. The retrosplenial circuit (section "The Head Direction Attractor Network and the Transformation Circuit") transforms this parietal window activity into BVC activity, which in turn drives PC activity in the pattern-completing MTL network (*O'Keefe and Burgess, 1996*; *Hartley et al., 2000*). Thus, simplified perceptual drive conveyed to the MTL allows the model to self-localize in the environment based purely on sensory inputs.

## The medial temporal lobe network
### Spatial context

The medial temporal lobe (MTL) network for spatial context is comprised of three interconnected neural populations: the PCs and BVCs code for the position of the agent relative to a given boundary configuration, and perirhinal neurons code for the identity (e.g. texture, color etc) of boundaries

(PRb neurons). Identity has to be signaled by cells separate from BVCs because the latter respond to any boundary at a given distance and direction.

## Discrete objects

The allocentric object code is comprised of two populations of neurons. First, similarly to extended boundaries, the identity of discrete objects must be coded for by perirhinal neurons (PRo neurons). Second, we hypothesize an allocentric representation of object location, termed object vector cells (OVCs), analogous to BVCs (*Figure 3B*), with receptive fields at a fixed distance in an allocentric direction.

Interestingly, cells which respond to the presence of small objects and resemble OVCs have recently been identified in the rodent literature (*Deshmukh and Knierim, 2013*; *Hoydal et al., 2017*), and could reside in the hippocampus proper or one synapse away. Although we treat them separately, BVCs and OVCs could in theory start out as one population in which individual cells specialize to respond only to specific types of object with experience (e.g. to small objects in the case of OVCs; see *Barry and Burgess, 2007*).

## The role of perirhinal neurons

OVCs, like BVCs and parietal window (PWo and PWb) neurons signal geometric relations between object or boundary locations and the agent, but not the identity of the object or boundary. OVCs and BVCs fire for any object or boundary occupying their receptive fields. Conversely, an object's or boundary's identity is indicated, irrespective of its location, by perirhinal neurons. They lie at the apex of the ventral visual stream (the 'what' pathway; *Ungerleider, 1982*; *Mishkin et al., 1983*; *Goodale and Milner, 1992*; *Davachi, 2006*; *Valyear et al., 2006*) and encode the identities or sensory characteristics of boundaries and objects, driven by a visual recognition process which is not explicitly modeled. Only in concert with perirhinal identity neurons does the object or boundary code uniquely represent a specific object or boundary at a specific direction and distance from the agent.

## Connections among medial temporal lobe populations

BVCs and OVCs have reciprocal connections to the transformation circuit, allowing them to be driven by perceptual inputs ('bottom up'), or to project their representations to the parietal window ('top down').

For simulations of the agent in a familiar environment, the connectivity among the medial temporal lobe populations which comprise the spatial context (PCs, BVCs, PRb neurons) is learned in a training phase, resulting in an attractor network, such that mutual excitatory connections between neurons ensure pattern completion. Hence, partial activity in a set of PCs, BVCs, and/or PRb neurons - will re-activate a complete, previously learned representation of spatial context in these populations. OVCs and PRo neurons are initially disconnected from the populations that represent the spatial context. The simulated agent can then explore the environment and encode objects into memory along the way.

## **The head direction attractor network and the transformation circuit**

Head direction cells (HDCs) are arranged in a simple ring-attractor circuit (*Skaggs et al., 1995*; *Zhang, 1996*). Current head direction, encoded by activity in this attractor circuit, is updated by angular velocity information as the agent explores the environment. The head direction signal enables the egocentric-allocentric transformation carried out by retrosplenial cortex.

Because of their identical topology, the PWb/BVC population pair and the PWo/OVC population pair can each make use of the same transformation circuit. For simplicity we illustrate its function via BVCs and their PWb counterparts. The retrosplenial transformation circuit (RSC/TR) consists of 20 sublayers. Each sublayer is a copy of the BVC population, with firing within each sublayer also tuned to a specific head-direction (directions are evenly spaced in the [0360] degree range). That is, individual cells in the transformation circuit are directionally modulated boundary vector cells, and connect egocentric (parietal) PWb neurons and allocentric BVCs (in the MTL) in a mutually consistent way. All connections are reciprocal. For example, a BVC with a receptive field to the East is mapped onto a PWb neuron with a receptive field to the right of the agent when facing North, but is mapped

onto a PWb neuron with a receptive field to the left of the agent when facing South. Similarly, a PWb neuron with a receptive field ahead of the agent is mapped onto a BVC with a receptive field to the West when facing West but is mapped onto a BVC with a receptive field to the North when facing North. *Figure 2C* depicts population snapshots that are mapped onto each other by the transformation circuit (also see *Video 1*), while *Figure 2—figure supplement 1* illustrates the connections and firing rate maps at the single cell level. We hypothesize that the egocentric-allocentric transformation circuit is set up during development (see Appendix for the setup of the circuit).

## Bottom-up vs top-down modes of operation

During perception, the egocentric parietal window representation is based on sensory inputs ('bottom-up' mode). The PW representations thus determine MTL activity via the transformation circuit. 'Running the transformation in reverse' ('top-down' mode), that is reconstructing parietal window activity based on BVCs/OVCs, is the BB-models account of visuo-spatial imagery. To implement the switch between modes of operation, we assume that the balance between bottom-up and top-down connections is subject to neuromodulation (see e.g. *Hasselmo, 2006*); Appendix, *Equation 3* and following). For example, connections from the parietal window (PWb and PWo) populations to the transformation circuit and thence onto BVCs/OVCs are at full strength in bottom-up mode, but down-regulated to 5% of their maximum value in top-down mode. Conversely, connections from BVCs/OVCs to the transformation circuit and onwards to the parietal window are down-regulated during bottom-up perception (5% of their maximum value) and reach full strength only during imagery (top-down reconstruction).

## Embedding object-representations into a spatial context: attention and encoding

Unlike boundaries, which are hard-coded in the simulations (corresponding to the agent moving in a familiar environment), object representations are learned on the fly (simulating the ability to remember objects found in new locations in the environment).

As noted above (section "The Role of Perirhinal Neurons"), to uniquely characterize the egocentric perceptual state of encountering an object within an environment requires the co-activation of perirhinal (PRo) neurons (signaling identity) and the corresponding parietal window (PWo) (signaling location in peripersonal space). Moreover, maximal co-firing of only one PRo neuron with one PWo neuron (or OVC, in allocentric terms) at a given location is required for an unambiguous association (*Figure 3A–C*). If multiple conjunctions of object location and identity are concurrently represented then it is impossible to associate each object identity uniquely with one location - that is, object-location binding would be ambiguous. To ensure a unique representation, we allow the agent to direct attention to each visible object in sequence (compare *Figure 3B and C*; for a review of attentional mechanisms see *VanRullen, 2013*). This leads to

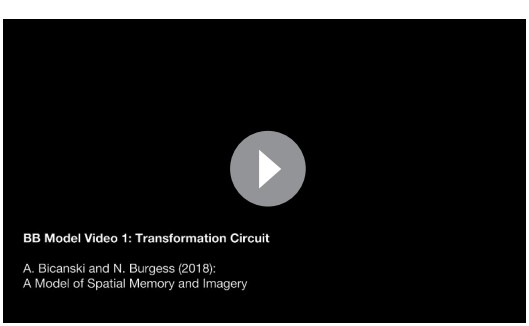

**Video 1.** Surface plots (heat maps) visualize theneural activity of populations of cells. The video shows a visualization of the simulated neural activity in the retrosplenial transformation circuit as a simulated agent moves in a simple, familiar environment (See Figure 2-figure supplement 1 for further details). Individual sublayers of the transformation circuit are shown in a circular arrangement around the head direction ring. Head direction cells track the agent's heading and confer a gain modulation on the retrosplenial sublayers. The transformation circuit then drives boundary vector cells (see main text). Surface plots (heat maps) visualize the neural activity of populations of cells. Individual cells correspond to pixels/polygons on the heat maps (compare to figures). Cells are arranged according to the distribution of their receptive fields; however, this arrangement does not necessarily reflect anatomical relations. Bright colors indicate strong firing. Abbreviations: PWb, Parietal Window, egocentric boundary representations (ahead is up); HDCs, Head Direction Cells; TR, Retrosplenial transformation sublayers; BVCs, Boundary Vector Cells (North is up); egoc. agent view, egocentric field of view of the agentwithin the environment, purple outlines denote visibleboundary segments which correspond to sensoryinputs to the PWb (ahead is up); alloc. agent position, allocentric position of the agent in the environment (North is up).
DOI: https://doi.org/10.7554/eLife.33752.005

a specific set of PWo, OVC and PRo neurons, corresponding to a single object at a given location, being co-active for a short period while connections between MTL neurons develop (including those with PCs, see *Figure 3D*). Then, attention is redirected and a different set of PWo, OVC and PRo neurons becomes co-active. We set a fixed length for an attentional cycle (600 time units). However, we do not model the mechanistic origins of attention. Attention is supplied as a simple rhythmic modulation of perceptual activity in the parietal window.

To encode objects in their spatial context the connections between OVCs, PRo neurons and currently active PCs are strengthened. By linking OVCs and PRo neurons to PCs, the object code is explicitly attached to the spatial context because the same PCs are reciprocally connected to the BVCs that represent the geometric properties of the environment (*Figure 3D*). A connection between PRo neurons and HDCs is also strengthened to allow recall to re-instantiate the head direction at encoding during imagery (see Simulation 1.0 below).

Finally, if multiple objects are present in a scene we do not by default encode all perceivable objects equally strongly into memory. We trigger encoding of an object when it reaches a threshold level of 'salience'. In general, 'salience' could reflect many factors; here, we simulate relatively few objects and assume that salience becomes maximal at a given proximity, and prevent any further learning thereafter.

## Imagery and the role of grid cells

Grid cells (GCs; Hafting et al. 2005) are thought to interface self-motion information with place cells (PCs) to enable vector navigation (*Kubie and Fenton, 2012*; *Erdem and Hasselmo, 2012*; *Bush et al., 2015*; *Stemmler et al., 2015*), shortcutting, and mental navigation (*Bellmund et al., 2016*); Horner et al. 2016. Importantly, both self-motion inputs (via GCs) and sensory inputs (e.g. mediated via BVCs and OVCs) converge onto PCs and both types of inputs may be weighted according to their reliability (*Evans et al., 2016*). GCs could thus support PC activity when sensory inputs are unreliable or absent. Here, GC inputs can drive PC firing during imagined navigation (see Section Novelty Detection (Simulations 1.3, 1.4)), whereas perceived scene elements, mediated via BVC and OVCs, provide the main input to PCs during unimpaired perception.

We include a GC module in the BB-model that, driven by heuristically implemented mock-motor-efference signals (self-motion signals with suppressed motor output), can update the spatial memory network in the absence of sensory inputs. The GC input allows the model to perform mental navigation (imagined movement through a known environment). By virtue of connections from GCs to PCs, the GCs can shift an activity bump smoothly along the sheet of PCs. Pattern completion in the medial temporal lobe network then updates the BVC representation according to the shifted PC representation. BVCs in turn update the parietal window representation (top-down), smoothly shifting the egocentric field of view in imagery (i.e. updating the parietal window representations) during imagined movement. Thus, self-motion related updating (sometimes referred to as 'path integration') and mental navigation share the same mechanism (*Tcheang et al., 2011*).

Connection weights between GCs and PCs are calculated as a simple Hebbian association between PC firing at a given coordinate (according to the mapping shown in *Figure 2A,B*) and pre-calculated firing rate maps of GCs (7 modules with 100 cells each, see Appendix for details).

## Model summary

An agent employing a simple model of attention alongside dedicated object-related neural populations in perirhinal, parietal and parahippocampal (BVCs and OVCs) cortices allow the encoding of scene representations (i.e. objects in a spatial context) into memory. Transforming egocentric representations via the retrosplenial transformation circuit yields viewpoint-independent (allocentric) representations in the medial temporal lobe, while reconstructing the parietal window representation (which is driven by sensory inputs during perception) from memory is the model's account of recall as an act of visuo-spatial imagery. Grid cells allow for mental navigation. *Figure 4* shows the complete schematic of the BB-model, see *Figure 2—figure supplement 1* for details of the RSC transformation circuit.

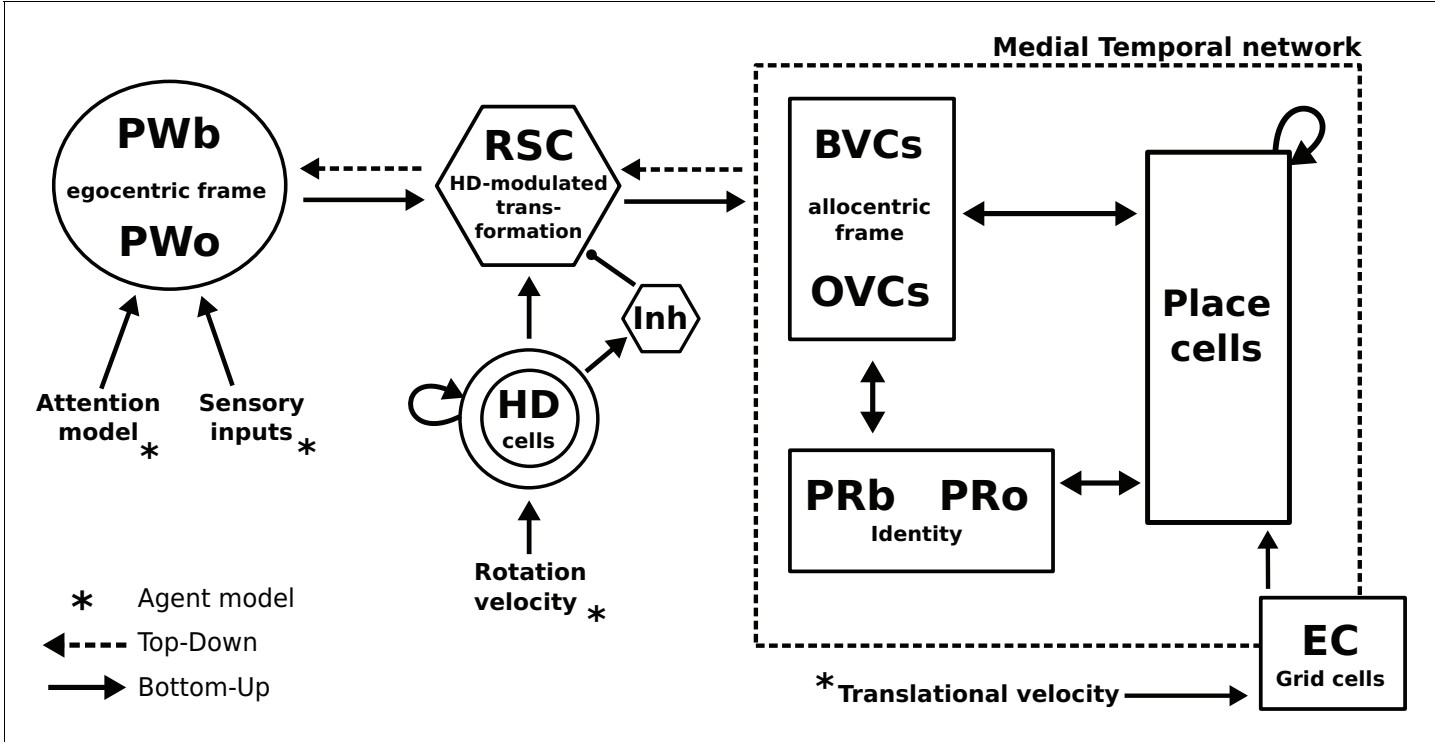

**Figure 4.** The BB-model. 'Bottom-up' mode of operation: Egocentric representations of extended boundaries (PWb) and discrete objects (PWo) are instantiated in the parietal window (PWb/o) based on inputs from the agent model while it explores a simple 2D environment. Attention sequentially modulates object-related PW activity to allow for unambiguous neural representations of an object at a given location. The angular velocity of the agent drives the translation of an activity packet in the head direction ring attractor network. Retrosplenial cortex (RSC) carries out the transformation from egocentric representations in the PW to allocentric representations in the MTL (driving BVCs and OVCs). The transformation circuit consists of 20 sublayers, each maximally modulated by a specific head direction while the remaining circuit is inhibited (Inh). In the medial temporal lobe network, perirhinal neurons (PRb/o) code for the identity of an object or extended boundary. PCs, BVCs and perirhinal neurons are reciprocally connected in an attractor network. Following encoding after object encounters, PCs are also reciprocally connected to OVCs and PRo neurons. 'Top-down' mode of operation: Activity in a subset of PCs, BVCs, and/or perirhinal neurons spreads to the rest of the MTL network (pattern completion) by virtue of intrinsic connectivity. With perceptual inputs to the PW disengaged (i.e. during recollection), the transformation circuit reconstructs parietal window (PWb/o) activity based on the current BVC and OVC activity. Updating PCs via entorhinal cortex (EC) GC inputs allows for a shift of viewpoint in imagery.
DOI: https://doi.org/10.7554/eLife.33752.007

## Quantification

To obtain a measure of successful recall or of novelty detection (i.e. mismatch between the perceived and remembered scenes), we correlate the population vectors of the model's neural populations between recall (the reconstruction in imagery) and encoding. These correlations are compared to correlations between recall and randomly sampled times as the agent navigates the environment in bottom-up mode. This measure of mismatch could potentially be compared to experimental measures of overlap between neuronal populations (e.g. *Guzowski et al., 1999*) in animals, or 'representational similarity' measures in fMRI, e.g. *Ritchey et al., 2013*).

## Simulations

In this section, we explore the capabilities of the BB-model in simulations and derive predictions for future research. Each simulation is accompanied by a Figure, a supplementary video visualizing the time course of activity patterns of neural populations, and a brief discussion. In Section Discussion, we offer a more general discussion of the model.

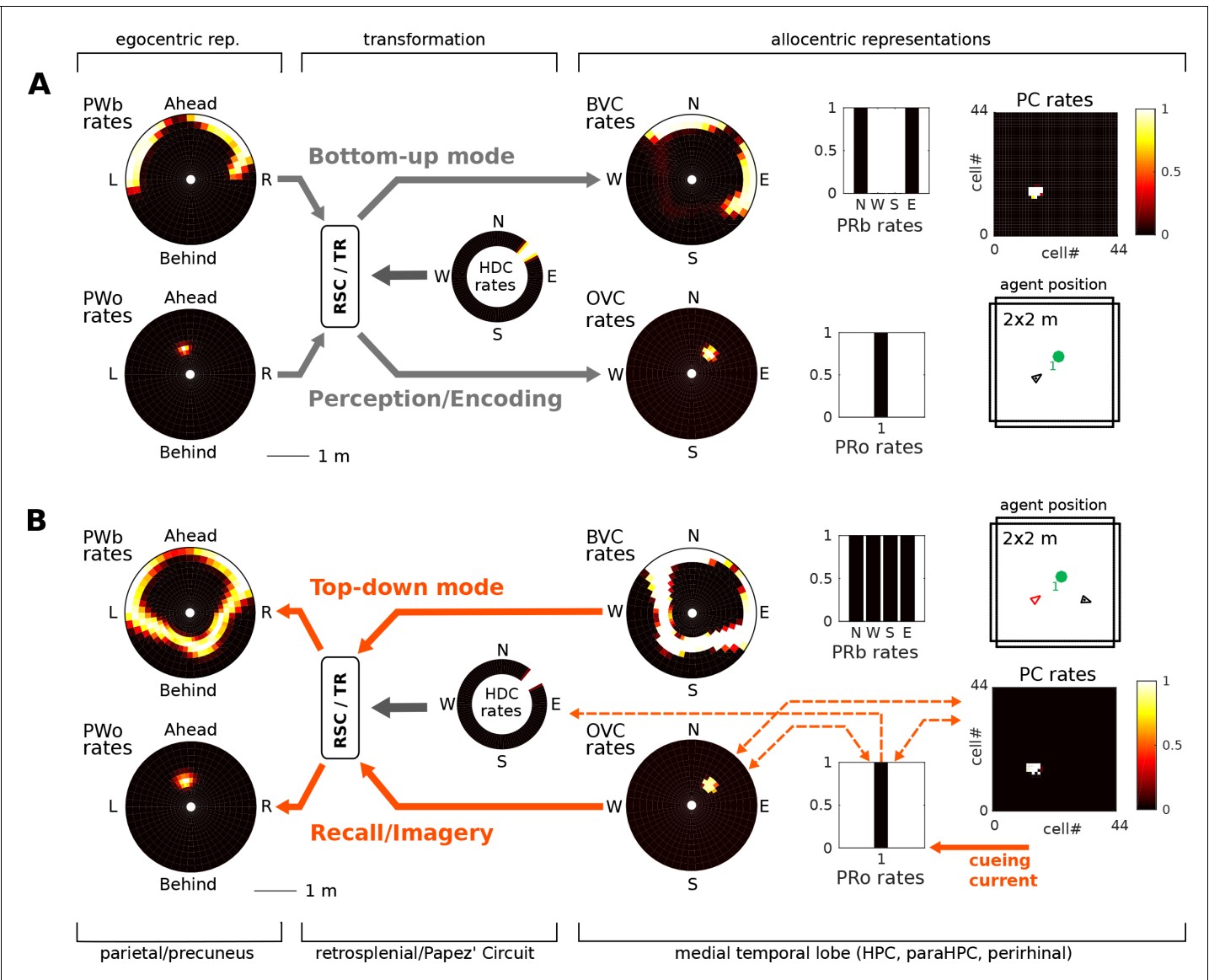

**Figure 5.** (A) Bottom-up mode of operation. Population snapshots at the moment of encoding during an encounter with a single object in a familiar spatial context. Left to right: PWb/o populations driven by sensory input project to the head-direction-modulated retrosplenial transformation circuit (RSC/TR, omitted for clarity, see *Video 1* and *Figure 2—figure supplement 1*); The transformation circuit projects its output to BVCs and OVCs; BVCs and PRb neurons constitute the main drive to PCs; perirhinal (PRb/o) neurons are driven externally, representing object recognition in the ventral visual stream. At the moment of encoding, reciprocal connections between PCs and OVCs, OVCs and PRo neurons, PCs and PRo neurons, and PRo neurons and current head direction are learned (see *Figure 3D*). Right-most panels show the agent in the environment and the PC population snapshot representing current allocentric agent position. (B) Top-down mode of operation, after the agent has moved away from the object (black triangle, right-most panel). Current is injected into a PRo neuron (bottom right of panel), modelling a cue to remember the encounter with that object. This drives PCs associated to the PRo neuron at encoding (dashed orange connections show all associations learned at encoding). The connection weights switch globally from bottom-up to top-down (connections previously at 5% of their maximum value now at 100% and vice versa; orange arrows). PCs become the main drive to OVCs, BVCs and PRb neurons. BVC and OVC representations are transformed to their parietal window counterparts, thus reconstructing parietal representations (PWb/PWo) similar to those at the time of encoding (compare left-most panels in A and B). That is, the agent has reconstructed a point of view embodied by parietal window activity corresponding to the location of encoding (red triangle, right-most panel). Heat maps show population firing rates frozen in time (black: zero firing; white: maximal firing).
DOI: https://doi.org/10.7554/eLife.33752.008

## Encoding of objects in spatial memory and recall (Simulation 1.0)

We let the agent model explore the square environment depicted in *Figure 3A*. However, the spatial context now contains an isolated object (*Figure 5*). During exploration, parietal window (PWb) neurons activate BVCs via the retrosplenial transformation circuit (RSC/TR), which in turn drive place cell activity. Similarly, when the object is present PWo neurons are activated, which drive OVCs via the transformation circuit. At the same time, object/boundary identity is signalled by perirhinal neurons (PRb/o). When the agent comes within a certain distance (here 55 cm) of an object, the following connection weights are changed to form Hebbian associations: PRo neurons are associated with PCs, HDCs, and OVCs; OVCs are associated with PCs and PRo neurons (also see *Figure 3D*); PCs are already connected to BVCs (in a familiar context). The weight change is calculated as the outer product of population vectors of the corresponding neuronal populations (yielding the Hebbian update), normalized, and added to the given weight matrix.

After the agent has finished its assigned trajectory, we test object-location memory via object-cued recall. That is, modeling some external trigger to remember a given object (e.g. 'Where did I leave my keys?'), current is injected into the PRo neuron coding for the identity of the object to-be-recalled. By virtue of learned connections, the PRo neuron drives the PCs which were active at encoding. Pattern completion in the MTL recovers the complete spatial context by driving activity in BVCs and PRb neurons. The connections from PRo neurons to head direction cells (*Figure 3D*) ensure a modulation of the transformation circuit such that allocentric BVC and OVC activity will be transformed to yield the parietal representation (i.e. a point of view) similar to the one at the time of encoding. That is, object-cued recall corresponds to a full reconstruction of the scene when the object was encoded. *Figure 5* depicts the encoding (*Figure 5A*) and recall phases (*Figure 5B*) of simulation 1.0. *Video 2* shows the entire trial. To facilitate matching simulation numbers and figures to videos, *Table 1* lists all simulations and relates them to their corresponding figures and videos.

Recollection in the BB-model results in visuo-spatial imagery of a coherent scene from a single viewpoint and direction, that is it implements a process of scene construction (*Burgess et al., 2001a*; *Byrne et al., 2007*; *Schacter et al., 2007*; *Hassabis et al., 2007*; *Buckner, 2010*) at the neuronal level. A mental image is re-constructed in the parietal window reminiscent of the perceptual activity present at encoding. Note that during imagery BVCs (and hence PWb neurons, *Figure 5B*) all around the agent are reactivated by place cells, because the environment is familiar (the agent having experienced multiple points of view at each location during the training phase). We do not simulate selective attention for boundaries (i.e. PWb neurons), although see *Byrne et al., 2007*.

Similar tasks in humans appear to engage the full network, including Papez' circuit, where head direction cells are found (for review see *Taube, 2007*); retrosplenial cortex (where we hypothesize the transformation circuit to be located) (*Burgess et al., 2001a*; *Lambrey et al., 2012*; *Auger and Maguire, 2013*; *Epstein and Vass, 2014*; *Marchette et al., 2014*; *Shine et al., 2016*); medial parietal areas (*Fletcher et al., 1996*; *Hebscher et al., 2018*); parahippocampus and hippocampus (*Hassabis et al., 2007*; *Addis et al., 2007*; *Schacter et al., 2007*; *Bird et al., 2010*); and possibly the entorhinal cortex (*Atance and O'Neill, 2001*; *Bellmund et al., 2016*; Horner et al. 2016; also see Simulation 4.0).

At the neuronal level, a key component of the BB-model are the object vector cells (OVCs) which code for the location of objects in peri-personal space. In Figure 5 the cells are organized according to the topology of their receptive fields in space, with the agent at the center (also compare to *Figure 2A2*). However, in rodent

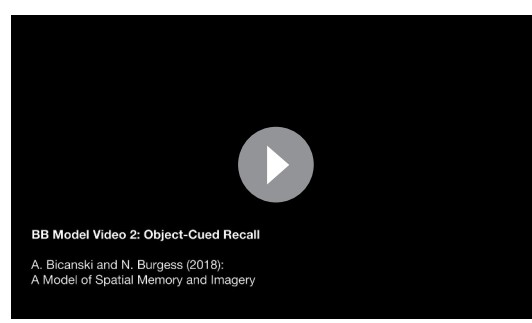

BB Model Video 2: Object-Cued Recall

A. Bicanski and N. Burgess (2018):
A Model of Spatial Memory and Imagery

**Video 2.** This video shows a visualization of the simulated neural activity as the agent moves in a familiar environment and encounters a novel object. The agent approaches the object and encodes it into long-term memory. Upon navigating past the object the agent initiates recall, reinstating patterns of neural activity similar to the patterns present during the original object encounter. Recall is identified with the re-construction of the original scene in visuo-spatial imagery (see main text). Please see caption of Video 1 for abbreviations.

DOI: https://doi.org/10.7554/eLife.33752.010

experiments individual spatially selective cells (like PCs or GCs) are normally visualized as time-integrated firing rate maps. We ran a separate simulation with three objects in the environment to examine firing rate maps of individual cells. OVCs show firing fields at a fixed allocentric distance and angle from objects (*Figure 6*). The BB-model predicts that OVC-like responses should be found as close as one synapse away from the hippocampus and were introduced as a parsimonious object code, analogous to BVCs and exploiting the existing transformation circuit. However, these rate maps show a striking resemblance to similar data from cells recently reported in the hippocampus of rodents (*Figure 6C*, compare to *Deshmukh and Knierim, 2013*). While *Deshmukh and Knierim (2013)* found these cells in the hippocampus, the object selectivity of these hippocampal neurons may have been inherited from other areas, such as lateral entorhinal cortex (*Tsao et al., 2013*), parahippocampal cortex (due to their similarities to BVCs) or medial entorhinal cortex (*Solstad et al., 2008*; *Hoydal et al., 2017*).

Anatomical connections between the potential loci of BVCs/OVCs and retrosplenial cortex (the suggested location of the egocentric-allocentric transformation circuit) exist. BVCs have been found in the subicular complex (*Lever et al., 2009*), and the related border cells and OVCs in medial entorhinal cortex (*Solstad et al., 2008*; *Hoydal et al., 2017*). Both areas receive projections from retrosplenial cortex (*Jones and Witter, 2007*), and project back to it (*Wyss and Van Groen, 1992*).

## Papez' circuit lesions induce amnesia (Simulations 1.1, 1.2)

*Figure 5* depicts the model performing encoding and object-cued recall. However, the model also allows simulation of some of the classic pathologies of long-term memory. Lesions along Papez' circuit have long been known to induce amnesia (*Delay and Brion, 1969*; *Squire and Slater, 1978*; *1989*; *Parker and Gaffan, 1997*; *Aggleton et al., 2016*). Thus, lesions to the fornix and mammillary bodies severely impact recollection, although recognition can be less affected (*Tsivilis et al., 2008*). In the context of spatial representations, Papez' circuit is notable for containing head direction cells (as well as many other cell types not in the model). That is, the mammillary bodies (more specifically the lateral mammillary nucleus, LMN), anterior dorsal thalamus, retrosplenial cortex, parts of the subicular complex and medial entorhinal cortex all contain head direction cells (*Taube, 2007*; *Sargolini et al., 2006*). Thus, lesioning Papez' circuit removes (at least) the head direction signal from our model, and is modeled by setting the input from head direction cells to the retrosplenial transformation circuit (RSC/TR) to zero.

In the bottom-up mode of operation (perception), the lesion removes drive to the transformation circuit and consequently to the boundary vector cells and object vector cells. That is, the perceived location of an object (present in the egocentric parietal representation) cannot elicit activity in the MTL and thus cannot be encoded into memory (*Figure 7*). Some residual MTL activity reflects input from perirhinal neurons representing the identity of perceived familiar boundaries (i.e. recognition mediated by perirhinal cells is spared). In the top-down mode of operation (recall) there are two effects: (i) Since no new elements can be encoded into memory, post-lesion events cannot be recalled (anterograde amnesia; Simulation 1.1, *Figure 7A*, Video 5); and (ii) For pre-existing memories (e.g. of an object encountered prior to the lesion), place cells (and thus the remaining MTL populations) can be driven via learned connections from perirhinal neurons (e.g. when cued with the object identity; Simulation 1.2, *Figure 7B*, Video 6), but no meaningful egocentric representation can be instantiated in parietal areas, preventing episodic recollection/imagery. Equating the absence of parietal activity with the inability to recollect is strongly suggested by the fact that visuo-spatial imagery in humans relies on access to an egocentric representation (as in hemispatial representational neglect; *Bisiach and Luzzatti, 1978*). Simulations 1.1 and 1.2 show that the egocentric neural correlates of objects and boundaries present in the visual field persist in the parietal window only while the agent perceives them (they could also be held in working memory, which is not modelled here). Note that perirhinal cells and upstream ventral visual stream inputs are spared, so that an agent could still report the identity of the object.

## Quantification, robustness to noise and neuron loss

*Figure 8A* shows correlations between population vectors of neural patterns during imagery/recall and those during encoding for Simulation 1.0 (Object-cued recall; *Figure 5*). OVCs and PCs exhibit correlation values close to one, indicating faithful reproduction of patterns. BVC correlations are

**Table 1.** List of simulations, their content, corresponding Figures and videos

| Simulation no. | Content | Related figures | Video no. |
|---|---|---|---|
| 0 | Activity in the transformation circuit | *Figure 2—figure supplement 1* | 1 |
| 1.0 | Object-cued recall | *Figures 5* and *6*,8A | 2 |
| 1.0n1 | Object-cued recall with neuron loss | *Figure 8B* | 3 |
| 1.0n2 | Object-cued recall with firing rate noise | *Figure 8C* | 4 |
| 1.1 | Papez' circuit Lesion (anterograde amnesia) | *Figure 7A* | 5 |
| 1.2 | Papez' circuit Lesion (retrograde amnesia) | *Figure 7B* | 6 |
| 1.3 | Object novelty (intact hippocampus) | *Figure 9A* | 7 |
| 1.4 | Object novelty (lesioned hippocampus) | *Figure 9B* | 8 |
| 2.1 | Boundary trace responses | *Figure 10A,B,C* | 9 |
| 2.2 | Object trace responses | *Figure 10D* | 10 |
| 3.0 | Inspection of scene elements in imagery | *Figure 11* | 11 |
| 4.0 | Mental Navigation | *Figure 12* | 12 |
| 5.0 | Planning and short-cutting | *Figure 13* | 13 |

DOI: https://doi.org/10.7554/eLife.33752.009

somewhat diminished because recall reactivates all boundaries fully, compared to a field of view of 180 degrees during perception with limited reactivation of cells representing boundaries outside the field of view. PW neurons show correlations below one because at recall reinstatement in parietal areas requires the egocentric-allocentric transformation (i.e. OVC signals passed through retrosplenial cells), which blurs the pattern compared to perceptual instatement in the parietal window (i.e. imagined representations are not as precise as those generated by perception).

To test the model's robustness with regard to firing rate noise and neuron loss, we perform two sets of simulations (modifications of Simulation 1.0, object-cued recall). In the first set we randomly chose cells in equal proportions in all model areas (except HDCs) to be permanently deactivated and assess recall into visuo-spatial imagery. Up to 20% of the place cells, grid cells, OVCs, BVCs,

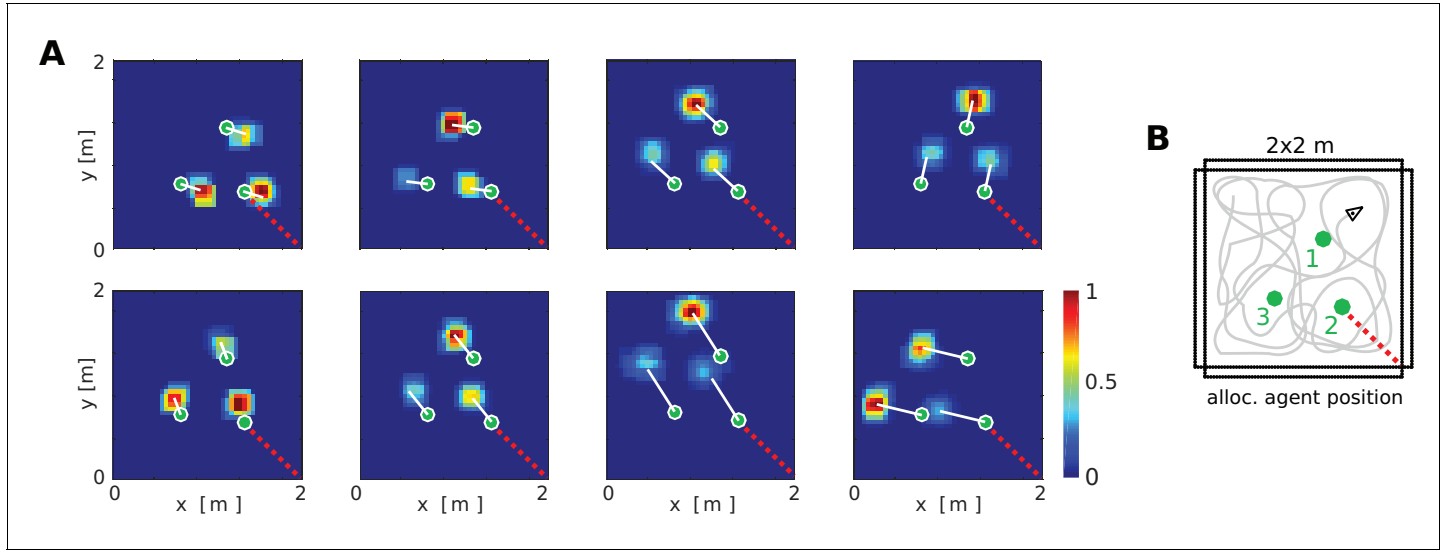

**Figure 6.** Firing fields of object vector cells. (**A**) Firing rate maps for representative object vector cells (OVCs), firing for objects with a fixed allocentric location and direction relative to the agent. Object locations superimposed as green circles. Note that the objects have different identities, which would be captured by perirhinal neurons, not OVCs. Compare to *Figure 4* in *Deshmukh and Knierim, 2013*. White lines point from objects to firing fields. Red dotted line added for comparison with B. (**B**) Distribution of the objects in the arena and an illustration of a possible agent trajectory.

DOI: https://doi.org/10.7554/eLife.33752.011

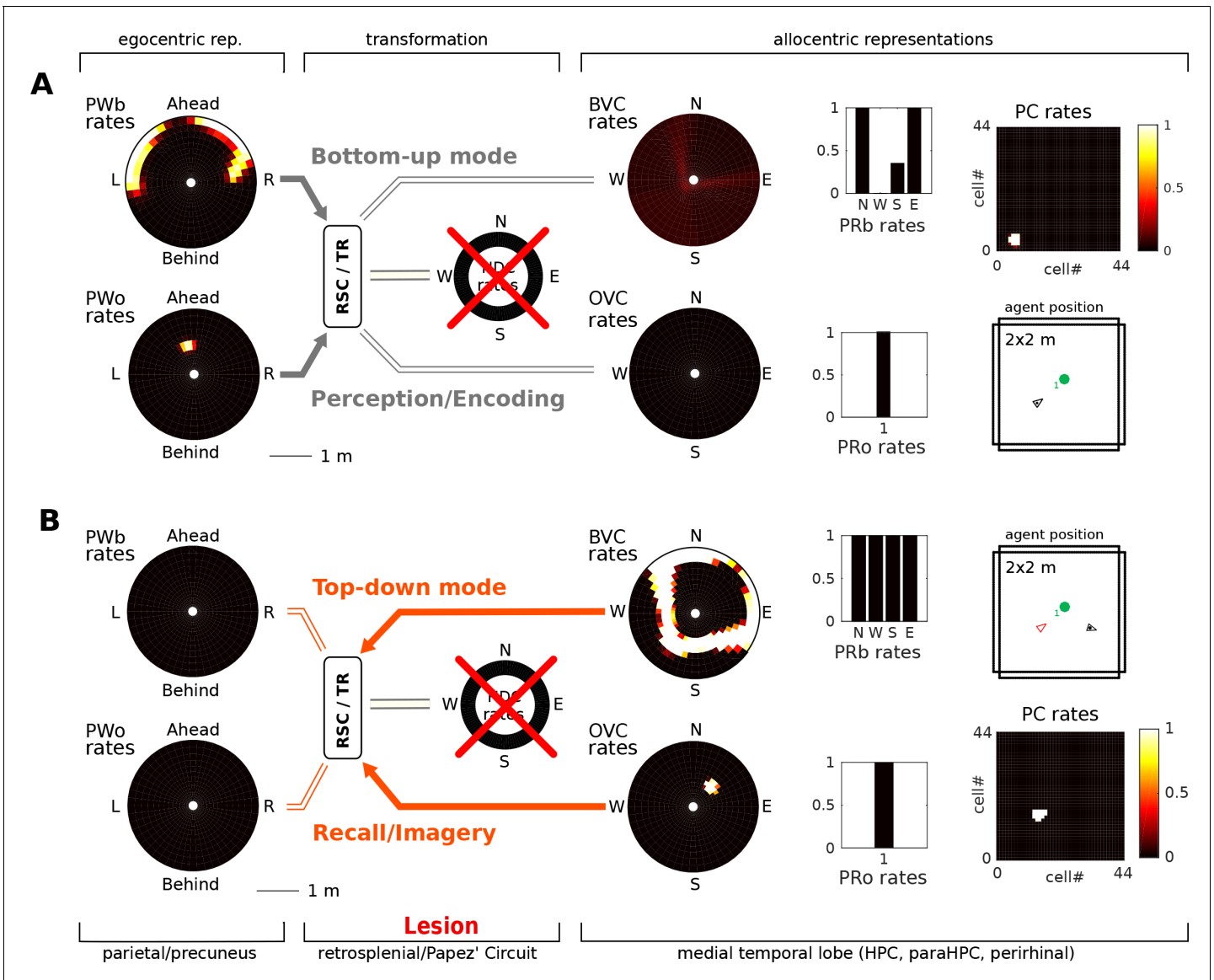

**Figure 7.** Papez' circuit lesions. (**A**) In the bottom-up mode of operation (perception), a lesion to the head direction circuit removes drive to the transformation circuit and consequently to the boundary vector cells (BVCs) and object vector cells (OVCs). A perceived object (present in the egocentric parietal representation, PWo) cannot elicit activity in the MTL and thus cannot be encoded into long-term memory, causing anterograde amnesia. Place cells fire at random locations, driven by perirhinal neurons. (**B**) For memories of an object encountered before the lesion, place cells can be cued by perirhinal neurons, and pattern completion recruits associated OVC, BVCs and perirhinal neurons, but no meaningful representation can be instantiated in parietal areas, preventing episodic recollection/imagery (retrograde amnesia for hippocampus-dependent memories).
DOI: https://doi.org/10.7554/eLife.33752.012

parietal and retrosplenial neurons were deactivated. Head direction cells were excluded because of the very low number simulated (see below). Although we do not attempt to model any specific neurological condition, this type of simulation could serve as a starting point for models of diffuse damage, as might occur in anoxia, Alzheimer's disease or aging. The average correlations between the population vectors at encoding versus recall are shown in *Figure 8B*.

The ability to maintain a stable attractor state among place cells and head direction cells is critical to the functioning of the model, while damage in the remaining (feed-forward) model components manifests in gradual degradation in the ability to represent the locations of objects and boundaries (see accompanying *Video 3*). For example, if certain parts of the parietal window suffer from neuron loss, the reconstruction in imagery is impaired only at the locations in peri-personal space encoded

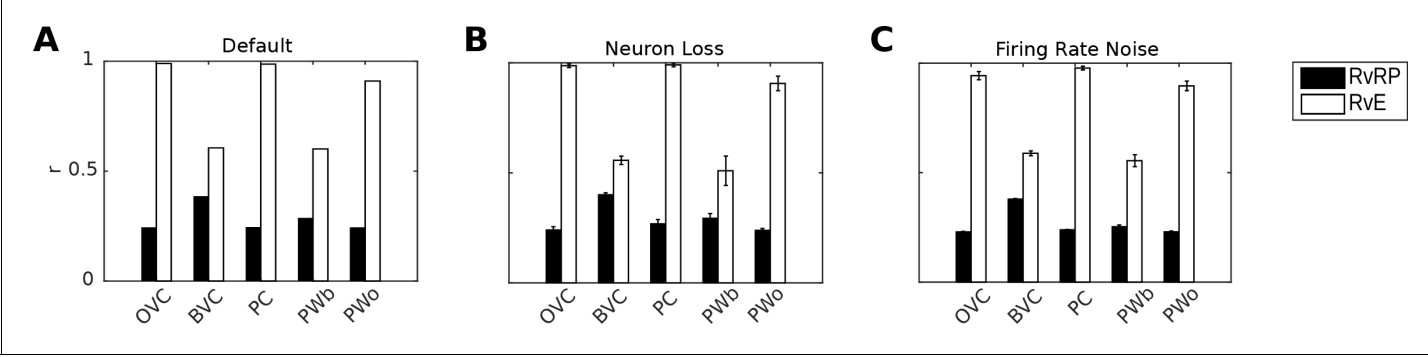

**Figure 8.** Correlation of neural population vectors between recall/imagery and encoding. (**A**) In the intact model, OVCs and place cells exhibit correlation values close to one, indicating faithful reproduction of patterns. (**B**) Random neuron loss (20% of cells in all populations except for the head direction ring). (**C**) The effect of firing rate noise. Noise is also applied to all 20 retrosplenial transformation circuit sublayers (as is neuron loss; correlations not shown for clarity). Firing rate noise is implemented as excursions from the momentary firing rate as determined by the regular inputs to a given cell (up to peak firing rate). The amplitudes of perturbations are normally distributed (mean 20%, standard deviation 5%) and applied multiplicatively at each time step). White bars show the correlation between the neural patterns at encoding vs recall (RvE), while black bars show the average correlation between the neural patterns at recall vs pattern sampled at random times/locations (here every 100 ms; RvRP). Each bar is averaged over 20 separate instances of the same simulation (with newly drawn random numbers). Error bars indicate standard deviation across simulations.
DOI: https://doi.org/10.7554/eLife.33752.013

by the missing neurons (indeed, this can model representational neglect; *Byrne et al., 2007*), see also *Pouget and Sejnowski, 1997*). The place cell population was more robust to silencing than the head-direction population (containing only 100 neurons), simply because greater numbers of neurons were simulated, giving greater redundancy. As long as a stable attractor state is present, the model can still encode and recall meaningful representations, giving highly correlated perceived and recalled patterns (*Figure 8B*).

The model is also robust to adding firing rate noise (up to 20% of peak firing rate) to all cells. Correlations between patterns at encoding and recall remain similar to the noise-free case, see *Figure 8C*. *Videos 3* and *4* show an instance from the neuron-loss and firing rate noise simulations respectively.

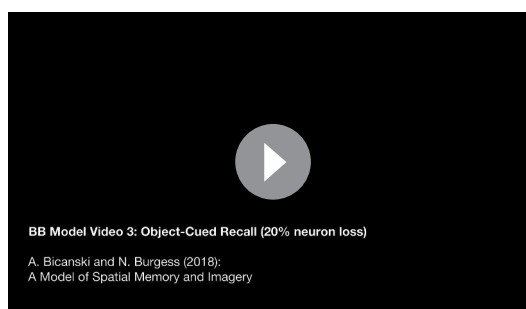

**Video 3.** This video shows the same scenario as *Video 2* (object-cued recall), however, with 20% randomly chosen lesioned cells per area. The agent moves in a familiar environment and encounters a novel object. The agent approaches the object and encodes it into long-term memory. Upon navigating past the object, the agent initiates recall, reinstating patterns of neural activity similar to the patterns present during the original object encounter. Please see caption of Video 1 for abbreviations.
DOI: https://doi.org/10.7554/eLife.33752.014

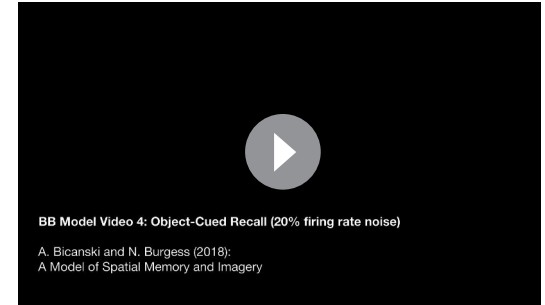

**Video 4.** This video shows the same scenario as *Video 2* (object-cued recall), however, with firing rate noise applied to all neurons (max. 20% of peak rate). The agent moves in a familiar environment and encounters a novel object. The agent approaches the object and encodes it into long-term memory. Upon navigating past the object the agent initiates recall, reinstating patterns of neural activity similar to the patterns present during the original object encounter. Please see caption of Video 1 for abbreviations.
DOI: https://doi.org/10.7554/eLife.33752.015

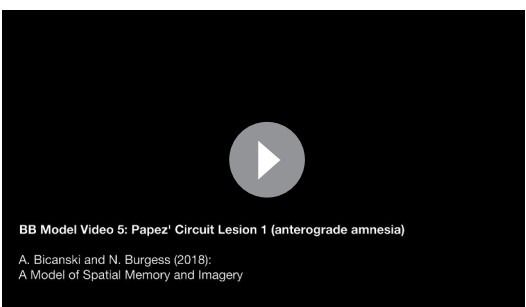

BB Model Video 5: Papez' Circuit Lesion 1 (anterograde amnesia)

A. Bicanski and N. Burgess (2018):
A Model of Spatial Memory and Imagery

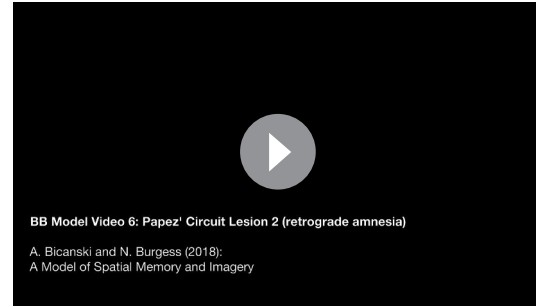

BB Model Video 6: Papez' Circuit Lesion 2 (retrograde amnesia)

A. Bicanski and N. Burgess (2018):
A Model of Spatial Memory and Imagery

**Video 5.** This video shows a visualization of the simulated neural activity as the agent encounters an object and subsequently tries to engage recall similar to Simulation 1.0 (*Video 2*). However, a lesion to the head direction system (head direction cells are found along Papez' circuit) precludes the agent from laying down new memories, because the transformation circuit cannot drive the medial temporal lobe. That is the transformation circuit cannot instantiate OVC/BVC representations derived from sensory input for subsequent encoding, leading to anterograde amnesia in the model agent (see main text). Please see caption of Video 1 for abbreviations.

DOI: https://doi.org/10.7554/eLife.33752.016

## Novelty detection (Simulations 1.3, 1.4)

In the model, hippocampal place cells bind all scene elements together. The locations of these scene elements relative to the agent are encoded in the firing of boundary vector cells (BVCs) and object vector cells (OVCs). Rats show a spontaneous preference for exploring novel/altered stimuli compared to familiar/unchanged ones. We simulate one of these experiments (*Mumby et al., 2002*), in which rats preferentially explore one of two objects that has been shifted to a new location within a given environment, a behavior impaired by hippocampal lesions. In Simulations 1.3 and 1.4, the agent experiences an environment containing two objects, one of which is later moved. We define a mismatch signal as the difference in firing of object vector cells during encoding versus recall (modelled as imagery, at the encoding location), and assume that the relative amounts of exploration would be proportional to the mismatch signal.

**Video 6.** This video shows a visualization of the simulated neural activity as the agent moves through an empty environment and tries to engage recall of a previously present object. A lesion to the head direction system (head direction cells are found along Papez' circuit) has been implemented similar to Simulation 1.1 (*Video 5*). The agent is supplied with the connection weights learned in Simulation 1.0 (*Video 2*), where it has successfully memorized a scene with an object. That is, the agent has acquired a memory before the lesion. However, even though cueing with the object re-activates the correct medial temporal representations, due to the lesion no reinstatement in the parietal window cannot occur, leading to retrograde amnesia for hippocampus-dependent memories in the model agent. Note, it is hypothesized that a cognitive agent only has conscious access to the egocentric parietal representation, as suggested by hemispatial representational negelct (*Bisiach and Luzzatti, 1978*) (see main text). Please see caption of Video 1 for abbreviations.

DOI: https://doi.org/10.7554/eLife.33752.017

With an intact hippocampus (*Figure 9*; *Video 7*), the moved object generates a significant novelty signal, due to the mismatch between recalled (top-down) OVC firing and perceptual (bottom-up) OVC firing at the encoding location. That detection of a change in position requires the hippocampus is consistent with place cells binding the relative location of an object (via object vector cells) to perirhinal neurons signalling the identity of an object.

Hippocampal lesions are implemented by setting the firing rates of hippocampal neurons to zero. A hippocampal lesion (*Figure 9*; *Video 8*) precludes the generation of a meaningful novelty signal because the agent is incapable of generating a coherent point of view for recollection, and the appropriate BVC configuration cannot be activated by the now missing hippocampal input. Connections between object vector cells and perirhinal neurons (see *Figure 3D*) can still form during encoding in the lesioned agent. Thus some OVC activity is present during recall due to these connections. However, this activity is not location specific. Without the reference frame of place cells and thence BVC activity this residual OVC activity during recall can be generated anywhere (see *Figure 9F–H*). It only tells the agent that it has seen this object at a given distance and direction, but not where in the environment it was seen. Hence, the mismatch signal is equal for both objects, and exploration

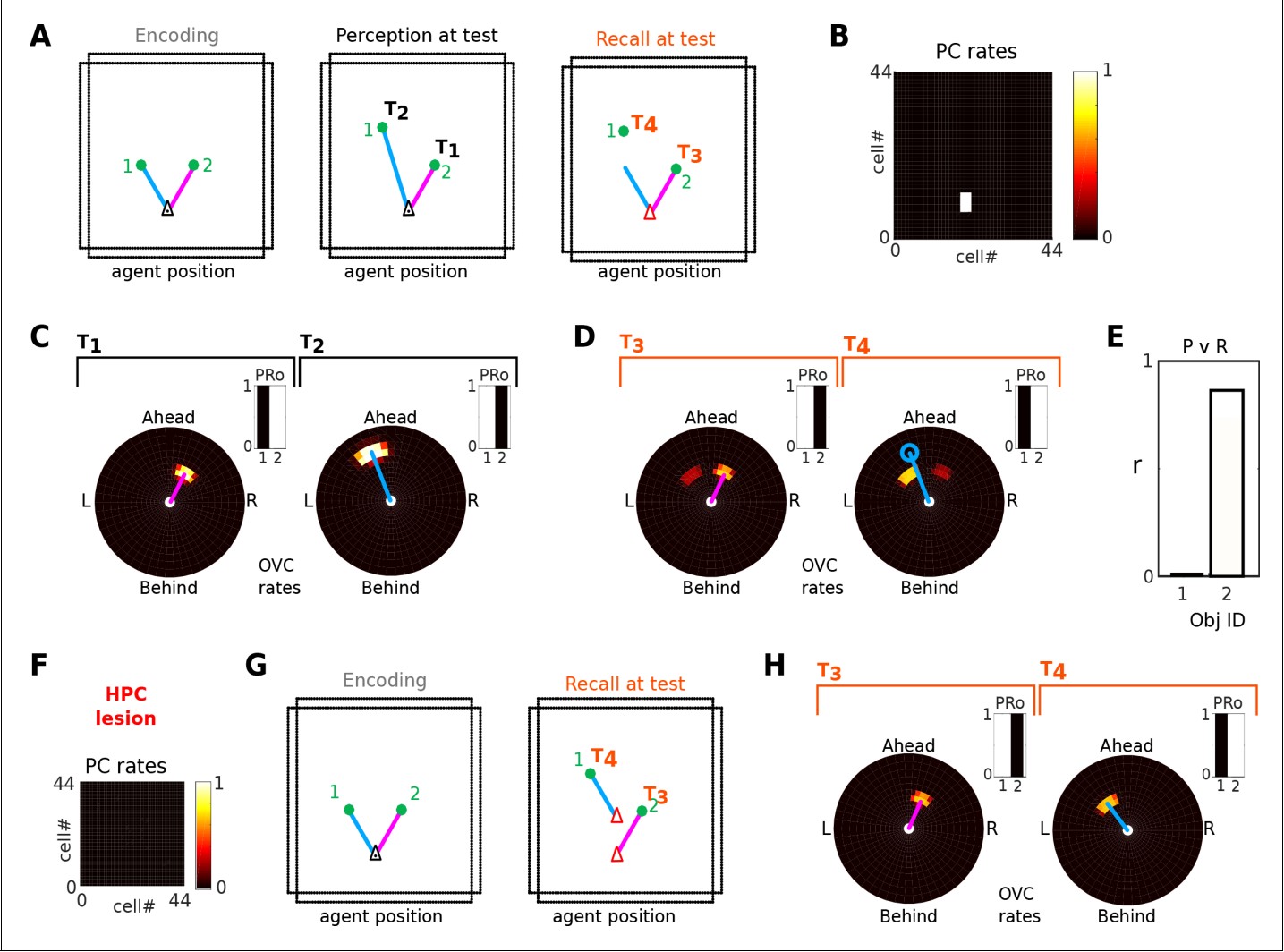

**Figure 9.** Detection of moved objects via OVC firing mismatch. (A) Two objects are encoded from a given location (left). After encoding, object one is moved further North. When the agent returns to the encoding location, the perceived position of object one differs from that at encoding (blue line, middle panel). When the agent initiates recall (right) the perceived location of object 1 (green filled circle) and its imagined location (end point of blue line) differ. (B) PC activity is the same in all three circumstances, that is PC activity alone is insufficient to tell which object has moved. (C-D) The perceived location as represented by OVCs during perception (C; objects 1 and 2 sampled sequentially at times T1, T2) and during recall (D; objects 1 and 2 sampled sequentially at times T3, T4). Blue circle in panel D indicates the previously perceived position of object 1. Inset bar graphs show the concurrent activity of perirhinal cells (PRo). (E) The mismatch in OVC firing results in near zero correlation between encoding and recall patterns for object 1 (black bar), while object 2 (white bar) exhibits a strong correlation, so that object one would be preferentially explored. Note, the correlation for object two is less than 1 because of the residual OVC activity of the other object (secondary peaks in both panels in D, driven by learned PC-to-OVC connections). (F) A hippocampal lesion removes PC population activity, so that OVC activity is not anchored to the agent's location at encoding. (G-H) An incidental match between learned and recalled OVC patterns can occur for either object at specific locations (red arrow heads in second panel in G), but otherwise mismatch is signaled for both objects equally and neither object receives preferential exploration.
DOI: https://doi.org/10.7554/eLife.33752.018

time would be split roughly evenly between them. However, if the agent happens to be at the same distance and direction from the objects as at encoding, then perceptual OVC activity will match the recalled OVC activity (*Figure 9G,H*), which might correspond to the ability of focal hippocampal amnesics to detect the familiarity of an arrangement of objects if tested from the same viewpoint as encoding (*King et al., 2002*; but see also *Shrager et al., 2007*).

Rats also show preferential exploration of a familiar object that was previously experienced in a different environment, compared with one previously experienced in the same environment, and this

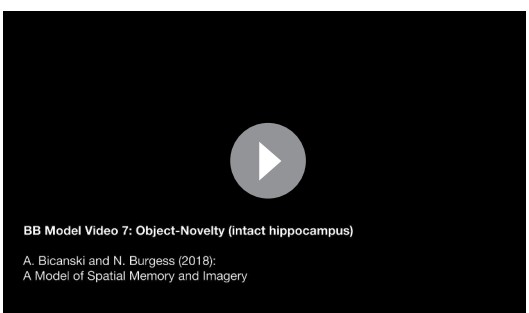

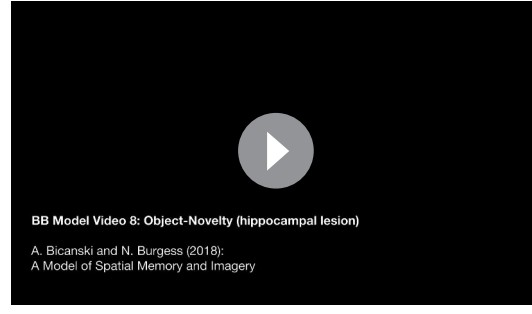

**Video 7.** This video shows a visualization of the simulated neural activity in a reproduction of the object novelty paradigm of *Mumby et al., 2002*; detecting that one of two objects has been moved). The agent is faced with two objects and encodes them (sequentially) into memory. Following some behavior one of the two objects is moved. Note, in real experiments the animal is removed for this manipulation. In simulation, this is unnecessary. Once the agent has returned to location of encoding, it is faced with the manipulated object array. The agent then initiates recall for objects one and two in sequence. The patterns of OVC re-activation can be compared to the corresponding patterns during perception (population vectors correlated, see main text). For the moved object, the comparison signals a change (near zero correlation). That object would hence be preferentially explored by the agent, and the next movement target for the agent is set accordingly (see main text). Please see caption of Video 1 for abbreviations.

DOI: https://doi.org/10.7554/eLife.33752.019

**Video 8.** should be compared to *Video 7*. It shows a reproduction of the object novelty paradigm of *Mumby et al., 2002*; detecting that one of two objects has been moved). The agent is faced with two objects and encodes an association between relative object location (signaled by OVCs) and object identity (signaled by perirhinal neurons) - see *Video 7* for encoding phase. Due to the hippocampal lesion, these associations cannot be bound to place cells. Once one of the two objects is moved (compare to Simulation 1.3) the agent initiates recall and the patterns of OVC re-activation are compared to the corresponding patterns during perception (population vectors correlated, see main text). Recall is initiated at two distinct locations to highlight the following effect of the lesion: Since associations between OVCs and perirhinal neurons are not bound to a specific environmental location a comparison of OVC patterns between perception and recall signals mismatch everywhere for both objects except for the two special locations at which imagery is engaged in the video. At each of those locations, the neural pattern due to the learned association happens to coincide with the pattern during perception for one of the two objects. Hence no object can be singled out for enhanced exploration. Match and Mismatch is signaled equally for both objects (see main text). Please see caption of Video 1 for abbreviations.

DOI: https://doi.org/10.7554/eLife.33752.020

preference is also abolished by hippocampal lesions (*Mumby et al., 2002*; *Eacott and Norman, 2004*; *Langston and Wood, 2010*). We have not simulated different environments (using separate place cell ensembles), but note that 'remapping' of PCs between distinct environments (i.e. much reduced overlap of PC population activity; e.g. *Bostock et al., 1991*; *Anderson and Jeffery, 2003*; *Wills et al., 2005*) suggests a mismatch signal for the changed-context object would be present in PC population vectors. Initiating recall of object A, belonging to context 1, in context 2, would re-activate the PC ensemble belonging to context 1, creating an imagined scene from context one which would mismatch the activity of PCs representing context two during perception. A hippocampal lesion precludes such a mismatch signal by removing PCs.

Finally, it has been argued that object recognition (irrespective of context) is spared after hippocampal but not perirhinal lesions (*Aggleton and Brown, 1999*; *Winters et al., 2004*; *Norman and Eacott, 2004*) which would be compatible with the model given that its perirhinal neuronal population signals an object's identity irrespective of location.

## 'Top-down' activity and trace responses (Simulations 2.1, 2.2)

Simulations 1.3 and 1.4 dealt with a moved object. Similarly, if a scene element (a boundary or an object) has been removed after encoding, probing the memorized MTL representation can reveal trace activity reflecting the previously encoded and now absent boundary or object.

Section (Bottom-up vs top-down modes of operation) summarizes how top-down and bottom-up phases are implemented by a modulation of connection strengths (see *Figures 1* and

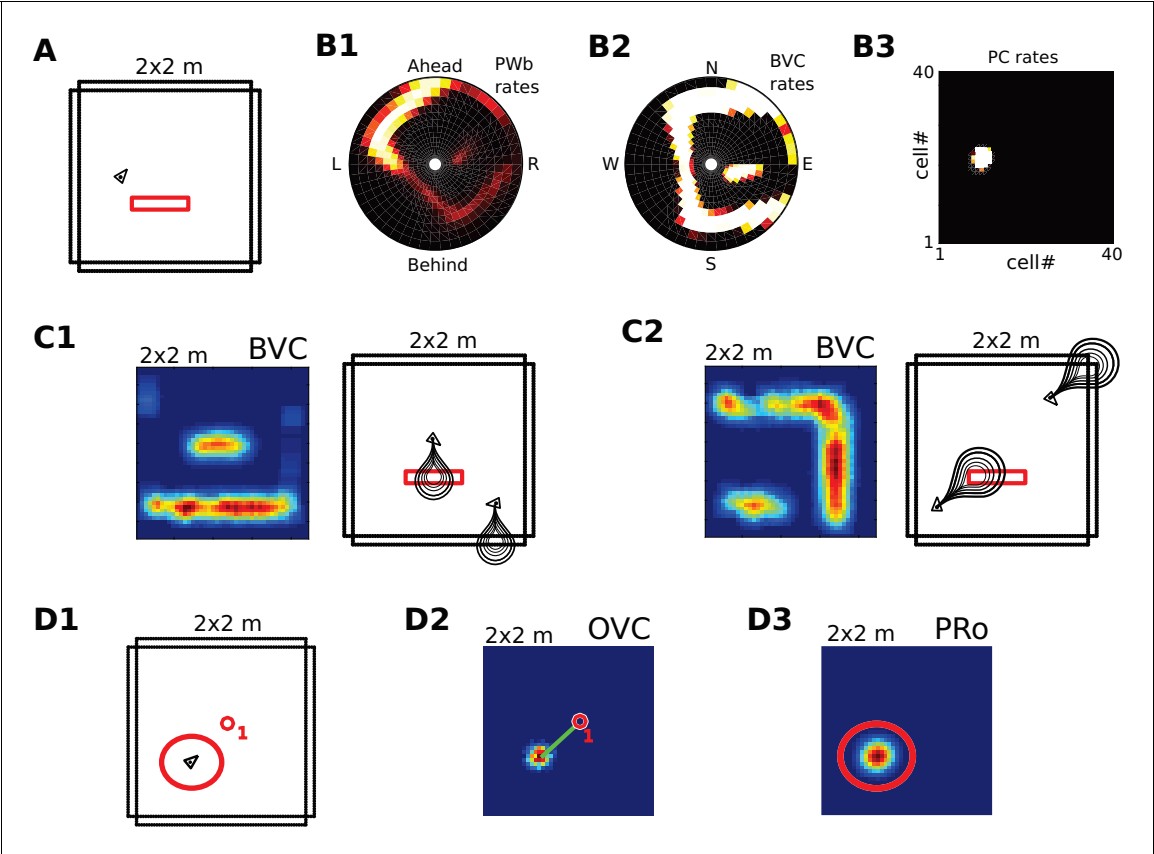

**Figure 10.** 'Top-down' activity and 'trace' responses. (A) An environment containing a small barrier (red outline) has been encoded in the connection weights in the MTL, but the barrier has been removed before the agent explores the environment again. (B) Activity snapshots for PWb (B1), BVC (B2) and PC (B3) populations during exploration. The now absent barrier is weakly represented in parietal window activity due to the periodic modulation of top-down connectivity during perception, although 'bottom-up' sensory input due to visible boundaries still dominates (see main text). (C1) High gain for top-down connections yields BVC firing rate maps with trace fields due to the missing boundary. Left: BVC firing rate map. Right: An illustration of the BVC receptive field (teardrop shape attached to the agent at a fixed allocentric direction and distance) with the agent shown at two locations where the cell in the left panel fires maximally. (C2) Same as C1 for a cell whose receptive field is tuned to a different allocentric direction. (D1) Similarly to the missing boundary in A, a missing object (small red circle) can produce 'trace' firing in an OVC (D2). Every time the agent traverses the location from which the object was encoded (large red circle in D1), learned PC-to-OVC connections periodically reactivate the associated OVC. (D3) The same PCs also re-activate the associated perirhinal identity cell (PRo), yielding a spatial trace firing field for a nominally non-spatial perirhinal cell (red circle).
DOI: https://doi.org/10.7554/eLife.33752.021

*4*, Materials and methods section Embedding Object-representations into a Spatial Context: Attention and Encoding, and Appendix). During perception, the 'top-down' connections from the MTL to the transformation circuit and thence to the parietal window are reduced to 5% of their maximum strength, to ensure that learned connections do not interfere with on-going, perceptually driven activity. During imagery, the 'bottom-up' connections from the parietal window to the transformation circuit and thence to the MTL are reduced to 5 percent of their maximum strength.

In rodents, it has been proposed that encoding and retrieval are gated by the theta rhythm (*Hasselmo et al., 2002*): a constantly present modulation of the local field potential during exploration. In humans, theta is restricted to shorter bursts, but is associated with encoding and retrieval (*Düzel et al., 2010*). If rodent theta determines the flow of information (encoding vs retrieval) then it may be viewed as a periodic comparison between memorized and perceived representations, without deliberate recall of a specific item in its context (that is, without changing the point of view). In Simulations 2.1 and 2.2, we implement this scenario. There is no cue to recall anything specific, regular sensory inputs are continuously engaged, and we periodically switch between bottom-up and top-down modes (at roughly theta frequency) to allow for an on-going comparison between perception and recall. Activity due to the modulation of top-down connectivity during perception

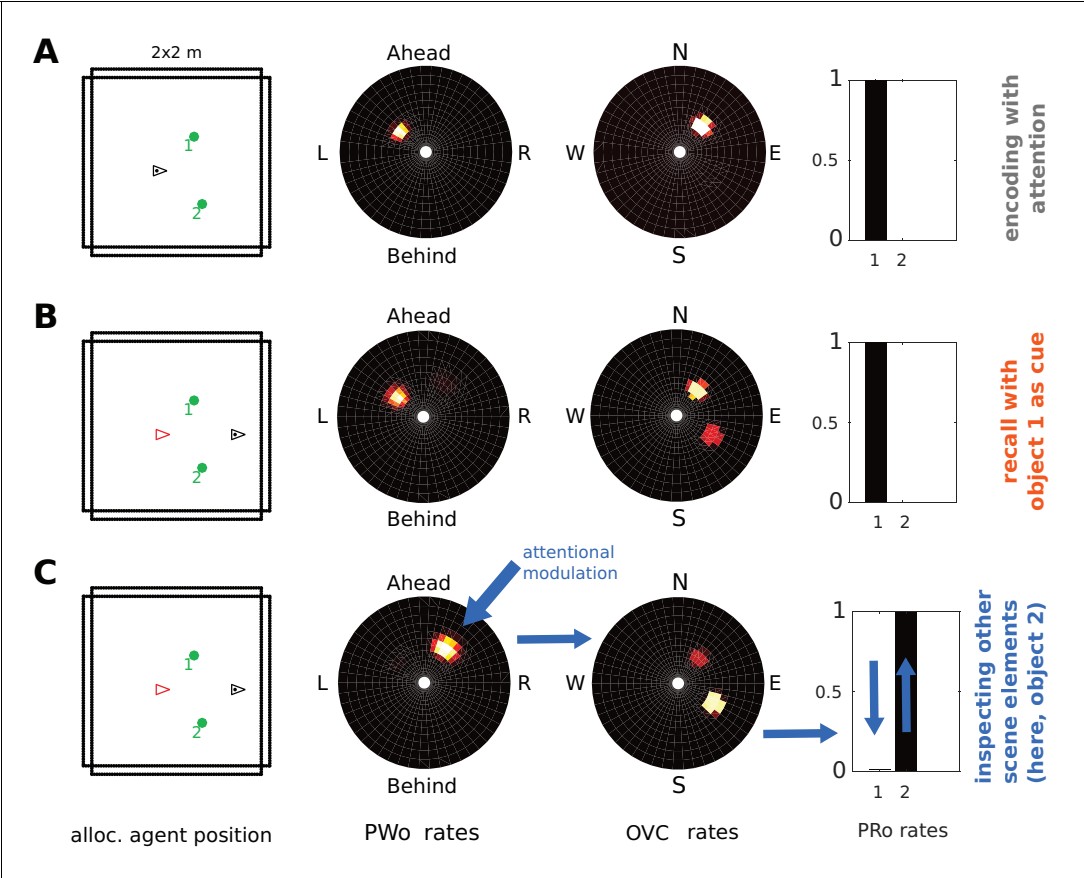

**Figure 11.** Inspecting scene elements in imagery. The agent encounters two objects. (**A**) Activity in PWo (left) and OVCs (right) populations when the agent is attending to one of the two objects during encoding. Both objects are encoded sequentially from the same location (time index 0.22 in *Video 11*). The agent then moves past the objects. (**B**) Imagery is engaged by querying for object 1, raising activity in corresponding PRo neurons (far right) and switching into top-down mode (similar to Simulation 1.0, *Figure 5* and *Video 2*), leading to full imagery from the point of view at encoding. Residual activity in the OVC population at the location of object 2 (encoded from the same position, that is driven by the same place cells) translates to weak residual activity in the PWo population. (**C**) Applying additional current (i.e. allocating attention) to the PWo cells showing residual activity at the location of object 2 (leftmost blue arrow) and removing the drive to the PRo neuron corresponding to object 1 (because the initial query has been resolved) leads to a build-up of activity at the location of object two in the OVC population (blue arrow between PWo and OVC plots). By virtue of the OVC to PRo connections (blue between OVC and Pro plots), the PRo neuron for object two is driven (and inhibits PRo neuron 1, right-most blue arrows). Thus, the agent has inferred the identity of object 2, after having initiated imagery to visualize object 1, by paying attention to its egocentric location in imagery.

DOI: https://doi.org/10.7554/eLife.33752.024

propagates to the parietal window representations (PWb/o), allowing for a detection of mismatch between sensorily driven and imagery representations.

In Simulation 2.1, the agent has a set of MTL weights which encode the contextual representation of a square room with an inserted barrier (i.e. a barrier was present in the training phase). However, when the agent explores the environment, the barrier is absent (*Figure 10A*). Due to the modulation of top-down connectivity, the memory of the barrier (in form of BVC activity) periodically bleeds into the parietal representation during perception (*Figures 10B1*, *2* and *3* and *Video 9*). The resultant dynamics carry useful information. First, letting the memory representation bleed into the perceptual one allows an agent, in principle, to localize and attend to a region of space in the egocentric frame of reference (as indicated by parietal window activity) where a change has occurred. A mismatch between the perceived (low bottom-up gain) and partially reconstructed (high bottom-up gain) representations, can signal novelty (compare to Simulations 1.3, 1.4), and could underlie the production of memory-guided attention (e.g. *Moores et al., 2003*; *Summerfield et al., 2006*). Moreover, the theta-like periodic modulation of top-down connectivity causes the appearance of 'trace' responses

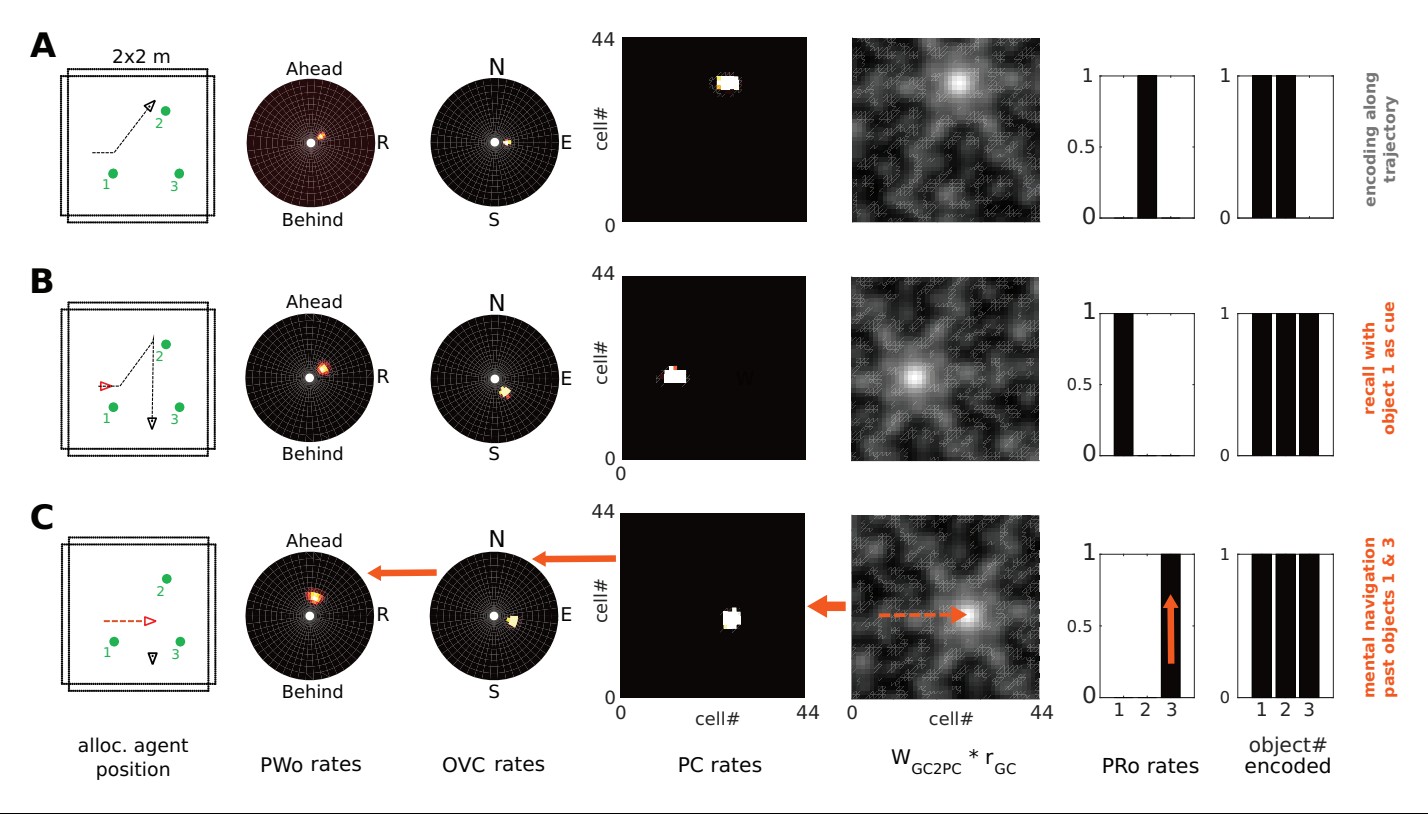

**Figure 12.** Mental navigation with grid cells. Left to right: allocentric agent position (black triangle) and recent trajectory (black dashed line); PWo, OVC, and PC population snapshots; GC input to PCs (i.e. GC firing rates multiplied by connection weights from GCs to PCs); PRo neurons. The rightmost panel indicates which objects have been encoded. (**A**) The agent is exploring the environment and has just encoded the second object into memory (right-most bar chart). Object one has been encoded near the start of the trajectory. (**B**) After encoding the third object and moving past it, the agent initiates imagery, recalling object one in its spatial context (top-down mode) from a point of view West of object 1, facing East (red triangle). (**C**) Mock motor efference shifts GC activity (dashed arrow on GC input to PCs) and thence drives the PC activity bump representing (imagined) agent location. The allocentric (BVCs) and egocentric (PWb) boundary representation follow suit (see main text and *Video 12*). As the PC activity bump passes the location at which object 3 was encoded, corresponding OVC activity is elicited by learned connections (and is transformed into PWo activity (solid orange arrows indicate GCs updating PCs, PCs updating OVCs, etc). NB object 3 appears in the reconstructed scene ahead-right of the agent (PWo snapshot, second panel), despite being encoded ahead-left of the agent when it moved southwards from object 2 toward object 3. The corresponding perirhinal neuron is also driven to fire by PCs (orange arrow in PRo panel).

DOI: https://doi.org/10.7554/eLife.33752.026

in BVC firing rate maps, indicating the location of previously encoded, now absent, boundary elements (*Figures 10C1* and *2*)

Simulation 2.2 (*Figures 10D1,D2* and *Video 10*) shows similar' trace' responses for OVCs. The agent has a set of MTL weights which encode the scene from Simulation 1.0 (*Figure 5*) where it encountered and encoded an object. The object is now absent (small red circle in *Figure 10D1*), but the periodic modulation of top-down connectivity reactivates corresponding OVCs, yielding trace fields in firing rate maps. This activity can bleed into the parietal representation during perception (e.g. at simulation time 9:40-10:00 in *Video 6*), albeit only when the location of encoding is crossed by the agent, and with weaker intensity than missing boundary activity (the smaller extent of the OVC representation leads to more attenuation of the pattern as it is processed by the transformation circuit).

Interestingly, perirhinal identity neurons, which normally fire irrespective of location, can appear as spatially selective trace cells due to the periodic modulation of top-down connectivity at the location of encoding. *Figure 10D3* shows the firing rate map of a perirhinal identity neuron. Every time the memorized representation is probed (high top-down gain), if the agent is near the location of

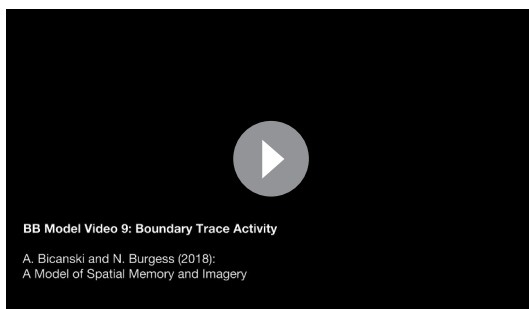

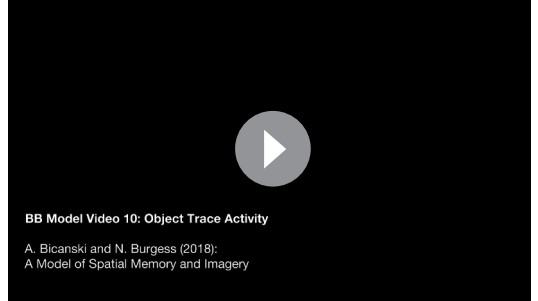

**Video 9.** This video shows a visualization of the simulated neural activity as the agent moves in a familiar environment. However, a previously present boundary has been removed. The agent is supplied with a periodic (akin to rodent theta) modulation of the top-down connection weights (please see main text). The periodic modulation of these connections allows for a probing of the memorized spatial context without engaging in full recall and reveals the memory of the environment to be incongruent with the perceived environment. BVC activity due to the memorized (now removed) boundary periodically 'bleeds' into the egocentric parietal window ¨representation, in principle allowing the agent to attend to the part of environment which has undergone change (location of removed boundary). Time integrated neural activity from this simulation yields firing rate maps which show traces of the removed boundary (see *Figure 10* in the manuscript). Note, the video is cut after 1 min to reduce filesize. The full simulation covers approximately 300 s of real time. Please see caption of Video 1 for abbreviations.

DOI: https://doi.org/10.7554/eLife.33752.022

**Video 10.** This video shows a visualization of the simulated neural activity as the agent moves in a familiar environment. However, a previously present (and encoded) object has been removed. The agent is supplied with a periodic (akin to rodent theta) modulation of the top-down connection weights (please see main text). The periodic modulation of these connections allows for a probing of the memorized spatial context. With every passing through the encoding location OVC activity (reflecting the now removed object) and perirhinal activity is generated by place cells covering the encoding location. This re-activation yields firing rate maps which show traces of the removed object in OVCs, and induces a spatial firing field for the nominally non-spatially selective perirhinal neuron (compare to *Figure 10* in the manuscript). Note, the video is cut after 1 min to reduce filesize. The full simulation covers approximately 300 s of real time. Please see caption of Video 1 for abbreviations.

DOI: https://doi.org/10.7554/eLife.33752.023

encoding, the learned connection from PCs to perirhinal cells (PRo) lead to perirhinal firing for the absent object, yielding a spatial trace firing field for this nominally non-spatial cell.

The presence of some memory-related activity during nominally bottom-up (perceptual) processing can have benefits beyond the assessment of change discussed above. For instance, additional activity in the contextual representations (BVCs, PC, PRb neurons) due to pattern completion in the MTL can enhance the firing of BVCs coding for scene elements outside the current field of view. This activity can propagate to the PW, as is readily apparent during full recall/imagery (*Figure 5*) but is also present in weaker form during perception. Such activity may support awareness of our spatial surrounding outside of the immediate field of view, or may enhance perceptually driven representations when sensory inputs are weak or noisy.

## Sampling multiple objects in imagery (Simulation 3.0)

Humans can focus attention on different elements in an imagined scene, sampling one after another, without necessarily adopting a new imagined viewpoint. This implies that the set of active PCs need not change while different objects are inspected in imagery. Moreover, humans can localize an object in imagined scenes and retrieve its identity (e.g. 'What was next to the fireplace in the restaurant we ate at?").

In Simulation 1.0 (encoding and object-cued recall, *Figure 5*), in addition to connection weights from perirhinal (PRo) neurons to PCs and OVCs, the reciprocal weights from OVCs to PRo neurons were also learned. These connections allow the model to sample and inspect different objects in an imagined scene. To illustrate this we place two objects in a scene and allow the agent to encode

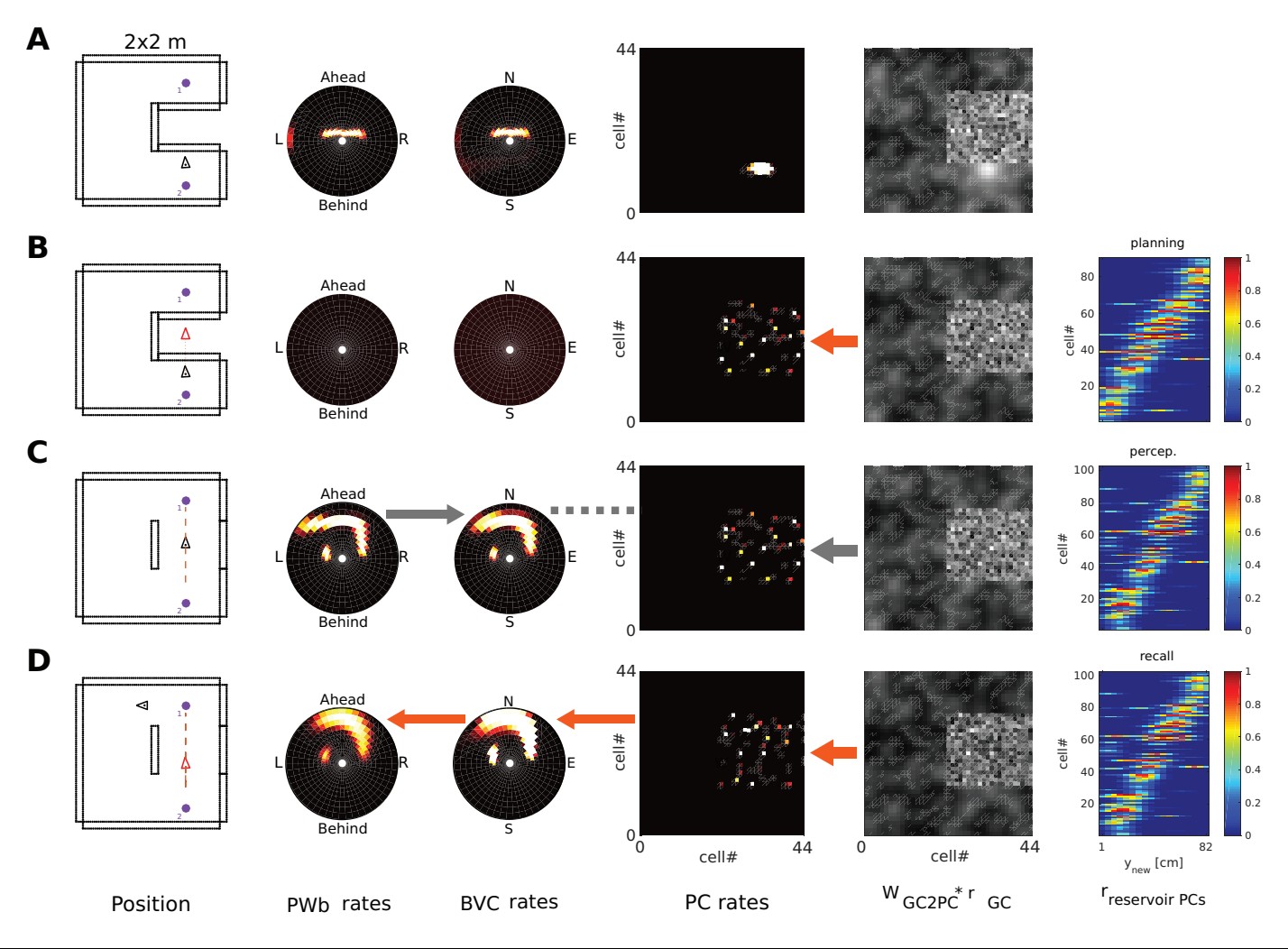

**Figure 13.** Planning, taking and imaging a trajectory across an unexplored area. The agent is located in an environment where the direct trajectory between two salient locations (purple dots, left column) covers an unexplored part of the environment. PCs potentially firing in the unexplored area ('reservoir cells') receive only random connections from GCs (see unstructured grid cell input in column 5). Left to right panel columns: allocentric agent position (triangle); PWb, BVC, PC population rates; GC inputs to PCs (see *Figure 12*); and (B-D only) firing of 'reservoir' PCs along the trajectory (x axis), stacked and ordered by time of peak firing along the trajectory (y axis). (**A**) Starting situation. (**B**) Phase 1; imagined movement across the obstructed space leads to preplay-like activity in reservoir PCs (rightmost panel). Red arrow indicates the reservoir PCs are driven by grid cells. No egocentric representation can be generated from BVCs because 'reservoir' PCs have no connections to BVCs, that is they are not yet part of the MTL attractor. (**C**) Phase 2; the barrier is removed and the agent navigates the trajectory in real space. GCs again drive PCs (thick grey arrow), so the temporal sequence of reservoir cell activity in (A) is recapitulated in the spatial sequence of PC activity. Sensory inputs drive the PW (bottom-up mode) and hence BVCs (grey arrow between panels 2 and 3). Hebbian learning proceeds between PCs and BVCs (dashed grey line), and from GCs to PCs (reinforcing the drive from GCs to PCs, grey arrow between panels 4 and 5). (**D**) Phase 3; having traversed the novel part of the environment, the agent initiates imagery and performs mental navigation along the newly learned trajectory. The learned connections now instantiate the correct BVC and PW activity in top-down mode (orange arrows indicating flow of information, similar to *Figure 12*).

DOI: https://doi.org/10.7554/eLife.33752.027

both visible objects from the same point of view. Encoding still proceeds sequentially. That is, our attention model first samples one object (boosting its activity in the PW) and then the other.

We propose that encoded objects that are not currently the focus of attention in imagery can attract attention by virtue of their residual activity in the parietal window (weak secondary peak in the PWo population in *Figure 11B*). Thus, any of these targets can be focused on by scanning the parietal window and shifting attention to the corresponding location (e.g. 'the next object on a table'). Boosting the drive to such a cluster of PWo cells in the parietal window leads to

corresponding activity in the OVC population (via the retrosplenial transformation circuit). The learned connection from OVCs to perirhinal PRo neurons will then drive PRo activity corresponding to the object which, at the time of encoding, was at the location in peripersonal space which is now the new focus of attention. Mutual inhibition between PRo neurons suppresses the previously active PRo neuron. The result is a top-down drive of perirhinal neurons (as opposed to bottom-up object recognition), which allows inferring the identity of a given object. That is, by shifting its focus of attention in peripersonal space (i.e. in the parietal window) during imagery the agent can infer the identity of scene elements which it did not initially recall.

Figure 11 and Video 11 show sequential (attention-based) encoding, subsequent recall and attentional sampling of scene elements. The agent sequentially encodes two objects from one location (Figure 11A), moves on until both objects are out of view, and engages imagery to recall object one in its spatial context (Figure 11B). The agent can then sample object two by allocating attention to the secondary peak in the parietal window (boosting the residual activity by injecting current in the PWo cells corresponding to the location of object 2). This activity spreads back to the MTL network, via OVCs, driving the corresponding PRo neuron (Figure 11C). Thus, the agent infers the identity of object 2, by inspecting it in imagery. Attention ensures disambiguation of objects at encoding, while reciprocity of connections in the MTL is necessary to form a stored attractor in spatial memory.

## Grid cells and mental navigation (Simulation 4.0)

The parietal window neurons encode the perceived spatial layout of an environment, in an egocentric frame of reference, as an agent explores it (i.e. a representation of the current point of view). In imagery, this viewpoint onto a scene is reconstructed from memory (top-down mode as opposed to bottom-up mode). We refer to mental navigation as internally driven translation and rotation of the viewpoint in the absence of perceptual input. In Simulation 4.0, we let the agent encode a set of objects into memory and then perform mental navigation with the help of grid and head direction cells.

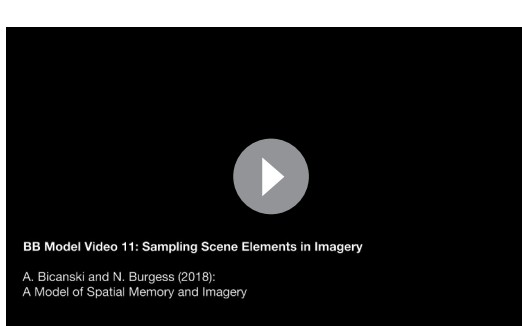

**Video 11.** This video shows a visualization of the simulated neural activity as the agent sequentially encodes two objects into long-term memory. Upon navigating past the objects the agent initiates recall, cueing with the first object. The OVC representations of both objects are bound to the same place cells. These place cells thus generate a secondary peak in the OVC representation corresponding to the non-cued object. This activity propagates to the parietal window. Allocating attention to this secondary peak in the egocentric parietal representation (i.e. injecting current), propagates back to OVCs, which then drive the perirhinal cells for the non-cued object. That is, the agent infers the identity of the second object which is part of the scene (see main text). Please see caption of Video 1 for abbreviations.

DOI: https://doi.org/10.7554/eLife.33752.025

Grid cell (GC) firing is thought to update the location represented by place cell firing, driven by signals relating to self-motion (O'Keefe and Burgess, 2005; McNaughton et al., 2006; Fuhs and Touretzky, 2006; Solstad et al., 2006). During imagination, we suppose that GC firing is driven by mock motor-efference signals (i.e. imagined movement without actual motor output) and used to translate the activity bump on the sheet of place cells. Pattern completion in the MTL network would then update the BVC population activity accordingly, which will spread through the transformation circuit and update parietal window activity. That is, mock motor efference could smoothly translate the viewpoint in imagery (i.e. scene elements represented in the parietal window flow past the point of view of the agent). Similarly, mock rotational signals to the head direction attractor could rotate the viewpoint in imagery. Both together are sufficient to implement mental navigation.

GCs are implemented heuristically, approximating the output of more sophisticated models (e.g., Burgess et al., 2007; Burak and Fiete, 2009; Bush and Burgess, 2014). Firing rate maps for 7 modules of 100 cells each are pre-calculated (see Appendix), providing the firing rates of GCs as a function of location. GC to PC weights are pre-calculated as Hebbian

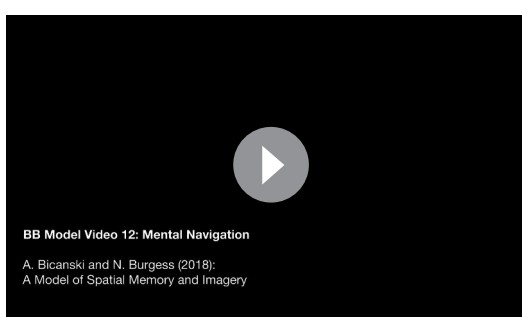

**Video 12.** This video shows a visualization of the simulated neural activity as the agent performs a complex trajectory and encodes three objects into long-term memory along the way. Upon navigating past the third object the agent initiates recall, cueing with the first object, and subsequently performs mental navigation (imagined movement in visuo-spatial imagery) with the help of grid cells. Grid cells update the place cell representation along the trajectory. The egocentric parietal representation is updated along the imagined trajectory, that is scene elements are flowing past the point of view of the agent. Note, the imagined trajectory does not correspond to a previously taken route. Nevertheless, when the imagined trajectory takes the agent past the encoding location of object 3, it is instantiated in the OVC and PWo representations (see main text). Grid cell firing rates are shown multiplied by their connection weights to place cells. Please see caption of Video 1 for further abbreviations.

DOI: https://doi.org/10.7554/eLife.33752.028

associations (to simulate a familiar environment), where the connection strength is maximal if the center of a PC's receptive field coincides with (one of) the GC's firing peaks. During (bottom-up) perception and navigation, GC input provides a small contribution to PC activity, which is mainly determined by BVC inputs (*O'Keefe and Burgess, 1996*; *Hartley et al., 2000*; *Lever et al., 2009*), to highlight the ability of the model to self-localize based on sensory inputs. Stronger grid cell input simply makes the location estimate more stable without detriment to the model. In the absence of reliable sensory information strong GC inputs are required to make PCs fire reliably (*Bush et al., 2014*; *Poucet et al., 2014*; *Evans et al., 2016*). Imagery is an extreme case of this situation, where no sensory input is provided to PCs. Consequently, GC input is up-regulated during imagery (similar to other connections in the switch from bottom-up to top-down modes), constituting a major input to PCs. This GC input can then translate the agent's viewpoint in imagery (via their effect on PCs) without directly affecting the transformation circuit.

*Figure 12* and *Video 12* show an example of mental navigation. The agent approaches three objects in sequence, encodes them into memory and then initiates recall cued by object 1. From that (imagined) location, it initiates mental navigation in a straight line. GCs shift the PC activity bump along the trajectory. The allocentric boundary representation (BVCs) follows the shifting PCs (due to pattern completion) and the retrosplenial transformation circuit (RSC/TR) translates the shifting BVC representation into a shifting (i.e. 'flowing past the agent') egocentric representation of boundary distance (imagery of motion in an imagined scene, not shown in *Figure 12*, however, see *Video 12*). Importantly, the imagined trajectory takes the agent through the area of space at which object three was encoded, however this time coming from a novel direction. The transformation circuit nevertheless instantiates the correct activity in the parietal window for object 3, making it appear to the agent's right, instead of to it's left (as during its original encoding, coming from object 2). Not only does the object populate the imagined scene as the agent mentally navigates past it, the event also generates an imagined representation which has never been experienced by the agent.

Translating an established BVC pattern due to updated perceptual input (in response to real motion) also translates the PC activity bump. In fact, this is how perceptual information updates the estimate of position (self-localization) in a familiar environment (PWb→RSC→BVC→PC) in the model. Similarly, shifting the activity pattern across PCs via GCs in mental navigation can update the parietal window (PWb) during mental navigation (GC→PC→BVC→RSC→PWb). With this account of mental exploration of different routes (including potentially novel imagined experiences; see next section), the model provides a neural implementation of important aspects of 'scene construction' (*Hassabis et al., 2007*) and 'episodic future thinking' (*Schacter et al., 2007*), although these concepts also extend beyond the capabilities of the model (see Discussion). The inclusion of GCs allows for a parsimonious account of mental navigation in humans, consistent with observation of grid-like activity during imagined movement through familiar environments (*Bellmund et al., 2016*; Horner et al. 2016).

## Shortcutting and 'preplay-like' activity (Simulation 5.0)

It is a small step from imagined movement to planned navigation. GCs have been suggested to compute the vector to a goal location from the current location (see *Kubie and Fenton, 2012*; *Erdem and Hasselmo, 2012*; *Bush et al., 2015*; *Stemmler et al., 2015*), a capability necessary to explain the ability to take a shortcut across previously unexplored territory (*Tolman, 1948*). We propose that this ability is based on mental (vector-based) navigation supported by GCs. In Simulation 5, we let the agent explore a novel part of the environment, extending a pre-existing representation of a spatial context. Simulation 5.0 consists of three distinct phases: planning movement across a previously unvisited area to a reward location (phase 1); actual navigation of this shortcut (phase 2); and finally mental navigation across the now familiar area (phase 3).

In phase 1, the agent generates a trajectory along the shortest path to the goal using GCs (i.e. a straight line where the barriers happened to be in the way, *Figure 13B*). However, unlike in Simulation 4.0 (*Figure 12*), this process differs from mental navigation since the unexplored part of the environment is devoid of any meaningful PC-BVC connections and so a scene cannot be generated in the parietal window (PWb). Extending the medial temporal lobe (MTL) representations requires incorporating additional place cells into the MTL attractor. These (future) place cells are referred to as 'reservoir cells' and have no relationship to physical space yet, so visualizing their firing rates in a topographic manner is not possible. However, as the agent generates a trajectory towards its goal using GCs, sparse random GC-to-PC connections cause a subset of the reservoir cells to fire (*Figure 13B* and *Video 13*). The activity of reservoir cells does not form an attractor bump, as PC-PC connections have not been learned, but their firing is normalised to a level of activity similar to when an attractor bump is present (implemented by an adaptive feedback current, see Appendix). In *Figure 13B* (rightmost panel) reservoir cells are ordered according to their time of maximum firing along the imagined trajectory.

In phase 2, the barriers are removed and the agent performs the previously imagined trajectory in real space, using the novel shortcut. The same GCs are active along the trajectory and hence the same reservoir cells which fired before exploring the area, now fire in a spatial sequence along the actual trajectory. Since the agent is now actively perceiving its environment BVCs are driven in a bottom-up manner and Hebbian-like plasticity can strengthen connections between BVCs and reservoir cells as they fire along the trajectory (an analogous mechanism should also associate perirhinal neu-rons, which is omitted here). Hence, the reservoir cells have now effectively become place cells, with firing fields tied to the agent's location in space (*Figure 13C* and *Video 13*). *Figure 13C* (rightmost panel) shows place cell activity along the trajectory during phase 2. Crucially, these cells are plotted in the order of activity shown during the previous imagined navigation (*Figure 13B*), indicating 'pre-play-like' behavior, in that the sequence of PC firing seen prior to first exploration is subsequently recapitulated during actual navigation (*Figure 13C*; *Dragoi and Tonegawa, 2011*; *Ólafsdóttir et al., 2015*).

Finally, in phase 3, the agent initiates imagery and performs mental navigation along the short-cut (i.e. recalls the episode of traversal), demonstrating that the MTL representation has been extended, and that a scene can be generated (*Figure 13D* and *Video 13*). The newly learned connections from reservoir PCs to BVCs complete the MTL representation of the spatial context and the transformation circuit reinstates the corresponding parietal window (PWb) representation (imagery, *Figure 10D*, panel 2).

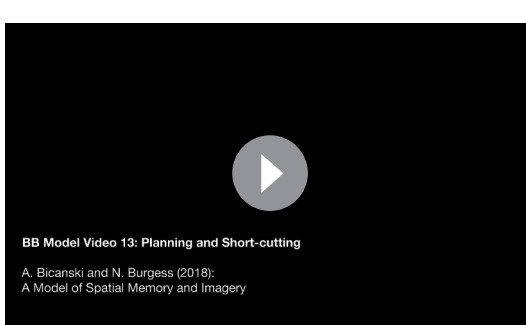

**BB Model Video 13: Planning and Short-cutting**

A. Bicanski and N. Burgess (2018):
A Model of Spatial Memory and Imagery

**Video 13.** This video shows a visualization of the simulated neural activity as the agent performs mental navigation across a blocked shortcut. Newly recruited cells in the hippocampus exhibit activity reminiscent of preplay. Upon removal of the barrier the agent traverses the shortcut and associates the newly recruited hippocampal cells with the perceptually driven activity in the MTL. Subsequent mental navigation across the short cut yields activity in hippocampal cells reminiscent of replay (see main text). Grid cell firing rates are shown multiplied by their connection weights to place cells. Please see caption of Video 1 for further abbreviations.
DOI: https://doi.org/10.7554/eLife.33752.029

The ability to plan a route by driving a sweep of PC activity with a sweep of GC activity via established GC-PC connections (*Figures 12–13*) could relate to the observation of 'forward sweeps' of PC activity during navigation (*Johnson and Redish, 2007*; *Pfeiffer and Foster, 2015*) and 'replay' during rest (*Wilson and McNaughton, 1994*; *Foster and Wilson, 2006*; *Diba and Buzsáki, 2007*; *Karlsson and Frank, 2009*; *Carr et al., 2011*). However, both of these phenomena, and the 'pre-play-like' activity discussed above, occur at compressed timescales in experimental animals. Thus, modeling forward sweeps, replay or pre-play would require a spiking neuron model able to capture the faster time scale of the sharp-wave ripple events associated with replay and pre-play, and the theta sequences associated with forward sweeps (*Burgess et al., 1994*; *Skaggs et al., 1995*; *Gupta et al., 2012*).

## Discussion

We propose a model of how sensory experiences, which are ultimately egocentric in nature, are transformed into viewpoint-invariant representations for long-term spatial memory in the medial temporal lobe (MTL) via processing in parietal and retrosplenial cortices. According to the model, imagery and recollection of scenes correspond to the re-construction of egocentric representations in parietal areas (the parietal window, PWb/o) from MTL representations. The MTL is the repository of viewpoint-invariant knowledge which is used to generate spatially coherent scenes by retrieving information consistent with perception from a single location and orientation. Pattern completion (via attractor dynamics) implements retrieval of a neural representation across the MTL, while head-direction cells enable the translation into egocentric coordinates via a retrosplenial transformation circuit, making use of gain-field neurons (*Snyder et al., 1998*; *Galletti et al., 1995*; *Pouget and Sejnowski, 1997*). Thus, for example, unilateral lesions in parietal regions could cause perceptual hemispatial neglect, and unilateral lesions to parietal or retrosplenial cortex could cause representational hemispatial neglect (in imagery) for a scene for which the MTL representation is complete (*Bisiach and Luzzatti, 1978*; see also *Pouget and Sejnowski, 1997*; *Burgess et al., 2001a*; *Byrne et al., 2007*).

The model can be used to account for human spatial cognition at the level of single neurons far from the sensory periphery: Place cells, head direction cells, gain-field neurons, boundary- and object-vector cells (BVCs and OVCs), and grid cells. Future work should try to integrate the present account of spatial cognition with recent progress concerning spatial coding in parietal areas (*Nitz, 2006*; *Nitz, 2009*; *Nitz, 2012*; *Harvey et al., 2012*; *Whitlock et al., 2012*; *Raposo et al., 2014*; *Vedder et al., 2017*), and a broader view of retrosplenial function (e.g., *Alexander and Nitz, 2015*; *Alexander and Nitz, 2017*). Notably, BVCs were predicted by an early predecessor of the present model (*Hartley et al., 2000*; *Burgess et al., 2001a*). Here, we have introduced OVCs to show how items introduced into a familiar environment may be coded for and incorporated into long-term memory. Intriguingly, OVC-like responses have been reported recently (*Deshmukh and Knierim, 2013*; *Hoydal et al., 2017*). We also explored how long-term memory might be probed to assess novelty. Finally, we incorporated grid cells and investigated their role in exploratory behavior and mental navigation. We can thus begin to frame abstract notions such as episodic future thinking and scene construction in terms of neural mechanisms, although we note that these concepts extend beyond our model to include completely fictional scenes/scenarios (*Burgess et al., 2001a*; *Hassabis et al., 2007*; *Schacter et al., 2007*).

### Recall of objects in a spatial context

We have proposed that items/objects are associated to a given (spatial) context via place cells, which index the local sensory panorama, including local objects. Attaching representations of discrete objects (in the form of object vector cell activity) to a contextual representation via place cells aligns well with neuropsychological experiments that show position specificity in visual object memory (*Hollingworth, 2007*). In such experiments, object memory is superior when the target object is presented at the same position in the scene as it had been viewed originally (also see object novelty Simulations 1.3, 1.4). The hippocampus in particular has been implicated in combining information about objects, locations, and contexts (*Warburton and Brown, 2010*; *Eacott and Gaffan, 2005*; *Barker and Warburton, 2015*), consistent with the model. Similarly, studies suggest the

hippocampus and precuneus are necessary for maintaining object-location binding even in working memory (*Piekema et al., 2006*; *Olson et al., 2006*).

The direction-independence of place cell firing in open environments implies that all possible local views at a given location could be associated with the corresponding place cells, potentially encompassing the boundaries in all directions around that location. Only by supplying head (or gaze) direction, and transforming the activity to the parietal window, a specific point of view can be represented. Note that given the anatomical loci of head direction cells along Papez circuit, the role of head direction as a modulatory factor in the egocentric-allocentric transformation (modeled as within retrosplenial cortex) provides a good explanation for impaired episodic memory resulting from Papez circuit lesions (*Figure 7*; *Delay and Brion, 1969*; *Squire and Slater, 1978*; *1989*; *Parker and Gaffan, 1997*; *Aggleton et al., 2016*; *Tsivilis et al., 2008*). It also explains why permanent landmarks should evoke stronger responses in retrosplenial cortex (*Auger et al., 2012*), because permanent landmarks provide a more stable directional reference for the transformation circuit (see also *Bicanski and Burgess, 2016*). In summary, head direction cells likely serve to specify a direction of view, and not a movement direction (*Raudies et al., 2015*), which could instead be expressed in the firing phase of grid cells or place cells (*Maurer et al., 2014*; *Cei et al., 2014*).

The encoding strategy for objects allows an agent to reactivate the set of place cells which were active when the object was encountered and thus reconstruct the local view at encoding in the parietal window. This models the explicit recollection of a spatial scene populated with objects as an act of visuo-spatial imagery. It provides an explanation for the neural activity seen in the MTL, retrosplenial cortex and precuneus during imagery for familiar scenes (*Burgess et al., 2001a*; *Hassabis et al., 2007*; *Schacter et al., 2007*). The agent could also use the place cells activated during imagery as 'goal cells' and use grid cells to calculate a vector to navigate to the remembered location (*Bush et al., 2015*; not simulated here), accounting for the role of the MTL in goal-directed navigation (e.g. reviewed in *Burgess et al., 2002*).

The present model of explicit recall for items in context is a small step on the long road to understanding episodic memory at the neuronal level. However, not all memories for items requires reconstruction of a spatial scene. Recall of factual information (semantic memory) is not modeled, while memory for the attributes of an object irrespective of its context would require only perirhinal involvement. The BB-model only applies to imagery for coherent spatial scenes, and suggests that this is necessary for episodic recollection in which the past event is 're-experienced' (*Tulving, 1983*), and certainly for remembering the spatial context of encountering an object.

Key components of the model are the 'bottom-up' transition from egocentric perceptual representations to allocentric MTL representations and the 'top-down' transition from MTL representations back to egocentric imagery. By informing perception in a top-down manner, the MTL can effectively predict perceptual input in familiar environments, allowing novelty detection and enhancing perception with remembered information. If we view imagery as a top-down reconstruction of perceptual representations, the MTL together with the retrosplenial transformation circuit could be seen as a generative model for scenes, consistent with generative models of memory such as (*Káli and Dayan, 2001*). It has been proposed that the bottom-up/top-down transition between encoding and retrieving occurs rhythmically at the frequency of the theta rhythm (*Hasselmo et al., 1996*; *Burgess et al., 2001a*; *Hasselmo et al., 2002*; *Byrne et al., 2007*; *Douchamps et al., 2013*). Theta might underlie a periodic probing of memorized representations; however, full recollection in imagery can last for long periods of time and need not correspond to specific phases of theta in humans (*Düzel et al., 2010*).

## Mnemonic effects of newly learned connections and 'trace cells'

We have proposed (*Figure 10*) that the relative strength of top-down and bottom-up connections can change smoothly and under control of the agent (e.g. via the release of a neuromodulator) to allow memory representations to influence neural activity during perception. This allows the agent to localize and attend to a region of space in the egocentric frame of reference where a given scene element used to be located, even if it has subsequently been moved, changed or removed. Moreover, the neural activity caused by increasing top-down connections can signal where the environment has changed. Interestingly, *Tsao et al. (2013)* recently reported 'trace cells' in lateral entorhinal cortex, whose firing reflects the previous presence of a now missing object, while related 'mis-place' cells have been reported in CA1 (*O'Keefe, 1976*).

We have shown that nominally non-spatially selective cells like perirhinal identity neurons can manifest a spatial trace firing field when re-activation occurs at the encoding location (*Figure 10D3*). This may help to reconcile the notion that lateral entorhinal cortex processes non-spatial information (*Van Cauter et al., 2013*; *Hargreaves et al., 2005*) with the spatial responses of trace cells (*Tsao et al., 2013*) in lateral entorhinal cortex. However, the trace cells of *Tsao et al. (2013)* do not fire when the object is present, but only in the subsequent absence of the object. Thus they might signal the mismatch between the remembered object and its absence, that is reflecting a comparison of perceptually driven and memory driven firing of the model perirhinal cells.

Finally, even in the absence of changes to the memorized spatial configuration, mnemonic representations can enhance perception, for example allowing the firing of cells coding for scene elements outside the current field of view. This activity is supported by pattern completion in the MTL, and may support people's awareness of the presence of boundaries or objects outside of their field of view within a familiar environment.

## Attention

Although we do not model the mechanistic origins of attention (see e.g. *Itti and Koch, 2001*), attentional modulation in the present model is crucial for unambiguous representations of multiple objects within a scene. If multiple objects are encoded from the same viewpoint, multiple OVC and perirhinal (PRo) neurons can be co-active, precluding the formation of a unique representation for each object-location conjunction, that is, precluding the solving the object-location binding problem. Thus, we require the objects to be sampled rhythmically and encoded sequentially in the parietal window (*Figures 3* and *11*), consistent with experimental literature suggesting rhythmic and sequential sampling (*VanRullen et al., 2007*; *Landau and Fries, 2012*; for review see *VanRullen, 2013*). If attentional cycles have a limited duration, then there may be insufficient time for activity to build up in the corresponding neuronal populations and support robust encoding into memory if there are too many objects within a scene, producing a capacity limit (see also *Lisman and Idiart, 1995*; *Bays and Husain, 2008*).

The attentional modulation described above can also act in imagery (within the parietal window), allowing the agent to inspect different parts of an imagined scene (see also *Byrne et al., 2007*). The model proposes that, in the absence of perceptual inputs, perirhinal neurons can be driven in a top-down fashion from hippocampus, thus reinstating an activity pattern in perirhinal cortex similar to the one present at encoding. Only the co-firing of these perirhinal neurons (PRo) and the corresponding BVCs and OVCs provides a unique representation of a given object in a given context, at a given location. The proposed binding of OVCs and PRo neurons, subject to attention, might also provide a functional interpretation of the hippocampus' role in memory-guided attention (e.g., *Summerfield et al., 2006*).

## Mental navigation, short-cutting, and planning

The model suggests a role for grid cell activity in human spatial cognition. Since both self-motion related inputs (via grid cells) and sensory inputs converge onto place cells, grid cells can update the point of view and allow an agent to translate its imagined location. If imagery can inform degraded perception (e.g. in the dark), obstacles can be identified and a suitable path can be planned. Thus, although mental navigation cannot be equated with path integration, we suggest that they reflect a common grid cell-dependent mechanism, which is required when sensory inputs are absent or unreliable. Indeed, humans likely make use of spatial imagery even in apparently non-visual tasks such as triangle completion in darkness (*Tcheang et al., 2011*), and there is evidence for grid-like brain activity during mental navigation (*Bellmund et al., 2016*; Horner et al. 2016).

The model of mental navigation provides a mechanistic neural-level account of some aspects of 'scene construction' and 'episodic future thinking' (*Schacter et al., 2007*; *Hassabis et al., 2007*; *Buckner, 2010*) with regard to familiar spaces. Mental navigation allows an agent to test future behavior, like the approach of a target from a new direction as depicted in *Video 12*. This suggests that the same neural infrastructure involved in scene perception and reconstruction also subserves planning and hypothesis testing (e.g. asking 'Which way should I go?' or 'What would I encounter if I went that way?'). If grid cells (acting on place cells) change the point of view during imagined movement this must be reconciled with the relationships between grid cells and place cells seen during

periods of rest or planning (see e.g., *Ólafsdóttir et al., 2016*; *O'Neill et al., 2017*; *Trettel et al., 2017*; *Buzsáki and Chrobak, 1995*).

Grid cells have been proposed to support the computation of vectors to a goal (*Kubie and Fenton, 2012*; *Erdem and Hasselmo, 2012*; *Bush et al., 2015*; *Stemmler et al., 2015*). That is, they can plan trajectories across known and potentially unknown terrain (shortcuts). We proposed that grid cells recruit new hippocampal cells (future place cells) in previously unexplored parts of a familiar environment (*Figure 13* and *Video 13*). Planning a trajectory across unexplored space engenders preplay-like activity in place cells (*Dragoi and Tonegawa, 2011*; *Ólafsdóttir et al., 2015*), whereas mental navigation is reminiscent of 'replay' (*Wilson and McNaughton, 1994*; *Foster and Wilson, 2006*; *Diba and Buzsáki, 2007*; *Karlsson and Frank, 2009*; *Carr et al., 2011*) or 'forward sweeps (*Johnson and Redish, 2007*; *Pfeiffer and Foster, 2015*), although the faster propagation speed (e.g. during sharp wave ripples) of these sequences of place cell activity are beyond the scope of the present model. Nevertheless, the model suggests that sweeps of activity in the grid cell population may play are role in these aspects of place cell firing, and could correspond to route planning (*Kubie and Fenton, 2012*; *Erdem and Hasselmo, 2012*; *Bush et al., 2015*; *Yamamoto and Tonegawa, 2017*).

## Conclusions

It has been argued that the MTL-retrosplenial-parietal system supports the construction of coherent scenes (*Burgess et al., 2001b*; *Byrne et al., 2007*; *Hassabis et al., 2007*; *Schacter et al., 2007*; *Buckner, 2010*). However, if recollection corresponds to the (re-)construction of something akin to a perceptual experience (the defining characteristic of episodic memory; Tulving 1985), then this places strong spatial constraints on how episodic memory works. A vast number of different combinations of information could be retrieved from the body of long-term knowledge in the MTL, but only a small subset would be consistent with a single point of view, making the episodic 're-experiencing' of events or visuo-spatial imagery congruent with perceptual experiences. The BB-model combines this insight with established knowledge and new hypotheses about how location, orientation, and surrounding environmental features are associated and represented by neural population activity.

This account includes functional roles for the specific firing characteristics of diverse populations of spatially selective cells across multiple brain regions, and distinguishes the egocentric representations supporting conscious (re-)experience from the more abstract (allocentric) representations involved in supporting computations. The resultant systems-level account provides a strong conceptual framework for considering the interplay between structures in the MTL, retrosplenial cortex, Papez circuit' and parietal cortex in support of spatial memory. It follows Tulving's theoretical specification of episodic memory, and - spanning Marr's theoretical, algorithmic and implementational levels (*Marr and Poggio, 1976*) - bridges the gap between a neuropsychological description of spatial cognition (founded on behavioral and functional imaging data) and the neural representations supporting it.

## Acknowledgements

We acknowledge funding by the ERC Advanced grant NEUROMEM, the Wellcome Trust, the European Union's Horizon 2020 research and innovation programme under grant agreement No. 720270 Human Brain Project SGA1 and grant agreement No. 785907 Human Brain Project SGA2, and the EC Framework Program 7 Future and Emerging Technologies project SpaceCog. We thank all members of the SpaceCog project, and James Bisby, Daniel Bush and Tim Behrens for useful discussions. The authors declare no competing financial interests.

## Additional information

**Competing interests**
Neil Burgess: Reviewing Editor, eLife. The other author declares that no competing interests exist.

## Funding

| Funder | Grant reference number | Author |
|---|---|---|
| European Research Council | NEUROMEM | Andrej Bicanski<br>Neil Burgess |
| Human Brain Project SGA1 | 720270 | Andrej Bicanski<br>Neil Burgess |
| European Commission | SpaceCog | Andrej Bicanski<br>Neil Burgess |
| Human Brain Project SGA2 | 785907 | Andrej Bicanski<br>Neil Burgess |
| Wellcome Trust | | Neil Burgess |

The funders had no role in study design, data collection and interpretation, or the decision to submit the work for publication.

## Author contributions

Andrej Bicanski, Conceptualization, Software, Formal analysis, Investigation, Visualization, Methodology, Writing—original draft, Project administration, Writing—review and editing; Neil Burgess, Conceptualization, Supervision, Funding acquisition, Methodology, Project administration, Writing—review and editing

## Author ORCIDs

Andrej Bicanski http://orcid.org/0000-0003-3356-1034
Neil Burgess https://orcid.org/0000-0003-0646-6584

## Decision letter and Author response

Decision letter https://doi.org/10.7554/eLife.33752.035
Author response https://doi.org/10.7554/eLife.33752.036

# Additional files

## Supplementary files

• Transparent reporting form
DOI: https://doi.org/10.7554/eLife.33752.030

## Data availability

Matlab code to build all model components and run all simulations will be made available on GitHub in the following repository: https://github.com/bicanski/HBPcollab/tree/master/SpatialEpisodicMemoryModel; copy archived at https://github.com/elifesciences-publications/HBPcollab/tree/master/SpatialEpisodicMemoryModel.

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

## Appendix 1

DOI: https://doi.org/10.7554/eLife.33752.031

## BB-Model Details

### Neuron model

All neuron populations in the BB-model, with the exception of for grid cells, are composed of rate-coded neurons and implemented according to the following equations.

$$x_i^{t+1} = x_i^t + \frac{dt}{\tau} k_i^t \tag{1}$$

$$r_i^{t+1} = \frac{1}{1 + exp\left(-2\beta_i\left(x_i^{t+1} - \alpha_i\right)\right)} \tag{2}$$

Where $x$ is the vector of activations (*Equation 1*, vectors/matrices displayed in bold) for all neurons belonging to the population marked by the subscript $i$ (e.g. PCs, BVC, etc.). Within a population all neurons are identical. The superscript indicates the temporal dimension, with $t + 1$ referring to the updated state variable for the next time step (step size $dt$). $\tau$ is the decay time-constant of the rate equation. The sigmoid with parameters $\alpha$, $\beta$ (*Equation 2*) serves as a non-linearity to map activations onto firing rates. The term $k_i$ in *Equation 1* contains all population specific inputs. *Equations 3* through 13 summarize the inputs to the model populations.

$$
\begin{aligned}
k_{PC} \quad &= -x_{PC} + \varphi_{PC,PC} W_{PC,PC} r_{PC} + P_{mod} \varphi_{BVC,PC} W_{BVC,PC} r_{BVC} \\
&+ \varphi_{PRb,PC} W_{PRb,PC} r_{PRb} + \varphi_{OVC,PC} W_{OVC,PC} r_{OVC} \\
&+ I_{mod} \varphi_{PRo,PC} W_{PRo,PC} r_{PRo} + \varphi_{GC,PC} W_{GC,PC} r_{GC} + I_{FB}
\end{aligned} \tag{3}
$$

Here (and below) $W_{i,j}$ is the matrix of connection weights from population $i$ to $j$, $\varphi_{i,j}$ is a gain factor, and $r_j$ refers to the vector of firing rates of population $j$. $I_{FB}$ is a feedback current ensuring a set total of activity in the place cell sheet (numerical value 15). $I_{mod}$ and $P_{mod}$ refer to neuromodulation for bottom-up vs top-down modes of operation, and the mode is set determined externally. that is setting these values according to behavioural needs of the agent (perception vs imagery/recollection) implements the switch between bottom-up and top-down modes. $P_{mod}$ is one in bottom-up mode of operation and 0.05 in top-down mode. $I_{mod}$ is 0.05 in bottom-up mode of operation and one in top-down mode. Abbreviations: PC; place cells, BVC; boundary vector cells, OVC; object vector cells; PRb; boundary selective perirhinal neurons, PRo; object selective perirhinal neurons, PW; parietal window neurons, TR; transformation circuit neurons, HDC; head direction cells, GC; grid cells.

$$
\begin{aligned}
k_{BVC} \quad &= -x_{BVC} + B I_{mod} \varphi_{PC,BVC} W_{PC,BVC} r_{PC} + \varphi_{BVC,PC} W_{BVC,PC} r_{BVC} \\
&+ \varphi_{PRb,BVC} W_{PRb,BVC} r_{PRb} + B^{-1} P_{mod} \varphi_{TRb,BVC} W_{TR,BVC} \sum_{i=1}^{20} r_{TRb_i}
\end{aligned} \tag{4}
$$

$$
\begin{aligned}
k_{OVC} \quad &= -x_{OVC} + \varphi_{OVC,OVC} W_{OVC,OVC} r_{OVC} + B I_{mod} \varphi_{PC,OVC} W_{PC,OVC} r_{PC} \\
&+ B I_{mod} \varphi_{PRo,OVC} W_{PRo,OVC} r_{PRo} + B^{-1} P_{mod} \varphi_{TRo,OVC} W_{TR,BVC} \sum_{i=1}^{20} r_{TRo_i}
\end{aligned} \tag{5}
$$

B is the 'bleed' parameter for a smooth modulation of bottom-up vs top-down connectivity, during perception (simulations 2.1, 2.2). Sums over transformation sublayers run from 1 to 20, the number of distinct sublayers (see description of transformation circuit in main text and below).

$$k_{PRb} = -x_{PRb} + I_{mod} \varphi_{PC,PRb} W_{PC,PRb} r_{PC} + \varphi_{BVC,PRb} W_{BVC,PRb} r_{BVC} + I_{PRb} \tag{6}$$

$$k_{PRo} = -x_{PRo} + \varphi_{PRo,PRo} W_{PRo,PRo} r_{PRo} + \varphi_{PC,PRo} W_{PC,PRo} r_{PC}$$
$$+ \varphi_{OVC,PRo} W_{OVC,PRo} r_{OVC} + I_{PRo} + I_{cue} \tag{7}$$

$I_{cue}$ is an externally supplied (i.e. not causally determined by other model components) trigger current to initiate recall in imagery.

$I_{PRo}$ and $I_{PRb}$ are externally supplied inputs to perirhinal identity neurons that represent the result of a recognition process along the ventral visual stream which is not explicitly modelled, and both inputs are only present in bottom-mode (i.e. during perception). $I_{PRo}$ is binary (object attended and present vs not attended/not present), while the magnitude of $I_{PRb}$ depends linearly on the extent of the boundary that is visible and its distance to the agent.

$$k_{PWb} = -x_{PWb} - PWb_{bath} \sum r_{PWb} + B^{-1} I_{PWb}^{agent}$$
$$+ B I_{mod} \varphi_{TR,PWb} \sum_{i=1}^{20} W_{TR,PWb_i} r_{TR_i} \tag{8}$$

$$k_{PWo} = -x_{PWo} - PWo_{bath} \sum r_{PWo} + B^{-1} I_{PWo}^{agent}$$
$$+ B I_{mod} \varphi_{TR,PWo} \sum_{i=1}^{20} W_{TR,PWo_i} r_{TR_i} \tag{9}$$

$PWb/o_{bath}$ is an inhibitory input based on the total activity in the PWb/o population (sum of the population vector in **Equations 8 and 9**). $I_{PWb/o}{}^{agent}$ refers to the sensory/perceptual inputs to the PWb/o populations. That is, these input currents are generated in response to the presence of boundaries/objects in the field of view in order to be injected into the corresponding populations.

$$r_{IP} = \left( 1 + exp\left( -2\beta_{IP} \left( \varphi_{HD,IP} \sum r_{HD} - \alpha_{IP} \right) \right) \right)^{-1} \tag{10}$$

connects onto the different sublayers of the transformation circuit (small hexagon in **Figure 4**), ensuring suppression of activity in all sublayers except where the positive modulatory input from HDCs ensures that inhibition is overcome.

$$k_{TRb}^i = -x_{TRb}^i - TRb_{bath} \sum r_{TRb}^i + \varphi_{HD,TRb} W_{HD,TRb}^i r_{HD}$$
$$+ \varphi_{IP,TRb} r_{IP} + I_{mod} \varphi_{BVC,TRb} W_{BVC,TRb}^i r_{BVC} +$$
$$B^{-1} P_{mod} \varphi_{PWb,TR} W_{PWb,TR}^i r_{PWb} \quad for\ i \in \{1, \dots 20\} \tag{11}$$

$$k_{TRo}^i = -x_{TRo}^i - TRo_{bath} \sum r_{TRo}^i + \varphi_{HD,TRo} W_{HD,TRo}^i r_{HD}$$
$$+ \varphi_{IP,TRo} r_{IP} + I_{mod} \varphi_{OVC,TRo} W_{OVC,TRo} r_{OVC} +$$
$$B^{-1} P_{mod} \varphi_{PWo,TRo} W_{PWo,TRo}^i r_{PWo} \quad for\ i \in \{1, \dots 20\} \tag{12}$$

The superscript $i$ in **Equations 11 and 12** refers to the individual sublayers of the retrosplenial transformation circuit ($i$ ranging from 1 to 20). For convenience and in order to visualize object (item) and boundary (contextual) related representations separately the transformation is applied separately to the PWb/o representations but the same connectivity is used. TRb/o$_{bath}$ are analogous to $PWb/o_{bath}$.

$$ll]k_{HD} = -x_{HD} + \varphi_{HD,HD} W_{HD,HD} r_{HD} + I_{cue} \varphi_{PRo,HD} W_{PRo,HD} r_{PRo}$$
$$+ \varphi_{rot} cw W_{rot} r_{HD} + \varphi_{rot} ccw W_{rot}' r_{HD} \tag{13}$$

cw and ccw in **Equation 13** are 0 or 1 depending on whether on the agent is performing a clockwise or counterclockwise turn, respectively. The scaling factor $\varphi_{rot}$ is set to ensure a match between the agent's rotations speed and the translation of the activity packet in the head direction ring attractor.

The firing rate dynamics of GCs are not modelled. GCs exist as firing rate maps which span the environment. GC rates are sampled from these rate maps by looking up the pixel value closest to the agent's location. See section Grid cell rate maps, mental navigation, and preplay setup for the generation of the grid maps.

See **Appendix 1—table 1** for population sizes.

**Appendix 1—table 1.** Model Parameters. Top to bottom: $\alpha$, $\beta$ sigmoid parameters; $\varphi$ connection gains; $\Phi$ constants subtracted from given weight matrices (e.g. PC to PC connections) to yield global inhibition; bath parameters; range thresholds for object encoding; $l$ learning rates for simulation 5; $S$ sparseness of connections for reservoir PCs; $\sigma_\rho$, $\sigma_\vartheta$ spatial dispersion of the rate function for BVCs. The additive constant ($\sigma_\rho = (r + 8) * \sigma_0$) corresponds to half the range of BVC grid and prevents $\sigma_\rho$ from converging to zero close to the agent. $N_i$ population sizes. Products of numbers reflect geometric and functional aspects. E.g. receptive fields of PCs tile 2 × 2 m arena with 44 × 44 cells. Polar grids are given by 16 radial distance units (see A.2) and 51 angular distance units. For the transformation circuit this number is multiplied by the number of transformation sublayers, that is 20.

| | |
|---|---|
| $\alpha$ | 5 |
| $\beta$ | 0.1 |
| $\alpha_{IP}$ | 50 |
| $\beta_{IP}$ | 0.1 |
| $\varphi_{PWb\text{-}TR}$ | 50 |
| $\varphi_{TR\text{-}PWb}$ | 35 |
| $\varphi_{TR\text{-}BVC}$ | 30 |
| $\varphi_{BVC\text{-}TR}$ | 45 |
| $\varphi_{HD\text{-}HD}$ | 15 |
| $\varphi_{HD\text{-}IP}$ | 10 |
| $\varphi_{HD\text{-}TR}$ | 15 |
| $\varphi_{HDrot}$ | 2 |
| $\varphi_{IP\text{-}TR}$ | 90 |
| $\varphi_{PC\text{-}PC}$ | 25 |
| $\varphi_{PC\text{-}BVC}$ | 1100 |
| $\varphi_{PC\text{-}PRb}$ | 6000 |
| $\varphi_{BVC\text{-}PC}$ | 440 |
| $\varphi_{BVC\text{-}PRb}$ | 75 |
| $\varphi_{PRb\text{-}PC}$ | 25 |
| $\varphi_{PRb\text{-}BVC}$ | 1 |
| $\varphi_{GC\text{-}PC}$ | 3 |
| $\varphi_{PWo\text{-}TR}$ | 60 |
| $\varphi_{TR\text{-}PWo}$ | 30 |
| $\varphi_{TR\text{-}OVC}$ | 60 |
| $\varphi_{OVC\text{-}TR}$ | 30 |
| $\varphi_{PC\text{-}OVC}$ | 1.7 |
| $\varphi_{PRo\text{-}OVC}$ | 6 |
| $\varphi_{PC\text{-}PRo}$ | 1 |
| $\varphi_{OVC\text{-}PC}$ | 5 |
| $\varphi_{OVC\text{-}oPR}$ | 5 |
| $\varphi_{PRo\text{-}PC}$ | 100 |
| $\varphi_{PRo\text{-}PRo}$ | 115 |
| $\varphi_{inh\text{-}PC}$ | 0.4 |
| $\varphi_{inh\text{-}BVC}$ | 0.2 |
| $\varphi_{inh\text{-}PRb}$ | 9 |

*Appendix 1—table 1 continued on next page*

Appendix 1—table 1 continued

| α | 5 |
|---|---|
| $\varphi_{inh\text{-}PRo}$ | 1 |
| $\varphi_{inh\text{-}HD}$ | 0.4 |
| $\varphi_{inh\text{-}TR}$ | 0.075 |
| $\varphi_{inh\text{-}TRo}$ | 0.1 |
| $\varphi_{inh\text{-}PW}$ | 0.1 |
| $\varphi_{inh\text{-}OVC}$ | 0.5 |
| $\varphi_{inh\text{-}PWo}$ | 1 |
| $\Phi_{PC\text{-}PC}$ | 0.4 |
| $\Phi_{BVC\text{-}BVC}$ | 0.2 |
| $\Phi_{PR\text{-}PR}$ | 9 |
| $\Phi_{HD\text{-}HD}$ | 0.4 |
| $\Phi_{OVC\text{-}OVC}$ | 0.5 |
| $\Phi_{PRo\text{-}PRo}$ | 01 |
| $PW_{bath}$ | 0.1 |
| $PW_{bath}$ | 0.2 |
| $TR_{bath}$ | 0.088 |
| Object enc. threshold | 18 cm |
| Object enc. Threshold (3.1) | 36 cm |
| $l_{GC\text{-}resPC}$ | $0.65*10{-}5$ |
| $l_{resPC\text{-}BVC}$ | $0.65*10{-}5$ |
| $l_{BVC\text{-}resPC}$ | $0.65*10{-}5$ |
| $S_{GC\text{-}resPC}$ | 3% |
| $S_{resPC\text{-}resPC}$ | 6% |
| $\sigma_{\rho}$ | $(r + 8) * \sigma_0$ |
| $\sigma_0$ | 0.08 |
| $\sigma_{\vartheta}$ | 0.2236 |
| $N_{PC}$ | $44 \times 44$ |
| $N_{BVC}$ | $16 \times 51$ |
| $N_{TRb/o}$ | $20 \times 16 \times 51$ |
| $N_{OVC}$ | $16 \times 51$ |
| $N_{PRb/o}$ | Dependent on simulation environment |
| $N_{PWb/o}$ | $16 \times 51$ |
| $N_{IP}$ | 1 |
| $N_{HD}$ | 100 |
| $N_{GC}$ | 100 per module |
| $N_{reservoir}$ | 437 |

DOI: https://doi.org/10.7554/eLife.33752.032

## Receptive fields of place cells and boundary vector cells

In the training phase for the contextual representation (see section Connection Profiles) BVC and PWb neurons have activation functions of the following type. If a boundary segment is located at the coordinates $(\rho, \vartheta)$, then the activity of each boundary selective cell is proportional to the distance of its receptive field from that boundary segment. If $(\rho_i, \vartheta_i)$ are the polar coordinates of the receptive field of the i-th BVC or PWb neuron, then the firing rate $r$ is calculated according to the following equation:

$$r_{BVC}^i = \frac{1}{\rho} exp\left(-\left(\frac{\vartheta_i^a - \vartheta^a}{\sigma_\vartheta}\right)^2\right) exp\left(-\left(\frac{\rho_i - \rho}{\sigma_\rho}\right)^2\right) \tag{14}$$

where $\sigma_\vartheta$ and $\sigma_\rho$ define the spatial dispersion of the rate function $r$. The radial dispersion increases with distance (i.e. $\sigma_\rho$ is a function of the radius; see e.g. Barry and Burgess 2007).

The radial separation of distance bins (see *Figure 2A2*) increases linearly from 0.21 to 1.71 along the radius of length 16 distance units (corresponding to approx. 145 cm for the 2 × 2 m environment). Internal to the model a distance unit is given by $2/N_{PC}$ (see place cell resolution below). The same function is used to calculate the perceptual input to the parietal window due to objects and boundaries during simulation and to calculate activations of parietal window neurons and retrosplenial cells during the setup of the transformation circuit (see below). The receptive fields of BVCs, OVCs, PWb, PWo neurons and retrosplenial cells tile the space in polar coordinates with a radial resolution of 1 receptive fields per arbitrary distance unit (range: 0–16, see above) and an angular resolution of 51 receptive fields over $2\pi$ radians.

Similarly, to set up the PC weights in the training phase PC rates are calculated via the following equation:

$$r_{PC}^i = exp\left(\frac{(x^i - x)^2 + (y^i - y)^2}{0.5^2}\right) \tag{15}$$

where $(x, y)$ is the location of the agent and $(x_i, y_i)$ the location of the receptive field of the PC in question. The firing fields of PCs tile the environment in a Cartesian grid with resolution 0.5 (i.e. two PCs per arbitrary distance unit). However, note that during simulations PCs are never driven by this activation function. Only BVCs, PR neurons and GCs drive PCs during simulation, unlike PWb/o neurons which must receive sensory/perceptual inputs in bottom-up mode.

## Connection profiles

The encoding procedure (section Bottom-up vs top-down modes of operation) describes how object related connections are learned. The contextual representation of BVC, PC and PRb neurons, as well as the connections to and from the transformation circuit are set up in a training phase prior to running any simulations. To set up the transformation circuit randomly oriented boundary segments are chosen (20.000 times per transformation sublayer for a total of 400.000 instances), and the corresponding firing rates (calculated according to *Equation 14*) for PWb neurons and the transformation circuit sublayers are instantiated. For each transformation circuit sublayer the randomly generated activity pattern is rotated by a different angle (rotation angle chosen from 20 evenly spaced head directions). Connection weights are then calculated as outer products of the population vectors, yielding a matrix of Hebbian-like associations between the populations. The connections from the retrosplenial transformation circuit to BVCs are one-to-one connections between BVCs and the cells in each of the 20 transformation sublayers (i.e. the connections are given by the identity matrix) since this connection only needs to convey the outcome of the gain modulation across the RSC sublayers. That is, rotations of activity patterns occur on the connection to and from the parietal window. *Video 1* shows all sublayers of the transformation circuit, subject to gain modulation from HDCs as a simulated agent navigates a simple environment, see also *Figure 2—figure supplement 1*. The entire transformation of egocentric boundary inputs to BVCs effectively constitutes a model of BVC generation from sensory inputs. Finally, connections between HDCs and the 20 transformation circuit sublayers are calculated algorithmically, by associating each sublayer with one of 20 evenly spaced HD activity bumps on the head direction ring.

With a functioning transformation circuit, and after specifying the location and extent of extended boundaries in the environment, the agent is placed at a random location and orientation in the environment and the activations of BVCs and PCs are calculated via

*equations 14 and 15*. PRb activations are instantiated based on the identity of the visible landmark segments. Connection weights between these three populations (supporting the contextual representation) are again calculated as outer products of the corresponding population vectors, yielding matrices of Hebbian-like associations between the populations.

Weights are normalized such that the sum total of weights converging on a given target neuron is 1, which is assumed to be the result of some homeostatic process, a widely agreed upon feature of synaptic plasticity (*Keck et al., 2017*). Weights are scaled by scalar gain factors $\varphi_i$ (see *Appendix 1—table 1*) to produce appropriate responses in targets of afferent connections.

## Grid cell rate maps, mental navigation, and preplay setup

Grid cells are implemented as firing rate maps. Each map consists of a matrix of the same dimensions as the PC sheet (44 × 44 pixels) and is computed as 60 degrees offset, superimposed cosine waves using the following set of equations.

$$b_0 = \begin{pmatrix} cos(0) \\ sin(0) \end{pmatrix} b_1 = \begin{pmatrix} cos\left(\frac{\pi}{3}\right) \\ sin\left(\frac{\pi}{3}\right) \end{pmatrix} b_2 = \begin{pmatrix} cos\left(\frac{2\pi}{3}\right) \\ sin\left(\frac{2\pi}{3}\right) \end{pmatrix} \tag{16}$$

$$z_i = R_j b_i \left( F\vec{x} + \vec{x}_{offset} \right) \tag{17}$$

$$r_{GC} = max(0, cos(z_0) + cos(z_1) + cos(z_2)) \tag{18}$$

Here $b_0$, $b_1$ and $b_2$ are the normal vectors for the cosine waves. $Rj$ is the standard 2D rotation matrix where the index j ranges from 1 to 7 and refers to the rotation angle of the matrix (7 random orientations for 7 grid modules, here 0, π/3, π/4, π/2, π/6, 1.2π, 1.7π). *F* is the frequency of the grids, starting at 0.0028*2π. The scales of successive grids are related by the scaling factor $\sqrt{2}$ (Stensola et al. 2012). For each grid scale offsets are sampled uniformly along the principle axes of two adjacent equilateral triangles on the grid (i.e. the rhomboid made of 4 grid vertices).

Motion through GC maps (i.e. a GC sweep) during mental navigation and preplay is implemented by sampling the GC rate along the imagined trajectory superimposed on the GC rate map. The firing rate value (i.e. the pixel of the rate map) is determined by rounding the x and y values of the imagined trajectory to the nearest integer value. This sampling is equivalent to a shift of a hexagonal pattern of activation on a 2D sheet of entorhinal cells, as suggested in mechanistic models of grid cells (*Burak and Fiete, 2009*).

For simulation 5.0 (planning; *Video 13* and *Figure 13*) the reservoir place cells are supplied with random afferent connections from grid cells (sparseness 3%), and are also randomly interconnected amongst themselves (sparseness 6%). Place cells representing the familiar context and reservoir PCs inhibit each other (inhibitory connections 50% stronger than the default inhibition among place cells representing the context). Weights among reservoir place cells are normalized to the mean of the total amount of positive weights converging onto a typical place cell representing the familiar context. Grid cell weights to reservoir place cells are similarly normalized (80% stronger than default). These additions suffice to produce random, preplay-like activity in reservoir place cells as soon as the central peak of the grid cell ensemble begins to drive the reservoir. The inhibitory connections to and from the context network assure that either the reservoir place cells or the context network wins out. No changes to the adaptive feedback current are necessary ($I_{FB}$ in *Equation 3*). Finally, during preplay connections from BVCs and perirhinal neurons to place cells are turned off to avoid interference which can arise due to the very simple layout of the environment (many boundary configurations experienced by the agent are similar). No other changes to the default model are necessary. To visualize the spatio-temporal sequence of the firing of reservoir place cells during the three phases of simulation 5.0 (planning, perception, recall; see main text) the firing of reservoir cells is recorded along the imagined

or real trajectory, and stacked (rightmost panels in *Figure 13*) to yield figures akin to typical preplay/replay experiments. The firing rates are normalized and thresholded at 10% of the maximum firing rate for clarity. That is, cells that do not fire, or fire at very low rates are not shown. Due to learning during the actual traversal of the novel part of the environment (phase 2, perception) some cells can increase their firing rate above the threshold. As a consequence the number of cells which is plotted in the stacked rate maps grows marginally between phase 1 (preplay) and phase 2. However, ordering PCs in phases 2 and 3 according to the sequence derived from the preplay is done before thresholding. Hence the correct order derived from phase 1 (preplay) is applied to the cells recorded in phases 2 and 3.

## Agent and attention models

To ensure an unambiguous representation of an object at a given location (see main text) we implement a heuristic model of directed attention. A fixed length for an attentional cycle (600 ms) is allocated and divided by the number of visible objects, yielding a time per object $t_O$. The PWo population is then driven for $t_O$ ms with the cueing current $I_{PWo}^{agent}$ (see *Equation 9*) for each visible object in sequence.

The agent moves in straight lines within the environment, following a path defined by a list of coordinates. Upon reaching a target the rotation towards the next subgoal is performed, followed by the next segment of translation. The rotational velocity is implicitly given by a fixed offset of the translation weights for the HD ring attractor (approximately 18 degrees; see e.g. *Zhang, 1996*; *Song and Wang, 2005*; *Bicanski and Burgess, 2016* for more sophisticated methods of integrating rotational velocity). Translational velocity is fixed at 25 cm per second.

The agent model is agnostic about the size of the arena and nature of the agent. It can be viewed as rodent like agent or alternatively a human-like agent. The environment is covered by 44 × 44 PCs. that is 1/44 of the length/width of the environment corresponds to one distance unit. Assuming a timestep of e.g. 1ms and an arena size of approximately 2 × 2 m$^2$ for a rodent-like agent yields a translation speed of approximately 10 cm/s. Assuming a human-like agent in an environment of approximately 10 × 10 m$^2$ yields a translation speed of approximately 56 cm/s, corresponding to a slow paced walk for a human subject. In either case the speed is orders of magnitude below the time scale of neural rate dynamics.

