## [Decision Letter]

Thank you for submitting your article "A Model of Spatial Memory and Imagery – From Single Neurons to Cognition" for consideration by *eLife*. Your article has been reviewed by three peer reviewers, one of whom is a member of our Board of Reviewing Editors, and the evaluation has been overseen by Michael Frank as the Senior Editor. The reviewers have opted to remain anonymous.

The reviewers have discussed the reviews with one another and the Reviewing Editor has drafted this decision to help you prepare a revised submission.

Summary:

This paper describes a model that proposes a framework to explain how multiple, highly complex regions interact to produce egocentric and allocentric representations of space. Although all reviewers found the model to be interesting and novel, they also agreed that the Results, Materials and methods, and rationale for the model were often not presented in a clear and concise manner that would be optimal for a broad readership. Reviewers also felt that the simulation movies were insufficient to demonstrate the importance of the model and request a number of essential revisions, including incorporation of clear performance goals for the model, together with appropriate quantitative measures.

Essential revisions:

1) A primary shortcoming is that the model is not used to simulate any specific behavioral tasks. Instead, after describing the architecture, the paper presents a series of several simulations (depicted by figures and videos) to demonstrate examples of neural population activity (over time periods lasting tens of seconds) during a few selected types of navigational behavior. These example simulations do not provide adequate support for strong claims made in the Discussion section, where it is argued that the model accounts for a wide range of findings including position specificity in visual object memory (Hollingworth, 2007), impaired episodic memory resulting from Papez circuit lesions (Delay and Brion 1969), neural activity seen during imagery for scenes in the MTL, retrosplenial cortex and precuneus (Burgess, Maguire et al., 2001; Hassabis et al. 2007; Schacter et al. 2007), and 'some aspects' of 'scene construction' and 'episodic future thinking' (Schacter et al., 2007; Hassabis et al., 2007; Buckner 2010). To support such claims, the model should more faithfully reproduce experimental designs from these prior studies (as is, there does not seem to be any quantitative metric by which these broad claims can be evaluated). It is also implied that the model accounts for "trace cells" that fire when the agent visits the prior location of a missing object (Tsao et al., 2015), and although trace-like activity is shown in some example simulations, it is not specifically stated which neural population in the model would correspond to trace cells, nor are simulated trace cell firing rate maps presented. It is additionally claimed that the model accounts for preplay and replay activity of place cells, but it appears that simulated preplay and replay events do not occur on a compressed time scale in the model as they do in real rodents, which is not addressed. In other words, the heavy reliance on qualitative (rather than quantitative) performance assessments make it difficult to offer anything more than a subjective evaluation of the model's capabilities. A more objective evaluation might be possible if the authors run more simulations to quantitatively compare simulated vs. real neural activity (or simulated vs. real task performance), explore how the model's performance degrades under realistic noise or uncertainty conditions, etc.

2) Several key mechanisms – such as sequential shifting of attentional focus (which is essential for solving the place-object binding problem), neuromodulation (which allows the model to transition between sensory processing and mental imagery modes), and population activity in the grid cell map (which is essential for generating preplay/replay trajectories) – are not explicitly simulated by the model. Rather, these signals are provided "for free" as inputs to the network. When reciting the list of phenomena that the model can explain (see prior point), the authors should take care to include only phenomena that fall within the purview of what is actually being simulated by the network, rather what is being provided for free.

3) The depiction and explanation of BVC and PWb data (initially shown in Figure 2A2, C1, and C2 and also described in subsequent figures) is unclear. Why are small receptive fields shown close to the agent in Figure 2A2? What does this mean? In the Video 1, receptive fields do not appear to get smaller or bigger as the agent moves around. Are the cells with receptive fields close to the agent the ones that fire at the actual boundaries, not a distance away from the boundaries? If not, in Figure 2C1, why are BVCs shown to be firing, presumably at the north and east boundaries of the environment, when the agent is in the center of the environment (i.e., not in the boundary)? Why are so many BVCs firing at the same time when the agent is in a particular location? Are these different cells that fire at different distances from the boundaries? Figure 2A2 is described as showing receptive fields for BVCs, but these receptive fields are usually more rectangular in shape, whereas they are depicted as circular here. In "bottom up" mode (e.g., Figure 5, top) it appears that only BVCs encoding boundaries ahead of the animal (those in the visual field?) are active. Is this consistent with the firing of real neurons (such as border cells)? Is there any evidence that real BVCs are active only when a rodent faces toward but not away from their coded boundary?

4) The authors show a schematic of their model in Figure 4, which includes top-down and bottom-up connections between different brain areas. Lacking from the paper are multiple citations to anatomical studies that demonstrate that these connections are realistic (e.g., Jones and Witter, 2007, for projections from retrosplenial cortex to deep layers of entorhinal cortex). For example, has it been shown that the entorhinal cortex projects back to the retrosplenial cortex?

5) In Figure 7D, why isn't anything activated for the new object?

6) The authors go to great lengths to realistically model MTL activity based on rodent literature. However, it was not clear how the model relates to rodent literature regarding parietal coding (e.g. Harvey, Coen and Tank, 2012; Raposo, Kaufman and Churchland, 2014; Whitlock et al., 2012) and retrosplenial coding (Vedder, Miller, Harrison, Smith, Cerebral Cortex, 2017; Alexander and Nitz, 2017; Alexander and Nitz, 2015). Can the model account for the type of coding observed in navigating rodents in these regions? Or are there specific predictions the model can make about what would be observed or how these types of observed neural codes can be interpreted in the context of spatial cognition?

7) In some places, terms are used that assume prior knowledge to a degree that can make the paper difficult to parse. Some examples include 'gain-field circuit' (Introduction), the current model as an extension of the 'BBB model' (Introduction), 'heuristically implemented', 'mock-motor-efference'. Terms like this could use additional (but very brief) explanation when introduced.

8) The paper can be a bit long in places – particularly the sections on preplay and replay.

9) Figure 4: was activity from place cells to grid cells considered?

10) The presence of object specific coding appears to be a 'strong prediction' of the model, as object coding reported thus far is minimally selective for specific objects. The authors detail potential places where this activity might be found but they may want to consider that these types of representations lay outside the cortex or emerge as a population level representation (i.e. would not be observable at the level of single cell tuning curve responses).

[Editors’ note: this article was subsequently rejected after the authors submitted their revisions but they were invited to resubmit after an appeal against the decision.]

Thank you for submitting your work entitled "A Neural-Level Model of Spatial Memory and Imagery" for consideration by *eLife*. Your article has been reviewed by one peer reviewer, and the evaluation has been overseen by a Reviewing Editor and a Senior Editor. The reviewers have opted to remain anonymous.

We regret to inform you that your work will not be considered further for publication in *eLife*.

After consultation, we feel that the paper remains too cumbersome for a general audience, and some points remain unclear.

Reviewer #3:

In this paper, Bicanski and Burgess present a network model to propose how multiple populations of spatially tuned neurons are functionally interconnected with one another to form a spatial memory network with multiple capabilities.

A core feature of the model is that it proposes how bidirectional transformations between the egocentric and allocentric reference frames might be performed by networks in the retrosplenial cortex. The paper ambitiously attempts to demonstrate numerous capabilities for the model, such as mapping a familiar environment, learning the locations of unique landmark objects in such an environment, detecting novelty when changes occur in an environment, and simulating effects of brain lesions upon memory and navigation. The paper incorporates 13 videos, 13 figures, and 18 methods equations. Despite its length (nearly 60 pages), the paper does not provide enough detail for readers to fully understand some key aspects of the model (see below). This should *not* be interpreted as an entreaty to make the paper even longer. Rather, the paper should probably be split up into at least two publications, each dedicated to exploring specific capabilities of the model with more clarity and depth.

As noted in the prior review, simulations of mental navigation and route planning do not seem to be rooted in the model's core ability to perform bidirectional egocentric / allocentric transformations. Instead, it is the addition ("for free") of a grid cell network connected to place cells endows the model with this capability. How does the egocentric / allocentric transformation network contribute to the mental navigation and route-planning process? What is the functional purpose for transforming allocentric replay events back into the egocentric coordinate frame to generate imagery? The grid cell driven simulations do not even appear until the subsection “Grid cells and mental navigation (Simulation 4.0)”, and given how information dense the preceding pages are, perhaps the addition of the grid cell network and accompanying simulations might be better suited to a second paper, dedicated more specifically to the topic of route planning?

Another problem raised in review, which has not been adequately addressed, is that the sequential shifting of attentional focus is not explicitly simulated by the model, and not explained fully enough for readers to comprehend how this process contributes to the simulation results. To solve the problem of binding each object or boundary's location to its identity, the model adopts a sequential shift of attention strategy. If I understand correctly, each population of boundary (BVC) or object (OVC) cells is activated one at a time in conjunction with its corresponding boundary (PRb) or object (PRo) identity cell. The length of each attentional cycle is stated to be 600 time units. How does information about multiple objects and boundaries get integrated across multiple attentional cycles to form a stable and non-fluctuating representation of the environment as a whole? Do PCs have slow activity decay kinetics that can span multiple attentional cycles? How is such temporal integration accounted for in analyses like that shown in Figure 8, where momentary "snapshots" of population vectors are being correlated with one another across encoding vs. recall? Because of the sequential attention mechanism, one would expect that an incomplete representation of the environment (i.e., just one object and one boundary) would be active at any given time. So I don't quite see how it is possible to use an instantaneous snapshot of the population vector to perform these correlation analyses, unless the snapshot is being taken at the end of some temporal integration process that spans multiple attention cycles?

Finally, since the heart of the model is the retrosplenial transformation network, it is a bit surprising that no simulation results are shown to demonstrate the predicted firing properties of retrosplenial neurons that perform the egocentric / allocentric transformation (in either direction) during imagery and recall. Do any testable predictions for unit recording studies arise from the firing properties of these model neurons? If so, then it seems like they should be included in the paper.

In summary, there are some innovative features of this model that would of interest to the research community, but the format of the paper (including its length) may not be best suited for publication in *eLife*. The authors might wish to consider splitting the paper up into two publications (e.g., one on coordinate transformation, and one on mental navigation and route following) so that some of the missing details of the model can be more thoroughly described without adding even more to the current manuscript's excessive length.

[Editors’ note: what now follows is the decision letter after the authors submitted for further consideration.]

Thank you for resubmitting your work entitled "A Neural-Level Model of Spatial Memory and Imagery" for further consideration at *eLife*. Your revised article has been favorably evaluated by Michael Frank (Senior Editor), a Reviewing Editor, and one new reviewer.

The manuscript has been improved but there are some remaining issues that need to be addressed before acceptance, as outlined below:

Reviewer #1:

In the revised manuscript, the authors have done a lot of work to respond to previous criticisms, including adding new results to respond to the initial criticisms.

There is much value and novelty in this model, which provides a potential explanation of how egocentric and allocentric representations of space are reconciled. Also, as the authors point out in their appeal, the excessive length of the manuscript is due to results that were added to satisfy the original reviews. I feel confident that the specific feedback provided below will allow the authors to clarify points that remain puzzling.

1) I didn't really understand what the "transformation sublayers" are. Are they between areas or in the retrosplenial cortex part of the model? How are they different from "individual sublayers" (subsection “The Head Direction Attractor Network and the Transformation Circuit”, second paragraph)?

2) In all of the figures and videos involving the top-down mode/recall/imagery, why does firing occur for all directions of boundaries in PWbs and BVCs?

3) In Figure 12A, why are OVCs to the east firing when object 2 is in the northeast of allocentric space?

4) In Video 8, I don't understand at time = 1.67 seconds how the OVC pattern matches that during perception for object 2. It looks like object 1. OVC activity at NW at 1.67 seconds looks the same as OVC activity at NW at 6.52 seconds, the time at which the text states that activity matches that during perception for object 1. Related to this, in Video 12, at time = 11.52 s, OVC do not seem to be representing object 1 (i.e., it is not in the SE). The mental navigation representation of object 3 in OVC at 15.21 seconds looks the same as the mental navigation representation of object 1 at time = 11.36 seconds.

---

## [Author Response]

Summary:This paper describes a model that proposes a framework to explain how multiple, highly complex regions interact to produce egocentric and allocentric representations of space. Although all reviewers found the model to be interesting and novel, they also agreed that the Results, Materials and methods, and rationale for the model were often not presented in a clear and concise manner that would be optimal for a broad readership. Reviewers also felt that the simulation movies were insufficient to demonstrate the importance of the model and request a number of essential revisions, including incorporation of clear performance goals for the model, together with appropriate quantitative measures.

We hope our changes and additions to the manuscript in response to the comments provided remedy any criticisms. We now replicate specific experimental findings to supplement the previously submitted simulations and movies and expand the quantitative treatment of results. Please see in particular our extensive response to comment 1.

Quantification

In addition to the replication of several experimental paradigms (i.e. replication of behaviour/phenomenology), we have introduced a quantitative measure to assess the performance of the model. Our model’s main claim is that it constitutes a model of spatial memory and imagery specified at the level of single neurons. Hence we now quantify the similarity between the model representations during encoding and during recall, by correlating the population vectors during these two distinct states for the model’s populations. That is, we now measure the extent of re-activation of population vectors to quantify the extent of memory or novelty being signaled by the various neural populations. The magnitude of the correlation is compared to correlations of the recalled pattern to randomly sampled times while the model operates in bottom-mode (i.e. during perception). Author response image 1 shows an example from simulation 1.0 (object-cued recall). This measure can also be compared between simulations with and without noise (see end of replies to comment 1). Perirhinal neurons are not shown because of their small number (4 for the 4 walls, 1 for the object) and because the correlation is trivially 1 when cueing with the object (current is injected into the object-specific perirhinal neuron to induce recall).

**Author response image 1. respfig1:** Example of pattern comparison via correlations of population vectors from simulation 1.0 (object cued recall). White bars show the correlation between the neural patterns during imagery/recall and those during encoding (RvE), while black bars show the average correlation between the neural patterns during imagery/recall and random patterns (sampled every 100 ms; RvRP). Note that OVCs and PCs exhibit correlation values close to one, indicating faithful reproduction of patterns. BVC correlations are somewhat diminished because recall fully reactivates all boundaries indiscriminately compared to a limited field of view during perception with only modest reactivation outside the field of view. PW neurons show correlations below one because at recall reinstatement in parietal areas requires the egocentric allocentric transformation (i.e. OVC signals passed through retrosplenial cells), which smears out the pattern compared to perceptual instatement in the parietal window (i.e. imagined representations are not as precise as their counterparts during perception).

This quantification of the mismatch between population vectors reflecting perceived and recalled representations could potentially be compared to experimental measures of overlap between neuronal populations (e.g. Guzowski et al., 1999 in animals, or ‘representational similarity’ measures in fMRI, e.g. Ritchey et al., 2013) if perception and recall can be separated experimentally. We use the same measure to infer the extent of novelty-related exploration (see response 1A, and original submission) and to assess robustness with regards to noise (please see reply section 1G).

Comment 1 requires the most comprehensive response and we split our reply in multiple parts (A-H) to focus on the individual points raised.

Essential revisions:1) A primary shortcoming is that the model is not used to simulate any specific behavioral tasks. Instead, after describing the architecture, the paper presents a series of several simulations (depicted by figures and videos) to demonstrate examples of neural population activity (over time periods lasting tens of seconds) during a few selected types of navigational behavior. These example simulations do not provide adequate support for strong claims made in the Discussion section, where it is argued that the model accounts for a wide range of findings including position specificity in visual object memory (Hollingworth, 2007), impaired episodic memory resulting from Papez circuit lesions (Delay and Brion 1969), neural activity seen during imagery for scenes in the MTL, retrosplenial cortex and precuneus (Burgess, Maguire et al., 2001; Hassabis et al. 2007; Schacter et al. 2007), and 'some aspects' of 'scene construction' and 'episodic future thinking' (Schacter et al., 2007; Hassabis et al., 2007; Buckner 2010). To support such claims, the model should more faithfully reproduce experimental designs from these prior studies (as is, there does not seem to be any quantitative metric by which these broad claims can be evaluated).

We agree with the referees that our more general simulations (designed to showcase the capabilities of the model) should be supplemented by replications of published experiments to better link our claims to the cited literature. We have added several items to the Results section, replicating behaviour/phenomenology from both human and rodent studies, focusing and on Papez’ circuit lesions and object novelty respectively. We have also restructured the section on top-down activity and trace cells, conducting new simulations replacing the previous simulations 2.1-4.

1A) Object novelty, hippocampal vs. perirhinal lesions

We now simulate experiments showing that hippocampal vs. perirhinal lesions differentially affect change detection, and suggest that object vector cells play a crucial role. In experiments the effects of lesions are commonly tested by exploiting the spontaneous preference that rats display for exploring novel/altered stimuli compared to familiar/unchanged ones. We assume that the relative amount of exploration will be proportional to the mismatch signal in the model (exploiting the introduction of a quantitative measure, please see above), and simulate key aspects of the following study.

Mumby et al. (2002) tested (among other manipulations) whether rats can detect that one of two objects has been shifted to a new location in a given environment. We have added simulations of this type of task (a moved object) with and without an intact hippocampus to the manuscript (simulations 1.3 and 1.4). The agent experiences an environment with two objects in which one is later moved. The mismatch signal is quantified as the difference in the firing of object vector cells during bottom-up mode (perception at the location of encoding) and during recall (imagery, at the same location).

Intact hippocampus:With an intact hippocampus (Simulation 1.3 and Video 7) the agent can generate a meaningful novelty signal, since the perceived location of the object relative to the agent is different from the recalled position. We compare the mismatch signal between the two objects in the arena and set the next movement target for the agent (to have a visual manifestation of the outcome of the comparison in the video) to the object for which the biggest mismatch has been detected (lowest correlation of population vectors). Figure 9 depicts the crucial aspects of this simulation and has been added to the manuscript as an application of the model to a real task. That detection of a change in position requires the hippocampus is consistent with place cells binding the relative location of an object (via object vector cells) to perirhinal neurons signalling the identity of an object.

Hippocampal lesion:Hippocampal lesions (Simulation 1.4 and Video 8) are implemented by setting the firing rates of hippocampal neurons to zero at all times. A hippocampal lesion precludes the generation of a meaningful novelty signal because the agent is incapable of generating a coherent point of view, since no appropriate BVC configuration can be selected by the now missing hippocampal input. A connection between object vector cells and perirhinal neurons (see Figure 3D in original manuscript) can still form in the lesioned agent. Thus upon recall some OVC activity is present (in imagery) due to a connection between perirhinal neurons and OVCs. However, such activity is not location specific. That is, without the reference frame of BVCs/PCs (place cells selecting active BVCs) this residual spurious activity cannot specify absolute object location since it can be generated everywhere (see Figure 2F-H). It only tells agent it has seen this object at given relative distance, but not where. Hence the mismatch signal is equal for both objects, and consequently exploration time would be split roughly 50/50 between the two objects.

On rare occasions, by chance, an object could be at the correct distance for the agent to give a good match between the perceived location and the unanchored OVC activity elicited by perirhinal neurons in imagery (Figure 9G, H). However, such an incidental match can occur for both the un-moved, as well as the moved object. Note that this account is corroborated by human data, which shows that some focal hippocampal amnesics can recall a familiar arrangement of objects if the point of view between test phase and sample phase is matched (King et al., 2002; but see also Shrager et al., 2007).

Contextual effects:Rats show preferential exploration of a familiar object that was previously experienced in a different environment, compared with one previously experienced in the same environment, and this preference is also abolished by hippocampal lesions (Mumby et al., 2002; Eacott and Norman 2004; Langston and Wood 2009). We have not simulated different environments (using separate place cell ensembles), but note that the ‘remapping’ of place cells between distinct environments (i.e. much reduced overlap of place cell population activity, Bostock, Muller and Kubie, 1991; Andersen and Jeffery 2003; Wills et al., 2005) suggests a mismatch signal for the changed-context object can be derived by comparing PC population vectors. Initiating recall of object 1, belonging to context 1, in context 2, would drive the place cells ensemble belonging to context 1, creating an imagined scene in context 1. During perception place cells representing context 2 would instead continue to be active. That is, since two environments would be represented by distinct place cells ensembles in the model (approximating remapping conditions) this would trivially generate a mismatch signal, and a hippocampal lesion obviously precludes such a mismatch signal specific to either the same-context or changed-context object.

Perirhinal lesions:Finally, it has been argued that object recognition (irrespective of context) is spared after hippocampal lesions but not perirhinal lesions (Aggleton and Brown, 1999; Winters et al., 2004; Norman and Eacott, 2004). These results are certainly compatible with the model given that its perirhinal neuronal population signals an object’s identity irrespective of location.

1B) The effect of Papez’ Circuit lesions

The second type of experimental finding we now replicate is the effect of lesions along Papez’ Circuit, outside of the hippocampus, in causing amnesia (Delay and Brion 1969; Squire and Slater 1978, Squire et al., 1989; Parker and Gaffan 1997; Aggleton et al., 2016). Lesions to the fornix and mammillary bodies severely impact recollection, although recognition is less affected (Tsivillis et al., 2008). Although Papez’ Circuit includes more regions than are covered by our model, in the context of spatial representations Papez’ circuit brain areas are notable for containing head direction cells. That is, the mammillary bodies (more specifically the lateral mammillary nucleus, LMN), anterior dorsal thalamus, retrosplenial cortex, parts of the subicular complex and medial entorhinal cortex. Thus lesioning Papez’ circuit corresponds to (at least) removing the head direction signal in our model. This signal is crucial for the operation of the transformation circuit. (In reality the disruption of inputs from the medial and lateral septum, conveying theta rhythmicity and cholinergic tone could also be important, but are not part of the current model.)

We have conducted new simulations to illustrate the ability of the model to account for the known results of such lesions (Simulations 1.1, 1.2 and Videos 5, 6). The lesion is modelled by setting the input provided by head direction cells to the retrosplenial transformation circuit to zero at all times.

In the bottom-up mode of operation (perception) a lesion to the head direction cell ensemble removes drive to the transformation circuit and consequently to the boundary vector cells and object vector cells. That is, the perceived location of an object (present in the egocentric parietal representation) cannot elicit activity in the MTL and thus cannot be encoded into memory. I.e. there is no BVC/OVC activity which could be associated with currently active place cells. In the top-down mode of operation (recall)there are two effects. 1) Since no new elements can be encoded into memory there is nothing to recall (anterograde amnesia) (Simulation 1.1), and 2) Even if there are pre-existing memories (e.g. of an object encoded prior to the lesion) and place cells can be driven via a learned connection from perirhinal neurons (e.g. when cued with the object encoded prior to the lesion; Simulation 1.2), no meaningful representation can be instantiated in parietal areas, preventing recollection/imagery, thus causing retrograde amnesia for hippocampus-dependent episodic memories (other forms of retrieval of neocortical memories would not necessarily be affected). Simulations 1.1 and 1.2 show that the egocentric neural correlates of objects and boundaries present in the visual field would persist in the parietal window only as long the agent perceives them (they could also be held in working memory, which we do not model). Figure 7 depicts the key elements of simulations 1.1 and 1.2.

Note that perirhinal cells and any presumed upstream ventral visual stream inputs are spared, and hence an agent could still report the identity of an attended object.

Also note that grid cell firing would also be disrupted by removal of head direction inputs (Winter, Clark and Taube, 2015).

Winter SS, Clark BJ, Taube JS. Disruption of the head direction cell network impairs the parahippocampal grid cell signal. Science. 2015 Feb 5:1259591

1C) Position specificity and Hollingworth (2007)

In Hollingworth (2007) participants viewed images of natural scenes or object arrays in a change detection task and memory performance was higher when the target object position remained the same from study to test. This advantage was lost when the relative spatial relationships among contextual objects was disrupted. The author concludes that episodic scene representations are formed through the binding of objects to scene locations, and object position is defined relative to a larger spatial representation coding the relative locations of contextual objects (also see Hannula et al., 2006). This seems like a very good match to the role played by object vector cells (OVCs) in our model. OVCs are bound to a context via place cells, which means that all objects, when represented by OVC firing (in conjunction with perirhinal identity cells), preserve their relative arrangements (see e.g. Simulations 1.3, 1.4, and 3.0 for two objects encoded from the same position).

Hannula DE, Tranel D, Cohen NJ. The long and the short of it: relational memory impairments in amnesia, even at short lags. Journal of Neuroscience. 2006 Aug 9;26(32):8352-9.

The rationale behind the inclusion of this citation on our part was that transferring Hollingworth’s paradigm into 3D (e.g. overlooking an arrangement of objects in a 3D scene), maps perfectly onto the toy environment our virtual agent moves in, and would provide a good example of the use of object vector cells. In fact, detecting a change of an object’s position within the scene maps 1:1 to the simulation conducted for 1A, which exploits position specificity conferred by object vector cells. Hence we hope the referees agree that the inclusion of the object novelty paradigm described in 1A warrants the citation of Hollingworth (2007).

1D) 'Scene construction' and 'episodic future thinking'

We address the concerns about our suggestion that the model serves as an explanation for some aspects of 'scene construction' and 'episodic future thinking' separately.

First, we clarify that the model provides a potential explanation for the neural activity seen in MTL, retrosplenial cortex and precuneus (e.g. Hassabis, Kumaran and Maguire, 2007) during imagery for previously experienced scenes (i.e. when subjects were instructed to perform recall of a past event) in terms of the overlap between the model structures and the reported brain regions (see also, Burgess et al., 2001; Schacter, Addis and Buckner, 2007). However, we now note that the explanatory power of the model with regards to imagined future (or potentially non-existent) scenes is more limited (please also see our brief discussion of episodic future thinking below).

With regard to the neural activity seen in experiments during imagery for scenes in the MTL Hassabis, Kumaran and Maguire (2007) note that “the imaginary constructions produced by four of the five patients ([hippocampal amnesics]) were greatly reduced in richness and content compared with those of controls. The impairment was especially pronounced for the measure of spatial coherence, indicating that the constructions of the amnesic patients tended to consist of isolated fragments of information, rather than connected scenes.” It is currently impossible to map our quantification onto subjective, participant-dependent metrics like ‘spatial coherence’ as used in Hassabis, Kumaran and Maguire (2007). However, we have shown that a hippocampal lesion (see 1A) at best allows for the association of fragments of a scene (e.g. perirhinal cells being linked OVCs). No coherent scene can be constructed in the model without hippocampal place cells binding all scene elements together. We would cautiously suggest that such a deficit could potentially underlie such lack of spatial coherence reported by amnesics in Hasssabis, Kumaran and Maguire (2007) with regard to previously experienced scenes.

Concerning the model providing an explanation for some aspects of 'episodic future thinking' (Schacter, Addis and Buckner, 2007; Hassabis, Kumaran and Maguire, 2007; Buckner 2010), we apologise if this was overstated. Episodic future thinking explicitly links memory with planning for future behaviour. “Brain regions that have traditionally been associated with memory appear to be similarly engaged when people imagine future experiences” Schacter, Addis and Buckner (2007). Our model does describe how spatial scenes can be imagined, and how this process relies on structures traditionally associated with memory. However, our simulations do not include imagination of completely novel scenes, and we now take care to state that the concept of episodic future thinking extends beyond our simulations. Nevertheless, our simulations do include imagination of novel trajectories within a familiar environment, which could be useful for planning future actions. For example, in simulation 4.0 (Figure 12; mental navigation), the agent imagines performing new trajectory and object 3 appears in the egocentric frame of reference in a relative position to the agent not experienced during encoding. We believe this constitutes a small (albeit crucial) step towards a neural account of episodic future thinking. We are now more careful in our phrasing regarding claims about episodic future thinking.

It is also implied that the model accounts for "trace cells" that fire when the agent visits the prior location of a missing object (Tsao et al., 2015), and although trace-like activity is shown in some example simulations, it is not specifically stated which neural population in the model would correspond to trace cells, nor are simulated trace cell firing rate maps presented.

1E) Trace cells

The reviewers are right that the topic of trace cells merits some clarification. In response to this comment we have restructured the section of the manuscript previously devoted to novelty and top-down activity. Novelty is now covered separately, by simulations 1.3,1.4 (replication of Mumby et al. 2002). What used to be simulation set 2 (simulations 2.1-4 in the previous version of the manuscript) is now devoted to a more in-depth discussion of trace responses.

The trace property is not necessarily tied to a specific anatomical area or cell type. We use the term trace cell to refer to any cell that used to fire in response to an object or boundary and subsequently fires in its absence. As requested we now show examples of rate maps which predict trace fields as a result of repeated memory probes which may occur roughly with the frequency of theta in rodents.

We define a memory probe as follows: a smooth modulation of top-down connectivity during perception, subject to the following criteria: 1) Sensory inputs are not disengaged, and 2) there is no cue to recall anything specific (like for instance an object in its context). This kind of probe of MTL representations can inform the perceptually driven egocentric parietal representation about absent scene elements (please see main manuscript text and Figure 10A, B). The exact duration of the probe is not important as long as it leaves enough time for activity to build up in the MTL. For the trace response simulations the memory probe occurs with a frequency of rodent theta (though no mechanism of theta generation is modelled) because it has previously been suggested that theta orchestrates the flow of information (Hasselmo, Bodelón and Wyble, 2002), at least in rodents.

However, note that some trace cell firing could also result from non-spatially selective cells. We now show that nominally non-spatially selective cells like perirhinal identity neurons can manifest a spatial trace firing field if encoding is restricted to a limited region of space (cf. Figure 10D3). This finding may help to reconcile the common notion that lateral entorhinal cortex processes non-spatial information (Van Cauter et al., 2012) with spatial responses of trace cells (Tsao, Moser and Moser, 2013) in lateral entorhinal cortex. However, unlike perirhinal neurons in the model the trace cells of Tsao et al. (2015) do not fire when the object is present, but only subsequently, in the absence of the object. Thus they might signal the mismatch between the expected presence of an object and its actual absence. However, crucially such a signal may be based on trace firing of cells similar to perirhinal cells in the model. I.e. an object specific cell which was active at the time of encoding must necessarily participate in the generation of a mismatch signal.

It is additionally claimed that the model accounts for preplay and replay activity of place cells, but it appears that simulated preplay and replay events do not occur on a compressed time scale in the model as they do in real rodents, which is not addressed.

1F) Replay and Preplay

We agree that the section on “replay” and “preplay” would benefit from a more careful presentation. We simplified the corresponding section considerably (which also addresses point 8 below, shortening the section), making clear the limitations of the model to explain these types of phenomena.

The same results are reported as before but we more correctly refer to them as simulations of planning and recall. The main aim of simulation 5.0 (Figure 13) was to show how the model can cope with shortcuts, extending its cognitive map into un-explored territory by incorporating new place cells from a reservoir. That is, the agent engages in planning, performing mental navigation along the shortest path to the goal by using a forward sweep of grid cell activity to drive corresponding activity in established place cells (across familiar ground) or in proto-place cells (across unexplored ground). We now simply point out that the internally-generated sequential activity of place cells is reminiscent of forward replay or preplay (referring to ‘preplay-like’ activity) but do not claim (and state so explicitly) to have fully modelled those phenomena.

The planning stage engenders preplay-like activity in place cells (Dragoi and Tonegawa 2011; Olafsdottir et al., 2016), whereas the mental navigation phase is reminiscent of replay (Wilson and McNaughton 1994; Foster and Wilson 2006; Diba and Buzsaki 2007; Karlsson and Frank 2009; Carr, Jadhav and Frank 2011) or ‘forward sweeps (Johnson and Redish 2007; Pfeiffer and Foster 2015), although the faster propagation of these sequences compared to those seen during actual navigation is not addressed. Both replay and preplay usually occur during sharp wave ripple events (at a compressed time scale). To account for sharp wave ripple events and theta sequences requires a spiking neuron model. Nevertheless, the causal role played by grid cells in the model suggests that some forms of replay in place cells could be driven by a sweep of activity in the grid cell population along a given trajectory, and may correspond to route planning (Kubie and Fenton 2012; Erdem and Hasselmo 2012; Bush et al. 2015; Yamamoto and Tonegawa 2017).

It is possible that a compressed time scale could be simulated if grid cell sweeps occur at faster timescales within SWRs or theta cycles, and without re-activation of parietal representations. But such extensions are outside the scope of the model (spiking neurons, theta rhythmicity, SWRs) and are not simulated.

In other words, the heavy reliance on qualitative (rather than quantitative) performance assessments make it difficult to offer anything more than a subjective evaluation of the model's capabilities. A more objective evaluation might be possible if the authors run more simulations to quantitatively compare simulated vs. real neural activity (or simulated vs. real task performance), explore how the model's performance degrades under realistic noise or uncertainty conditions, etc.

1G) Comparison to real tasks, quantification, and noise

We hope to have addressed the reviewers’ main concerns regarding quantification and comparisons to real tasks by replicating several finding reported in the literature (please see 1A, B, E), and by providing a quantitative measure of the model’s performance (correlation of population vectors). We have also provided more examples of neural activity shown in the form of rate maps (trace cells) for comparison with experimental data. We note that the design of the model was also constrained by known data in terms of the tuning curves used for spatial cells and the assumed anatomical connections between regions.

Noise/robustness

As far as noise and robustness are concerned, we perform two additional sets of simulations (all modifications of simulation 1.0, object-cued recall). In the first simulation set we randomly chose cells throughout the model to be permanently deactivated (equal proportions of place cells, grid cells, OVCs, BVCs, parietal and retrosplenial neurons etc.). This allows us to assess the robustness of the model’s capability to perform encoding and recall in visuo-spatial imagery with regards to neuron loss. Simulations with neuron loss could also serve as a model for the effects of diffuse damage, as might occur in anoxia, Alzheimer’s disease or normal aging, although we do not attempt to model any specific neurological condition. We have repeated this simulation 20 times (each time with newly drawn random numbers) and present the average correlation between the population vectors at encoding vs. recall in Figure 8.

We have tested silencing up to 20% of the cells per affected model component. The agent is still able to encode and recall the corresponding representations (cf. Figure 8). The ability to maintain a stable attractor state among place cells and head direction cells is critical to the functioning of the model, while damage in the remaining (feed-forward) model components manifests in gradual degradation in the ability to represent the locations of objects and boundaries (please see accompanying Video 3). For example, if certain parts of the parietal window suffer from neuron loss, the reconstruction in imagery is impaired only at the locations in peri-personal space which were coded for by the now missing neurons (indeed, this can be the basis for a model of hemi-spatial neglect, Byrne, Becker and Burgess 2007). The place cell populations were more robust to silencing than the head-direction population, simply because they were simulated in comparatively greater numbers of neurons and therefore exhibit redundancy.

Similarly, the model is also robust to adding firing rate noise (up to 20% of peak firing rate) to all cells. Correlations between patterns at encoding and recall remain similar to the noise-free case. Figure 8 shows the correlation between the population vectors at encoding vs. recall for all model components for 20 iterations. We include one representative example from both simulation sets as new videos (Videos 3 and 4).

2) Several key mechanisms – such as sequential shifting of attentional focus (which is essential for solving the place-object binding problem), neuromodulation (which allows the model to transition between sensory processing and mental imagery modes), and population activity in the grid cell map (which is essential for generating preplay/replay trajectories) – are not explicitly simulated by the model. Rather, these signals are provided "for free" as inputs to the network. When reciting the list of phenomena that the model can explain (see prior point), the authors should take care to include only phenomena that fall within the purview of what is actually being simulated by the network, rather what is being provided for free.

We wholeheartedly agree with the reviewers. It was certainly not our intention to claim that the model explains the mechanistic origins of attention, how grid cell firing patterns are generated by the brain, or the exact nature of the proposed neuromodulatory signal. We now emphasise more strongly in the main text that these phenomena are not explained by the model at a mechanistic level. However, we believe it remains fair to say that the model proposes what role attention plays is memory encoding. This statement can be divorced from the mechanistic origins of attention and thus constitutes a valid prediction. Similarly, assuming grid cell-like inputs to the model (e.g. in the form of pre-calculated firing rate maps) allows for predictions about the role of grid cells (e.g. in mental navigation, driving the point of view) without needlessly replicating one of the numerous articles that have proposed models of the mechanistic origin of grid cell signals.

3) The depiction and explanation of BVC and PWb data (initially shown in Figure 2A2, C1, and C2 and also described in subsequent figures) is unclear. Why are small receptive fields shown close to the agent in Figure 2A2? What does this mean? In the Video 1, receptive fields do not appear to get smaller or bigger as the agent moves around. Are the cells with receptive fields close to the agent the ones that fire at the actual boundaries, not a distance away from the boundaries? If not, in Figure 2C1, why are BVCs shown to be firing, presumably at the north and east boundaries of the environment, when the agent is in the center of the environment (i.e., not in the boundary)? Why are so many BVCs firing at the same time when the agent is in a particular location? Are these different cells that fire at different distances from the boundaries? Figure 2A2 is described as showing receptive fields for BVCs, but these receptive fields are usually more rectangular in shape, whereas they are depicted as circular here. In "bottom up" mode (e.g., Figure 5, top) it appears that only BVCs encoding boundaries ahead of the animal (those in the visual field?) are active. Is this consistent with the firing of real neurons (such as border cells)? Is there any evidence that real BVCs are active only when a rodent faces toward but not away from their coded boundary?

We have endeavoured to clarify these aspects in the first presentation of boundary vector cell and PW firing. Figure 2 was poorly explained previously and we take care now to be more precise. In particular the relationship between receptive field centers mapped on a polar grid and the firing of individual cells is now hopefully clearer.

Further clarifications

- The videos and figures never show receptive fields (with the exception of Figure 2A2 now), but neural firing of cells arranged according to where in space their receptive fields are centered.

- In the previously submitted version of the manuscript, receptive fields only scaled along the angular axis with distance (constant angular resolution) because we had modelled PW neurons and BVCs after the previously published model of Byrne, Becker and Burgess (2007). However, a better fit to the known data on BVCs is that the centers of receptive fields should also exhibit increasing radial separation as shown in panel A2 now. This has now been remedied. Similarly, the radial dispersion of the receptive fields (given by σ_rho in Equation 14) now also scales with distance from the center, such that far away receptive fields still cover multiple bins/grid points. We have rerun all model simulations with this updated topology of polar receptive fields (see Appendix), which provides a closer fit to Barry and Burgess (2007) and Lever et al. (2009).

- There are BVCs for all distances, as suggested by the arrangement of the receptive fields as a polar grid around the agent. This is why BVCs fire for boundaries of the environment, when the agent is e.g. in the center of the environment.

- Many BVCs firing at the same time collectively represent all the currently perceived boundaries. That is, for extended boundaries multiple grid points/receptive fields are covered by a boundary and hence multiple cells fire. Without extra allocation of attention to specific portions of a boundary BVC firing thus represents, to a first approximation, a context of the environment (in conjunction with place cells).

- The shape of the receptive is not rectangular but rather given by the product of exponentials in Equation 14. Panel A2 hopefully makes this clearer.

- To emphasise the clear distinction between perception and recall/imagery we have restricted perceptual activity to the field of view. To the best of our knowledge there currently is not enough data to assess if BVCs or border cells fire only when a boundary is actively perceived as long an animal is not engaged in recall. However note that boundaries outside the field of view are weakly represented due to pattern completion (see for instance in panel C1). A periodically recurring memory probe as outlined for the simulation that yielded BVC trace rate maps (see 1E) can increase this effect further.

4) The authors show a schematic of their model in Figure 4, which includes top-down and bottom-up connections between different brain areas. Lacking from the paper are multiple citations to anatomical studies that demonstrate that these connections are realistic (e.g., Jones and Witter, 2007, for projections from retrosplenial cortex to deep layers of entorhinal cortex). For example, has it been shown that the entorhinal cortex projects back to the retrosplenial cortex?

We thank the referees for catching this oversight. We now include proper citations for all the connections. Jones and Witter (2007) report that “projections from the remaining cingulate areas [including retrosplenial cortex] preferentially target the postrhinal and medial entorhinal cortices as well as the presubiculum and parasubiculum”. Since we make no strong claims about the locus of OVCs we believe this reference is compatible with the present model, regardless of whether BVCs/OVCs are preferentially located in the subicular complex (Lever et al., 2009; with OVC co-located by analogy) or in medial entorhinal cortex (Solstad et al., 2008; Høydal et al., 2017).

Regarding the back projection to retrosplenial cortex, Wyss and van Groen (1992) have reported such connections. We briefly discuss these connections when we discuss the possible locus of OVCs.

5) In Figure 7D, why isn't anything activated for the new object?

Figure 7D depicted the neural representations during imagery and the novel object had not yet been encoded into memory. In addition, if it had been, it would have needed to be in the focus of attention to be accompanied by, for instance, perirhinal activity. For a similar simulation, note that, if it had been encoded in addition to an old, removed object, the situation would have been covered by what was previously Figure 8 (sampling multiple items in imagery, now Figure 11). However, we have restructured the corresponding manuscript section to focus on trace cells, which can be compared to experimental findings and which yield explicit predictions.

6) The authors go to great lengths to realistically model MTL activity based on rodent literature. However, it was not clear how the model relates to rodent literature regarding parietal coding (e.g. Harvey, Coen and Tank, 2012; Raposo, Kaufman and Churchland, 2014; Whitlock et al., 2012) and retrosplenial coding (Vedder, Miller, Harrison, Smith, Cerebral Cortex, 2017; Alexander and Nitz, 2017; Alexander and Nitz, 2015). Can the model account for the type of coding observed in navigating rodents in these regions? Or are there specific predictions the model can make about what would be observed or how these types of observed neural codes can be interpreted in the context of spatial cognition?

We agree with the reviewers that the rodent literature concerning the parietal side of the model has been neglected in the text. We have instead been more inspired by the human side of the literature for this model component. This is mainly due to the fact that neurons resembling parietal window cells, as presented in the model, have only recently been reported (Hinman, Chapman and Hasselmo 2017), compared to vast literature on place cells, grid cells, head direction cells, and, to some extent, boundary (vector) cells. Some findings, e.g. the firing patterns correlated with multiple frames of reference (e.g. Wilber et al., 2014) have parallels with the model (in this case with our retroplenial model component. However, although intriguing correlations have been reported in parietal cortex (Nitz 2006, 2009, 2012; Harvey, Coen and Tank 2012; Whitlock et al., 2012; Raposo, Kaufman and Churchland 2014; Vedder et al., 2016) it remains difficult (compared to place cells or grid cells) to identify a unified representational code underlying all of the reported recordings. It is however an intriguing topic for future work. Also because trying to incorporate these findings would considerably inflate the already lengthy manuscript.

7) In some places, terms are used that assume prior knowledge to a degree that can make the paper difficult to parse. Some examples include 'gain-field circuit' (Introduction), the current model as an extension of the 'BBB model' (Introduction), 'heuristically implemented', 'mock-motor-efference'. Terms like this could use additional (but very brief) explanation when introduced.

We now clarify these terms upon first occurrence. The BBB abbreviation has been removed since it only occurred once.

8) The paper can be a bit long in places – particularly the sections on preplay and replay.

We have shortened these sections. We realize that the rest of the paper is a bit long in places and have endeavoured to be more concise where possible, within the constraints of the requested corrections and additional work.

9) Figure 4: was activity from place cells to grid cells considered?

We did not consider this type of activity for multiple reasons. 1) On a practical level, since grid cells are implemented as firing rate maps it is not clear how any influence of place cells on grid cells could be implemented. 2) On a theoretical level, it has been argued that place cells (when driven by sensory inputs) can help anchor grid cells, which exhibit drift due to path integration errors. Regarding the dynamics of the model, since grid cells are represented by static firing rate maps, there was no need for stabilizing place cell inputs to grid cells. We believe the functional meaning of the connection from place cells to grid cells is not relevant for the model in its current form, although intriguing new work on the issue (e.g. grid cells performing PCA on place cell representations; Dordek et al., 2016) might warrant a closer look in future work.

10) The presence of object specific coding appears to be a 'strong prediction' of the model, as object coding reported thus far is minimally selective for specific objects. The authors detail potential places where this activity might be found but they may want to consider that these types of representations lay outside the cortex or emerge as a population level representation (i.e. would not be observable at the level of single cell tuning curve responses).

We are not quite sure of the question here. Our model does indeed take into account that ‘object coding reported thus far is minimally selective for specific objects.’ A given object vector cell fires for any object, as long as an object is at a certain distance in a certain direction: it is not selective for specific objects. This is a key prediction, similar to the original prediction that BVCs respond to any boundary at the correct distance and direction (Hartley et al., 2000). Object specific coding is only present in the perirhinal identity cells, and there are many reports of this type of coding there (e.g. Miller et al., 1993; Xiang and Brown 1998; Zhu et al., 1995). Only the conjunction of OVC firing and perirhinal identity cells uniquely defines a specific object at a specific place relative to the agent. Uniquely specifying absolute position requires place cells as well. Thus, we agree that, although individual cells have well-defined tuning curves, it is a population code of OVCs, PCs and perirhinal cells which uniquely defines the neural representation of an object in its place.

Miller, E. K., Li, L. & Desimone, R. Activity of neurons in anterior inferior temporal cortex during a short-term memory task. J. Neurosci. 13, 1460–1478 (1993).

Xiang, J. Z. & Brown, M. W. Differential neuronal encoding of novelty, familiarity and recency in regions of the anterior temporal lobe. Neuropharmacology 37, 657–676 (1998).

Zhu, X. O., Brown, M. W. & Aggleton, J. P. Neuronal signalling of information important to visual recognition memory in rat rhinal and neighbouring cortices. Eur.J. Neurosci. 7, 753–765 (1995).

[Editors’ note: the authors’ responses to first round of re-review follow]

Reviewer #3:In this paper, Bicanski and Burgess present a network model to propose how multiple populations of spatially tuned neurons are functionally interconnected with one another to form a spatial memory network with multiple capabilities.A core feature of the model is that it proposes how bidirectional transformations between the egocentric and allocentric reference frames might be performed by networks in the retrosplenial cortex. The paper ambitiously attempts to demonstrate numerous capabilities for the model, such as mapping a familiar environment, learning the locations of unique landmark objects in such an environment, detecting novelty when changes occur in an environment, and simulating effects of brain lesions upon memory and navigation. The paper incorporates 13 videos, 13 figures, and 18 methods equations. Despite its length (nearly 60 pages), the paper does not provide enough detail for readers to fully understand some key aspects of the model (see below). This should not be interpreted as an entreaty to make the paper even longer. Rather, the paper should probably be split up into at least two publications, each dedicated to exploring specific capabilities of the model with more clarity and depth.As noted in the prior review, simulations of mental navigation and route planning do not seem to be rooted in the model's core ability to perform bidirectional egocentric / allocentric transformations. Instead, it is the addition ("for free") of a grid cell network connected to place cells endows the model with this capability. How does the egocentric / allocentric transformation network contribute to the mental navigation and route-planning process? What is the functional purpose for transforming allocentric replay events back into the egocentric coordinate frame to generate imagery? The grid cell driven simulations do not even appear until the subsection “Grid cells and mental navigation (Simulation 4.0)”, and given how information dense the preceding pages are, perhaps the addition of the grid cell network and accompanying simulations might be better suited to a second paper, dedicated more specifically to the topic of route planning?

Contrary to the reviewer’s new statement (not mentioned in the first review), mental navigation and route planning do indeed require the model's core ability to perform bidirectional egocentric / allocentric transformations. As stated multiple times throughout the manuscript one of the core hypotheses of the present model is that an agent only has conscious access to the egocentric representation of space around itself, which embodies a point of view. This hypothesis is strongly supported by striking, classical empirical findings like hemispatial neglect. Thus, updating the (imagined) location of the agent by the action of grid cells on place cells in the medial temporal lobe still requires the egocentric-allocentric transformation to allow imagery of the simulated outcome. Without this transformation, there would be no mental navigation.

Another problem raised in review, which has not been adequately addressed, is that the sequential shifting of attentional focus is not explicitly simulated by the model, and not explained fully enough for readers to comprehend how this process contributes to the simulation results. To solve the problem of binding each object or boundary's location to its identity, the model adopts a sequential shift of attention strategy. If I understand correctly, each population of boundary (BVC) or object (OVC) cells is activated one at a time in conjunction with its corresponding boundary (PRb) or object (PRo) identity cell. The length of each attentional cycle is stated to be 600 time units. How does information about multiple objects and boundaries get integrated across multiple attentional cycles to form a stable and non-fluctuating representation of the environment as a whole? Do PCs have slow activity decay kinetics that can span multiple attentional cycles? How is such temporal integration accounted for in analyses like that shown in Figure 8, where momentary "snapshots" of population vectors are being correlated with one another across encoding vs. recall? Because of the sequential attention mechanism, one would expect that an incomplete representation of the environment (i.e., just one object and one boundary) would be active at any given time. So I don't quite see how it is possible to use an instantaneous snapshot of the population vector to perform these correlation analyses, unless the snapshot is being taken at the end of some temporal integration process that spans multiple attention cycles?

With regard to the details of the attentional modulation, the OVCs are subject to the attentional modulation because we simulate the encoding of novel objects into a familiar context. Attention is required at encoding to allow the relevant representations to be bound together (in this case OVCs, place cells and object-identity neurons). At retrieval, the associations have been formed, and a complete scene representation is stably present. Population vectors are sampled at the time of encoding, which is a precise moment in simulation: there is no need for averaging across a cycle because the encoded object is in the focus of attention at the time of encoding. These population vectors are then compared to the stable representations during retrieval/imagery.

Finally, since the heart of the model is the retrosplenial transformation network, it is a bit surprising that no simulation results are shown to demonstrate the predicted firing properties of retrosplenial neurons that perform the egocentric / allocentric transformation (in either direction) during imagery and recall. Do any testable predictions for unit recording studies arise from the firing properties of these model neurons? If so, then it seems like they should be included in the paper.

The firing properties of retrosplenial neurons have been shown in Video 1, and we would happily add a static supplementary figure (e.g. of rate maps) to formulate an additional prediction.

In summary, there are some innovative features of this model that would of interest to the research community, but the format of the paper (including its length) may not be best suited for publication in eLife. The authors might wish to consider splitting the paper up into two publications (e.g., one on coordinate transformation, and one on mental navigation and route following) so that some of the missing details of the model can be more thoroughly described without adding even more to the current manuscript's excessive length.

In addition, the second round reviewer requests full models of attention, grid cells, and parietal cortex. We do not think that would be possible, even if split over 2 papers. More importantly, we do not see how the absence of these additional models details diminishes the thorough and comprehensive theoretical account of spatial memory we have proposed.

[Editors’ note: the author responses to the re-review follow.]

Thank you for resubmitting your work entitled "A Neural-Level Model of Spatial Memory and Imagery" for further consideration at eLife. Your revised article has been favorably evaluated by Michael Frank (Senior Editor), a Reviewing Editor, and one new reviewer.The manuscript has been improved but there are some remaining issues that need to be addressed before acceptance, as outlined below:Reviewer #1:In the revised manuscript, the authors have done a lot of work to respond to previous criticisms, including adding new results to respond to the initial criticisms.There is much value and novelty in this model, which provides a potential explanation of how egocentric and allocentric representations of space are reconciled. Also, as the authors point out in their appeal, the excessive length of the manuscript is due to results that were added to satisfy the original reviews. I feel confident that the specific feedback provided below will allow the authors to clarify points that remain puzzling.1) I didn't really understand what the "transformation sublayers" are. Are they between areas or in the retrosplenial cortex part of the model? How are they different from "individual sublayers" (subsection “The Head Direction Attractor Network and the Transformation Circuit”, second paragraph)?

The transformation sublayers are the same as the individual sublayers. The whole transformation circuit consists of 20 individual sublayers (please see Video 1 and the new supplementary figure, Figure 2—figure supplement 1), each maximally modulated by a different head direction. We apologise for the confusion and have standardized the terminology throughout, using the term ‘sublayer’ consistently with retrosplenial cortex. The term ‘circuit’ is used to refer to the whole of the retrosplenial model component, and the generic term ‘layer’ is replaced by the term ‘population’. We have also partially rewritten (and shortened) subsection “The Head Direction Attractor Network and the Transformation Circuit” to further clarify the structure of the retrosplenial component further.

2) In all of the figures and videos involving the top-down mode/recall/imagery, why does firing occur for all directions of boundaries in PWbs and BVCs?

This is because the simulated environment is familiar, so that the agent has experienced it from many different points of view at each location. Thus there are reciprocal connections between place cells and BVCs for all boundaries, for a given location. Since, during imagery, PWb neurons are driven by BVC activity (via the transformation circuit) they fire for boundaries in all directions (see Figure 2—figure supplement 1), even though the PWb neurons behind the agent would not be driven by perception. We now make extra note of this at the first occurrence of recall/imagery (near Figure 5).

3) In Figure 12A, why are OVCs to the east firing when object 2 is in the northeast of allocentric space?

The object is in the North East of the map, but relative to the agent it lies East. Hence OVCs representing the location of object 2 are firing East of the origin of the OVC plot because the object is East of the agent. The OVC grid of receptive fields (like the BVC grid) is anchored to the agent (i.e. the origin is the location of the agent and moves with it). The allocentric aspect lies in the independence of BVC/OVC responses with regard to the orientation of the agent (e.g. whether the object is to the left or the right). We now make an extra note of this in the discussion of receptive field topology (in the caption of Figure 2). Hopefully the new Figure 2—figure supplement 1 also helps to clarify the nature of these receptive fields.

4) In Video 8, I don't understand at time = 1.67 seconds how the OVC pattern matches that during perception for object 2. It looks like object 1. OVC activity at NW at 1.67 seconds looks the same as OVC activity at NW at 6.52 seconds, the time at which the text states that activity matches that during perception for object 1. Related to this, in Video 12, at time = 11.52 s, OVC do not seem to be representing object 1 (i.e., it is not in the SE).

In Video 8 the time in imagery (during the 1^st^ recall episode) can be divided into 2 phases. From roughly 1.60 to 2.70s object 1 is being recalled (with corresponding OVC pattern). From 2.70 to 2.80s object 2 is being recalled (with corresponding OVC pattern). Only for object two there is a match to the perceptual representation (experienced prior to recall). The confusion is likely due to the fact that the annotation appears already at time 1.60 but actually refers mainly to phase two, when object 2 is recalled after object 1. We apologize for the inaccuracy and have now split the annotation to be more precise.

The mental navigation representation of object 3 in OVC at 15.21 seconds looks the same as the mental navigation representation of object 1 at time = 11.36 seconds.

This is again due to the fact that the OVC grid (similar to the BVC grid) translates with the agent, but is independent of the orientation of the agent. At both times (around 11.35s and around 15.21s) the object is roughly South East to the agent at a similar distance (though not precisely at the same angle) but, both times with different place cell representations, which disambiguate the representational code overall. We hope the changes we made in response to comment 3 also clarify this case.